# A human embryonic limb cell atlas resolved in space and time

Bao Zhang[1,21], Peng He[2,3,21], John E. G. Lawrence[3,4,21], Shuaiyu Wang[1,5,21], Elizabeth Tuck[3], Brian A. Williams[6], Kenny Roberts[3], Vitalii Kleshchevnikov[3], Lira Mamanova[3,18], Liam Bolt[3,19], Krzysztof Polanski[3], Tong Li[3], Rasa Elmentaite[3], Eirini S. Fasouli[3,20], Martin Prete[3], Xiaoling He[7,8], Nadav Yayon[2,3], Yixi Fu[1], Hao Yang[1], Chen Liang[1], Hui Zhang[9], Raphael Blain[10], Alain Chedotal[10,11,12], David R. FitzPatrick[13], Helen Firth[3], Andrew Dean[14], Omer Ali Bayraktar[3], John C. Marioni[2,3], Roger A. Barker[7,8], Mekayla A. Storer[8], Barbara J. Wold[6], Hongbo Zhang[1,15,16 ✉] & Sarah A. Teichmann[3,17 ✉]

Human limbs emerge during the fourth post-conception week as mesenchymal buds, which develop into fully formed limbs over the subsequent months[1]. This process is orchestrated by numerous temporally and spatially restricted gene expression programmes, making congenital alterations in phenotype common[2]. Decades of work with model organisms have defined the fundamental mechanisms underlying vertebrate limb development, but an in-depth characterization of this process in humans has yet to be performed. Here we detail human embryonic limb development across space and time using single-cell and spatial transcriptomics. We demonstrate extensive diversification of cells from a few multipotent progenitors to myriad differentiated cell states, including several novel cell populations. We uncover two waves of human muscle development, each characterized by different cell states regulated by separate gene expression programmes, and identify musculin (MSC) as a key transcriptional repressor maintaining muscle stem cell identity. Through assembly of multiple anatomically continuous spatial transcriptomic samples using VisiumStitcher, we map cells across a sagittal section of a whole fetal hindlimb. We reveal a clear anatomical segregation between genes linked to brachydactyly and polysyndactyly, and uncover transcriptionally and spatially distinct populations of the mesenchyme in the autopod. Finally, we perform single-cell RNA sequencing on mouse embryonic limbs to facilitate cross-species developmental comparison, finding substantial homology between the two species.

Human limb buds emerge by the end of the fourth post-conception week (PCW) and develop to form arms and legs during the first trimester. By studying model organisms such as the mouse and the chick, it is known that development of the limb bud begins in the form of two major components. The parietal lateral plate mesodermal (LPM) cells condense into the skeletal system as well as forming tendon, fibrous and smooth muscle populations, whereas skeletal muscle progenitor (SkMP) cells migrate from the paraxial mesoderm to the limb field, forming striated muscle[3]. The mesoderm is encapsulated within a thin layer of ectoderm, a subset of which (the apical ectodermal ridge) governs mesenchymal proliferation and aids in the establishment of the limb axes through fibroblast growth factor (FGF) signalling[4]. Limb maturation continues in a proximal–distal manner, controlled by a complex system of temporally and spatially restricted gene expression programmes, in which small perturbations can result in profound changes to the structure and function of the limb[1,5]. Indeed, approximately 1 in 500 humans are born with congenital limb malformations[2]. Although model organisms have provided key insights into cell fates

[1]The Key Laboratory for Stem Cells and Tissue Engineering, Ministry of Education, Zhongshan School of Medicine, Sun Yat-sen University, Guangzhou, China. [2]European Molecular Biology Laboratory, European Bioinformatics Institute (EMBL-EBI), Wellcome Genome Campus, Hinxton, UK. [3]Wellcome Sanger Institute, Wellcome Genome Campus, Hinxton, UK. [4]Department of Trauma and Orthopaedics, Cambridge University Hospitals NHS Foundation Trust, Addenbrooke's Hospital, Cambridge, UK. [5]Department of Obstetrics, Guangzhou Institute of Pediatrics, Guangzhou Women and Children's Medical Center, Guangzhou Medical University, Guangzhou, China. [6]Division of Biology and Biological Engineering, California Institute of Technology, Pasadena, CA, USA. [7]John van Geest Centre for Brain Repair, Department of Clinical Neurosciences, University of Cambridge, Cambridge, UK. [8]Wellcome-MRC Cambridge Stem Cell Institute, University of Cambridge, Cambridge, UK. [9]Institute of Human Virology, Key Laboratory of Tropical Disease Control of Ministry of Education, Zhongshan School of Medicine, Sun Yat-sen University, Guangzhou, China. [10]Sorbonne Université, INSERM, CNRS, Institut de la Vision, Paris, France. [11]Institut de pathologie, groupe hospitalier Est, hospices civils de Lyon, Lyon, France. [12]University Claude Bernard Lyon 1, MeLiS, CNRS UMR5284, INSERM U1314, Lyon, France. [13]MRC Human Genetics Unit, IGC, University of Edinburgh, WGH, Edinburgh, UK. [14]Department of Clinical Neurosciences, Cambridge University Hospitals NHS Foundation, Cambridge, UK. [15]Advanced Medical Technology Center, the First Affiliated Hospital, Zhongshan School of Medicine, Sun Yat-sen University, Guangzhou, China. [16]Department of Histology and Embryology, Zhongshan School of Medicine, Sun Yat-sen University, Guangzhou, China. [17]Theory of Condensed Matter Group, Department of Physics, Cavendish Laboratory, University of Cambridge, Cambridge, UK. [18]Present address: Enhanc3D Genomics Ltd, Cambridge, UK. [19]Present address: Genomics England, London, UK. [20]Present address: Basic Research Center, Biomedical Research Foundation, Academy of Athens, Athens, Greece. [21]These authors contributed equally: Bao Zhang, Peng He, John E. G. Lawrence, Shuaiyu Wang. ✉e-mail: zhanghongbo@mail.sysu.edu.cn; st9@sanger.ac.uk

and morphogenesis, how precisely their biology translates to human development and disease remains unclear. The lack of complementary spatial information in such studies further precludes the assembly of a comprehensive tissue catalogue of human limb development.

Here we performed single-cell transcriptomic RNA sequencing (scRNA-seq) and spatial transcriptomic sequencing to detail the development of the human hindlimb (or lower limb) in space and time. We identified 67 distinct cell clusters from 125,955 captured single cells, and spatially mapped them across four first trimester timepoints to shed new light on limb development. At PCW8, we applied VisiumStitcher to map cells to a sagittal section of an entire fetal hindlimb. In addition, our spatial transcriptomic data provide insights into the key patterning events in the developing limb, with a focus on genes associated with limb malformation. We performed scRNA-seq on mouse embryonic limbs to compare limb development across species, revealing extensive homology between a classical model organism and human. Our data can be freely accessed at https://developmental.cellatlas.io/embryonic-limb.

## Cellular heterogeneity of the developing limb in space and time

To track the contribution of the different lineages to the developing limb, we collected single-cell embryonic limb profiles from PCW5 to PCW9 (Fig. 1a and Supplementary Table 5). In total, we analysed 125,955 single cells that passed quality control filters and identified 67 distinct cell clusters (Fig. 1b, Supplementary Table 1 for marker genes and Extended Data Fig. 1a,b). Thirty-four clusters were derived from the LPM. They contain mesenchymal, chondrocyte, osteoblast, fibroblast and smooth muscle cell states, consistent with previous studies[6]. A further eight states formed the muscle lineage, derived from the somite. Other non-LPM cell clusters included haematopoietic ($n = 14$), endothelial ($n = 3$), neural crest-derived ($n = 5$) and epithelial ($n = 3$), including one apical ectodermal ridge-like cluster, which broadly expressed *SP8* and *WNT6* and contained nine *FGF8*⁺ cells originating from PCW5 and PCW6 (Extended Data Fig. 2a–d), supported by RNA in situ hybridization (RNA-ISH) (Extended Data Fig. 2e). The cellular composition of the developing limb changed markedly over time; progenitor states for each lineage were chiefly dominating in PCW5 and PCW6, with more differentiated cell states emerging thereafter (Extended Data Fig. 3a,b).

To give spatial context to this cellular heterogeneity, we performed spatial transcriptomic experiments using the 10x Visium assay, generating high-quality transcriptomic profiles for samples from PCW5 to PCW8 (Extended Data Fig. 1c). We then deconvolved Visium voxels against the scRNA-seq data (see Methods; Extended Data Fig. 1d for quality control). This demarcated the tissue sections into distinct regions, separating three distal progenitor populations that we named 'distal' (*LHX2*⁺*MSX1*⁺*SP9*⁺), '*RDH10*⁺distal' (*RDH10*⁺*LHX2*⁺*MSX1*⁺) and 'transitional' (*IRX1*⁺*MSX1*⁺) mesenchyme (Fig. 1c,d and Extended Data Fig. 4a–d,f,g). The distal mesenchymal cells are located at the distal periphery of the limb. Proximal to it are the transitional mesenchyme together with chondrocyte progenitors of the developing autopod (Fig. 1c). Although all of them sit in proliferative regions (Extended Data Fig. 4e), subtle transcriptomic differences exist, with the distal mesenchyme expressing digit patterning genes, including *LHX2* and *TFAP2B* (Fig. 1c and Extended Data Fig. 4a–d). Mutations in *TFAP2B* cause Char syndrome, a feature of which is postaxial polydactyly[7]. The *RDH10*⁺ distal mesenchyme strongly expresses *RDH10*, encoding the primary enzyme of retinaldehyde synthesis, which is critical in interdigital cell death[8] (Extended Data Fig. 4f,g). The transitional mesenchyme expresses *IRX1* and *IRX2*, which are key genes in digit formation and chondrogenic boundary definition[9] (Fig. 1c and Extended Data Fig. 4d,f). We further examined the distributions of these genes at PCW5 and PCW6 in three dimensions using tissue clearing and light-sheet fluorescence microscopy (Supplementary Video 1). *IHH*⁺ prehypertrophic

chondrocytes (PHCs) localized to the mid-diaphysis of the forming tibia and the metatarsals (Fig. 1c). At the proximal limit of the sample, both *MEIS2*⁺*WT1*⁺ proximal mesenchymal cells and *CITED1*⁺ mesenchymal cells (Mes3) were observed (Fig. 1c and Extended Data Fig. 4f).

At PCW8, we placed three anatomically continuous sections from the hindlimb on separate capture areas of the same Visium chip. We subsequently integrated data from this chip to obtain a spatial transcriptomic readout of a complete sagittal section of the limb (Fig. 1d and Extended Data Fig. 5a). At this stage, articular chondrocytes mapped to the articular surfaces of the developing joints, whereas osteoblasts mapped to the mid-diaphyseal bone collar of the tibia and femur. Perichondrial cells matched to a comparable region, although they extended along the full length of the tibia and femur (Fig. 1d), a finding confirmed by immunofluorescence staining for *RUNX2* and *THBS2* alongside *COL2A1* (Extended Data Fig. 5b). *COL10A1*⁺ hypertrophic chondrocytes (HCCs) mapped to the mid-diaphysis of the tibia (Fig. 1d). Glial cells expressing myelin genes (Extended Data Fig. 1b) were co-located with a *FOXS1*⁺ fibroblast subtype ('neural fibroblast') in the periphery of the sciatic and tibial nerves (Fig. 1d and Extended Data Fig. 5c–e). We captured few neurons ($n = 28$) in our single-cell data, probably due to the distant location of their cell bodies within the spinal ganglia.

Cell states with related (but not identical) transcriptomic profiles did not necessarily occupy the same location. For example, one group of three fibroblast clusters were co-located with basal and periderm cells[10], prompting their annotation as dermal fibroblasts (DermFiB) and their precursors (*F10*⁺DermFiBP and *HOXC5*⁺DermFiBP)(Extended Data Fig. 2f,g). Conversely, another group of two fibroblast clusters expressing ADH family members (*ADH*⁺Fibro, InterMusFibro) colocalized with muscle cells (Extended Data Fig. 2h–j). Similarly, two clusters expressed the tendon markers scleraxis (*SCX*) and tenomodulin (*TNMD*), with one expressing the extracellular matrix genes biglycan (*BGN*) and keratocan (*KERA*), whereas the other expressed pro-glucagon (*GCG*) (Extended Data Fig. 5f,g). The former cluster matched to the hamstrings, quadriceps and patellar tendons, whereas the latter cluster matched to the perimysium surrounding the muscles (Fig. 1d and Extended Data Fig. 5h). We therefore annotated these clusters as tenocyte and perimysium, respectively. This integrated analysis serves as an example of how spatial transcriptomic methodologies can improve our understanding of tissue architecture and locate cell states within a dynamic anatomical structure such as the developing limb.

## Patterning, morphogenesis and developmental disorders in the limb

We utilized spatial transcriptomic data to investigate patterning genes and found consistency with classical expression patterns in model organisms (Extended Data Fig. 6a–e). This included genes that govern proximal identity, such as *MEIS1*, *MEIS2*, *PBX1* and *IRX3*, as well as genes regulating limb outgrowth and distal morphogenesis such as *WNT5A*, *GREM1*, *ETV4* and *SALL1* (ref. 5) (Extended Data Fig. 6b,c). Similarly, classical mammalian anterior–posterior genes were captured, including *HAND1*, *PAX9*, *ALX4* and *ZIC3* (anterior) and *HAND2*, *SHH*, *PTCH1* and *GLI1* (posterior)[5,11–15] (Extended Data Fig. 6d,e). Our spatial transcriptomic data captured the expression patterns of the HOXA and HOXD gene clusters at PCW5.6 (Extended Data Fig. 6f). As expected, their expression matched the second wave of Hox expression in mice, with a loss of asymmetry in the HoxA cluster and its maintenance in the HoxD cluster, which showed increased restriction to the posterior margin of the limb with increased 5′ position. For both clusters, an increase in group number corresponded to more distally restricted expression, with group 13 genes limited to the autopod. An exception was *HOXA11*, which showed no overlap with *HOXA13*, in keeping with their expression patterns in mice. Our data revealed a flip to the antisense transcript of *HOXA11* in the distal limb (Extended Data Fig. 6f),

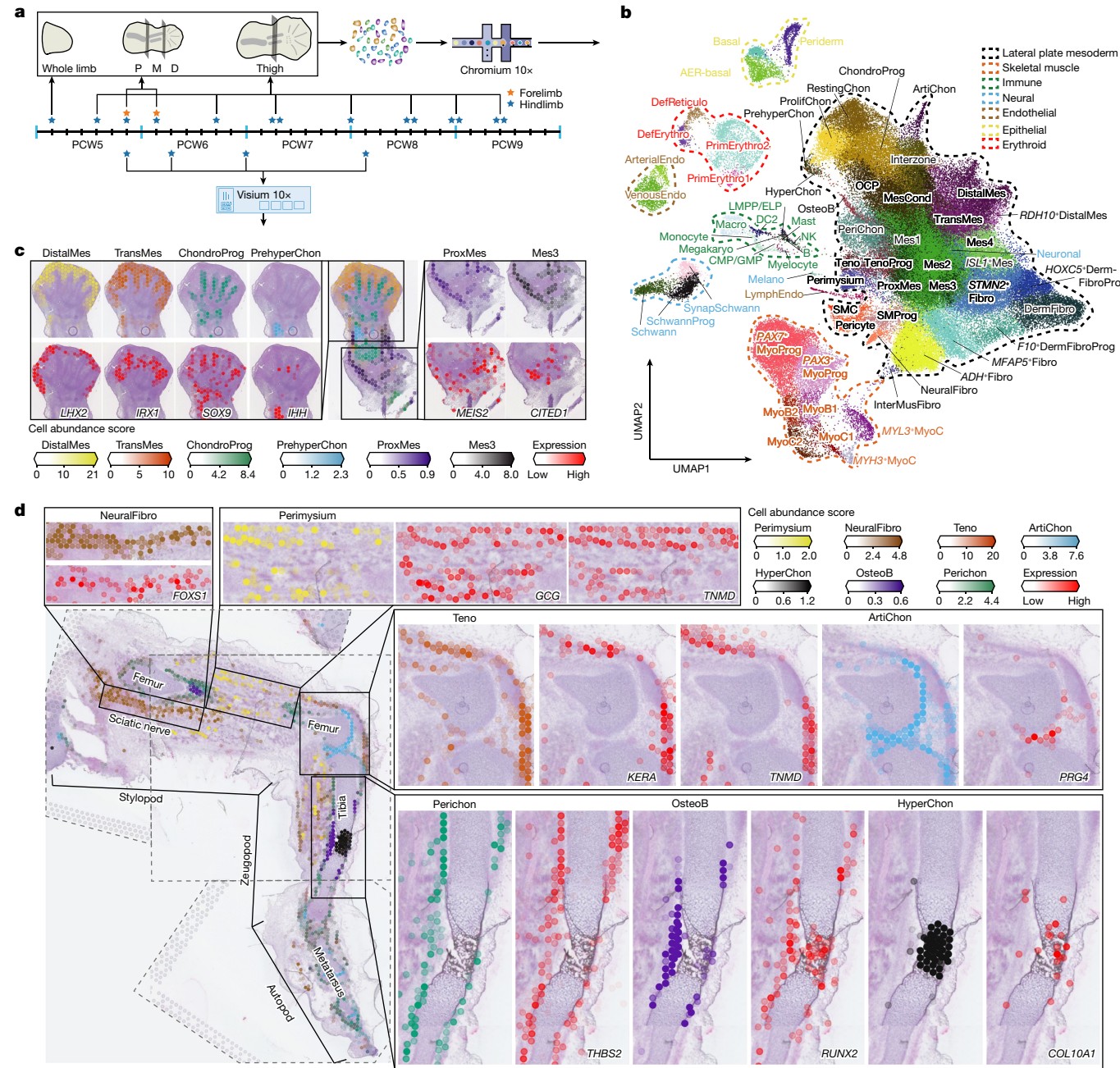

**Fig. 1 | A single-cell temporal–spatial atlas of the human embryonic limb.**
**a**, Overview of samples and the experimental scheme. The stars indicate
timepoints. D, distal; M, middle; P, proximal. **b**, Uniform manifold approximation
and projection (UMAP) visualization of 125,955 human embryonic limb cells
with cluster labels (see Supplementary Information). **c**,**d**, Spatially resolved
heatmaps across tissue sections from the PCW6.2 (**c**) and PCW8.1 (**d**) human
hindlimbs assembled from three slides showing population abundance and
corresponding marker genes.

which may be due to HOXA13- and HOXD13-dependent activation of
the *HOXA11-AS* enhancer[16].

To investigate gene expression patterns during digit formation,
we obtained coronal sections through a PCW6.2 foot plate to reveal
the forming digits and interdigital space (IDS; Fig. 2a). We annotated
digital, interdigital and distal mesenchyme and other regions (Fig. 2a;
see Methods). Differential expression testing between the digital space
and IDS demonstrated an enrichment of genes involved in IDS cell
death, such as *BMP7*, *BMP2* and *ADAMTS1* (refs. 17,18) (Fig. 2b). IDS
also showed an enrichment of *RDH10* and *CRABP1*, whereas *CYP26B1*,
encoding a retinoic acid-metabolizing enzyme, was upregulated in
the digital regions, highlighting the role of retinoic acid in triggering
IDS cell death[19] (Fig. 2b,c). Other digit-specific genes included *TGFB2*,

a vital molecule in interphalangeal joint specification, and *WWP2*, a
regulator of chondrogenesis[20,21]. In addition, *PIEZO2*, which promotes
bone formation via calcium-dependent activation of NFATc1, YAP1 and
β-catenin, was restricted to the digits, together with *C1QL1* (encoding a
calcium-binding molecule), which correlates with *COL2A1* expression
during in vitro chondrogenesis[22,23] (Fig. 2b).

Finally, we annotated each digit to search for genes that vary with
digit identity (Extended Data Fig. 6i). Anterior genes *ID2* and *ZNF503*, as
well as the regulators of cell proliferation *PLK2* and *LEMD1* (refs. 24–26)
were upregulated in the great toe; whereas *HOXD11* was downregulated
as is found in mice and chicks. We found no differentially expressed
genes in the remaining digits, although statistical power was limited
by the sample size.

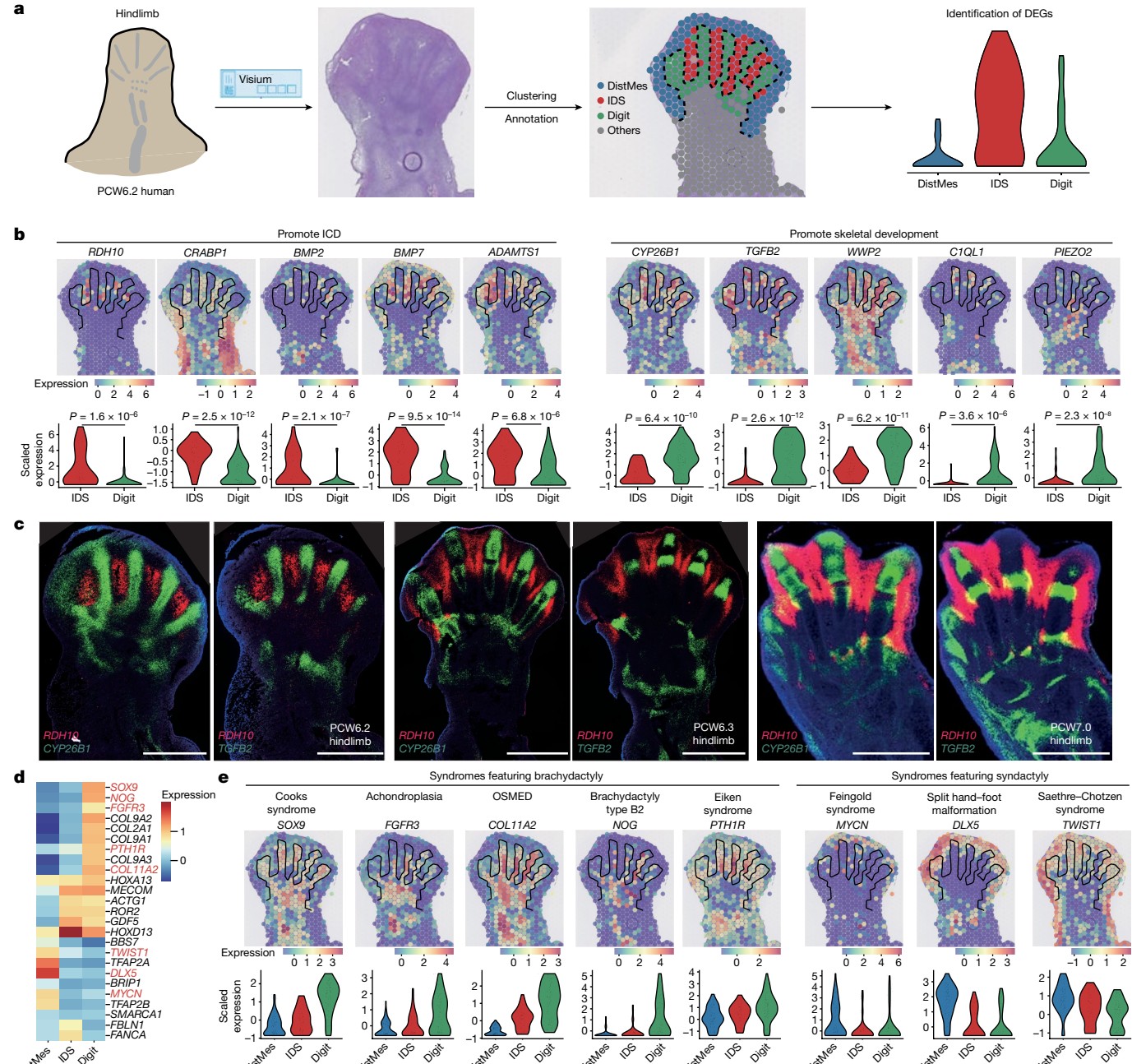

**Fig. 2 | Spatial expression pattern of genes involved in digit formation and phenotype. a**, Scheme to identify genes involved in digit formation and interdigital cell death (ICD). DEG, differentially expressed gene; DistMes, distal mesenchyme; IDS, interdigital space. **b,e**, Spatial expression (normalized and log-transformed) of genes promoting ICD (**b**, left panel) and digital tissue survival (**b**, right panel) and genes associated with digit malformation

(**e**) in the PCW6.2 human hindlimb, and their distributions in IDS and digit regions (*P* values were determined by Wilcoxon rank sum test). OSMED, otospondylomegaepiphyseal dysplasia. **c**, RNA-ISH of *RDH10*, *CYP26B1* and *TGFB2* in the human hindlimb. Scale bars, 1 mm. **d**, Heatmap showing the expression (*Z* scores) of genes associated with digit malformation.

We next cross-referenced the list of spatial differentially expressed genes against a list of 2,300 genetic conditions. Genes involved in several types of non-syndromic brachydactyly were upregulated in the digits, including *NOG* (brachydactyly type B2), *PTH1R* (Eiken syndrome), *COL11A2* (otospondylomegaepiphyseal dysplasia), *SOX9* (Cooks syndrome) and *FGFR3* (achondroplasia)[27–29] (Fig. 2d,e and Supplementary Table 2 for all differentially expressed genes). Conversely, genes implicated in syndromes featuring syndactyly were significantly upregulated in the IDS and distal mesenchyme. These include *DLX5* (split hand–foot malformation), *MYCN* (Feingold syndrome type 1) and *TWIST1* (Saethre–Chotzen syndrome)[30–33]. Where mouse models

of these conditions exist, their phenotype is broadly comparable with the human models (Supplementary Table 3). Thus, our spatial atlas provides a valuable reference of gene expression under homeostatic conditions for comparison with genetic variations for which phenotypes may begin to penetrate during embryonic development.

## Regulation of cell-fate decisions of mesenchymal-derived lineages

To better understand what may control their specification, we inferred cellular trajectories in the mesenchyme-associated states by combining

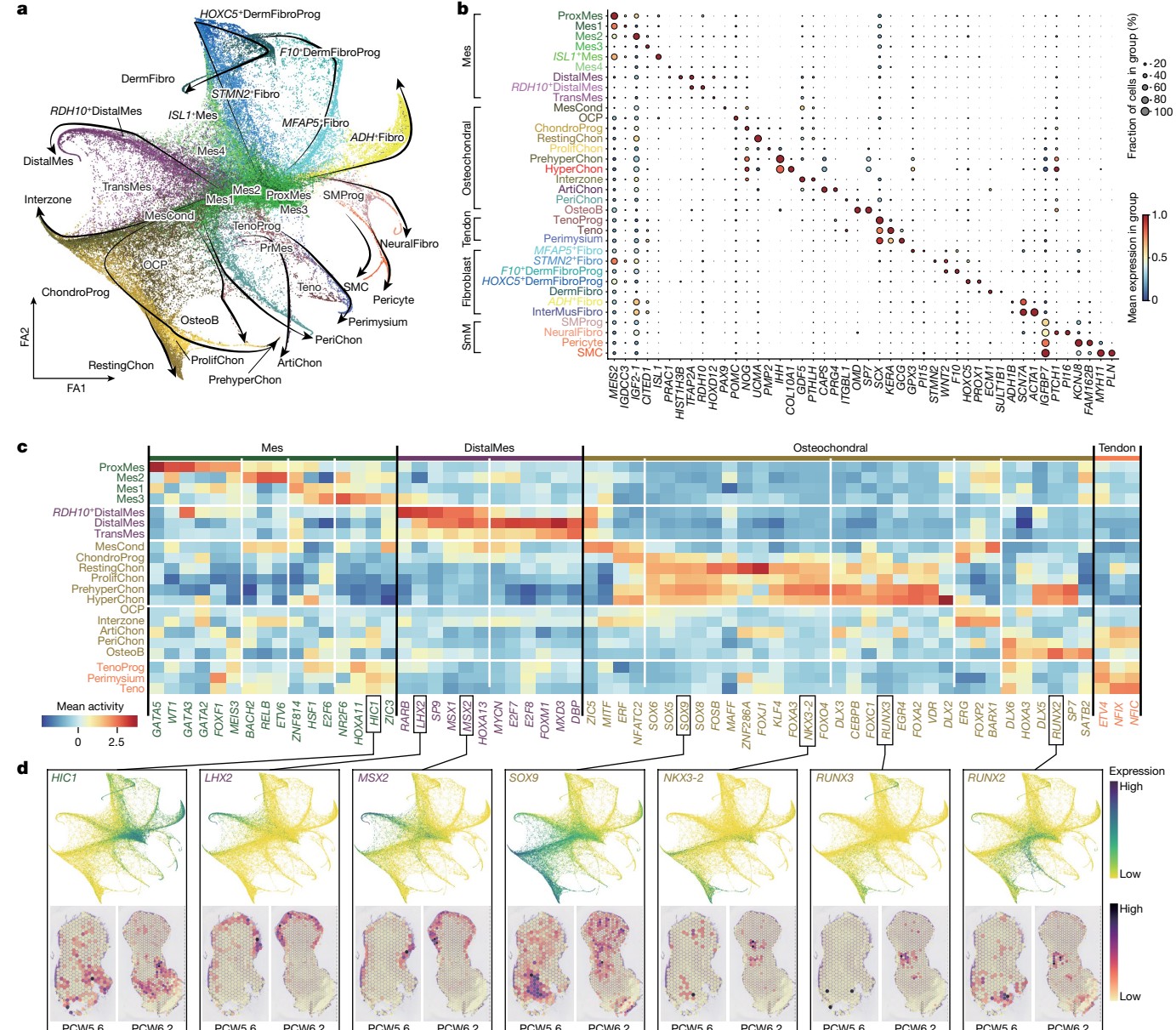

**Fig. 3 | Cell lineage diversification and transcription factor specificity of LPM during human embryonic limb development. a**, Force-directed graph layout of cells associated with the LPM, coloured by cell clusters. The black arrows indicate differentiation directions. Cluster abbreviations are the same as in Fig. 1. FA, force atlas. **b**, Dot plot showing selected marker genes for each cell cluster. The colour bar indicates the average expression level in linearly scaled values. **c**, Heatmap illustrating the vertically normalized mean activity of selected genes encoding transcription factors for each cell cluster. **d**, Force-directed graph (top) and Visium heatmaps (bottom) from the human hindlimb showing the expression of genes encoding transcription factors (normalized and log-transformed).

diffusion maps, partition-based graph abstraction and force-directed graph (see Methods). The global embedding revealed clusters of lineage-committed cells radiating outward from a hub of six mesenchymal states (Fig. 3a,b and Extended Data Fig. 4f). A first mesenchymal population, proximal mesenchymal cells, expressed the regulator of stylopod identity, *MEIS2*, together with *WT1*, which marks the point of limb–torso junction[34] (Fig. 3a,b and Extended Data Fig. 4f,h). Mes1 cells exhibited a similar expression profile but lacked *WT1*, probably representing the mesenchyme just distal to the limb–torso junction (Fig. 3a,b and Extended Data Fig. 4f,h). We also identified a mesenchymal population within the posterior aspect of the developing hindlimb that expressed *ISL1* (*ISL1*+Mes) in addition to *MEIS2*, but not *WT1* (Extended Data Fig. 4f,h). Two further clusters (Mes2 and Mes3) expressed *CITED1*, a gene encoding a molecule that localizes to the proximal domain of the limb and has an unclear role in limb development[35]. Mes2 also expressed *MEIS2*, suggesting a proximal–anterior location (Extended Data Fig. 4f,h). The Mes4 cluster exhibited similar expression patterns to distal and transitional mesenchymal cells but lacked *LHX2* and *IRX1*. These six cell states form the majority of the cells during PCW5 (particularly notable at PCW5.1 (PCW5 plus 1 day) and PCW5.4, at 85% and 65%, respectively; Extended Data Fig. 7c), supporting their early mesenchymal identity. However, their numbers declined thereafter, with almost none present by PCW8.

We next looked for modules of active transcription factor networks associated with progression through each lineage (see Methods; Fig. 3c,d, Supplementary Table 4 and Extended Data Fig. 7a,b). In addition to *WT1*, *MEIS1* and *MEIS2*, *GATA3* (detected in the proximal developing mouse limb) and *GATA5* (a putative proximal–distal patterning gene

in the *Xenopus* limb) were both predicted to be active in the proximal mesenchyme[36,37]. *HOXA11*, which defines the zeugopod, was active in Mes3. The distal mesenchyme showed activation of *LHX2*, *MSX1* and *MSX2*, as previously described in the mouse, as well as *HOXA13*, which defines the autopod[38,39]. *HIC1* was predicted to have activity in several mesenchymal populations (Fig. 3c,d). *HIC1*+ mesenchymal cells are known to migrate into the limb from the hypaxial somite, differentiating into a range of tissues such as chondrocytes and tenocytes, while maintaining *HIC1* expression[40]. Indeed, *HIC1* was active in chondrocyte and tendon populations (Fig. 3c,d).

The chondrocyte lineage increased in number over time, shifting from progenitors to more mature clusters during the period studied (Extended Data Fig. 7c). Mesenchymal condensate cells, *SOX*low*COL2A1*low *PRRX1*hi osteochondral progenitors and *SOX9*hi*COL2A1*low chondrocyte progenitors gave way to three populations of *SOX9*hi*COL2A1*hi chondrocytes: *UCMA*+ resting, *UCMA*−*IHH*− proliferating (with a greater proportion of cells in G2, M and S phases) and *IHH*+ PHCs (Fig. 3b,c, Supplementary Table 1 and Extended Data Fig. 7d–g). In addition, we captured 14 *COL10A1*+*MMP13*+ HCCs (Extended Data Fig. 7d). Curiously, both partition-based graph abstraction and RNA velocity analyses suggested chondrocyte progenitors for an individual sample may progress to either the resting state before proliferation or proceed directly to proliferation (Fig. 3a and Extended Data Fig. 7e,f), although further work is required to investigate this finding. The transition from mesenchymal condensate to committed chondrocytes (with the latter localizing to chondrocyte condensations at PCW5.6 and the developing long bones at PCW6.2) was associated with activity in *SOX5*, *SOX6* and *SOX9*, as well as *MAFF* and *NKX3-2*, which encode chondrogenic transcription factors[41–43] (Fig. 3d). Several regulators of chondrocyte hypertrophy were active in PHCs and HCCs (the latter localizing to the tibial diaphysis at PCW6.2), including *SP7*, *DLX2/3* and *RUNX3* (ref. 41). *RUNX2* was predicted to be active in the perichondrium and osteoblasts, in addition to PHCs and HCCs (Fig. 1d). The osteogenic regulator *SATB2* was highly specific to osteoblasts[44].

To capture cells of the interzone, mesenchymal cells that reside at the sites of future synovial joints and give rise to their constituent parts, we sectioned two forelimbs and two hindlimbs into proximal, middle (containing the knee and elbow regions) and distal segments. These data contained a cluster expressing the interzone marker *GDF5* and articular chondrocytes expressing lubricin (*PRG4*; log fold change = 5.15, $P = 4.1 \times 10^{-26}$) (Fig. 3b and Extended Data Fig. 7d). Notably, *FOXP2* (a negative regulator of endochondral ossification) and *ERG* (a positive regulator of articular chondrocyte) were active in interzone cells but *SOX5*, *SOX6* and *SOX9* were not[45,46] (Fig. 3c).

*SCX*hi*TNMD*low tendon progenitors emerged during PCW5 before being replaced by *TNMD*hi tenocytes and perimysium from PCW7 onwards (Fig. 3b and Extended Data Fig. 7c). *SCX*+*SOX9*+ cells previously shown to give rise to the entheses were also captured, a finding confirmed by spatial transcriptomic and RNA-ISH[47] (Extended Data Fig. 7h–k). Several tenogenic transcription factors were predicted to be active in these clusters, including *ETV4* and *NFIX*[48,49] (Fig. 3c). Finally, fibroblast and smooth muscle cell populations within the limb exhibited clearly distinct transcription factor activities (Extended Data Fig. 7a,b). Dermal fibroblasts showed activity in known regulators of this lineage, including *RUNX1* and *TFAP2C*[50,51], whereas smooth muscle cells and their precursors both showed activity in *GATA6*, which is thought to regulate their synthetic function[52]. In addition, smooth muscle cells showed activity in transcription factors with known roles in smooth muscle function, such as *ARNTL*[53].

## Regulation of embryonic and fetal myogenesis

Limb muscle formation begins with delamination and migration from the somite regulated by *PAX3* and co-regulators such as *LBX1* and *MEOX2*. Two waves of myogenesis follow: embryonic and fetal[3,54].

During embryonic myogenesis, a portion of *PAX3*+ embryonic skeletal muscle progenitors differentiate and fuse into multinucleated myotubes. These primary fibres act as the scaffold for the formation of secondary fibres derived from *PAX7*+ fetal skeletal muscle progenitors, which are themselves derived from *PAX3*+ muscle progenitors[54,55]. To dissect these developmental trajectories in humans, we re-embedded muscle cells using diffusion mapping combined with partition-based graph abstraction and force-directed graph. Three distinct trajectories with an origin in *PAX3*+SkMP emerged (Fig. 4a). The first (labelled first myogenesis, in keeping with embryonic myogenesis) progresses from *PAX3*+SkMP to an embryonic myoblast state (MyoB1) followed by an early embryonic myocyte state (MyoC1), and finally mature embryonic myocytes. The second runs from *PAX3*+SkMP to *PAX3*+*PAX7*+ cells, followed by a heterogeneous pool of mostly *MyoD*low*PAX7*+ SkMP cells (Fig. 4a,b). This represents a developmental path that generates progenitors for subsequent muscle formation and regeneration. The final trajectory (labelled second myogenesis) connects cell states that express *PAX7* first to fetal myoblasts (MyoB2), then to early fetal myocytes (MyoC2), and finally to mature fetal myocytes (Fig. 4a,c and Extended Data Fig. 8b).

Comparing these pathways, *PAX3* is almost absent in fetal myogenesis, whereas it persists to late states along the embryonic pathway, consistent with a previous study that captured *Pax3*+*Myog*+ cells in the mouse limb[56] (Fig. 4b). *ID2* and *ID3*, which have been shown to attenuate myogenic regulatory factors[57,58], were highly expressed in embryonic myogenesis, which may imply different upstream regulatory networks. Additional genes such as *FST*, *RGS4*, *NEFM* and *SAMD11* were also identified to be marking the first myogenic pathway, whereas *TNFSF13B*, *KRT31* and *RGR* mark the second myogenic pathway (Fig. 4b). In fact, keratin genes have been shown to facilitate sarcomere organization[59].

Next, we applied SCENIC to search for transcription factors driving each myogenic stage (Fig. 4d, Supplementary Table 4 and Extended Data Fig. 8a). Although *MYOD1* and *MYOG* showed similar activities across fetal and embryonic myogenesis, several transcription factors were predicted to have higher activity in one or the other. For example, *PITX2* exhibited a higher activity score and abundance during embryonic myogenesis (Fig. 4d), possibly related to its different regulatory roles[60]. By contrast, *PITX1* exhibited comparable activity in both trajectories. Despite its known role as a hindlimb-specific transcription factor, *PITX1* is expressed in both forelimb and hindlimb muscle cells, including *PAX3*+ cells as early as PCW5 (Fig. 4e,f and Extended Data Fig. 8c), suggesting a regulatory role in embryonic myogenesis. Other genes encoding transcription factors specific to embryonic myogenesis included *MSX1*, which maintains the early progenitor pool, the MyoD activator *SIX2* and the satellite cell homeodomain factor *BARX2* (refs. 61–63). Next, we investigated transcriptional repressor expression in muscle. We observed specific expressions of *ID1*, *ID2*, *ID3*, *HEY1*, *MSC* and *HES1* in *PAX7*+ skeletal progenitors (Fig. 4b). The most prominent repressor, *MSC* (also known as musculin, *ABF1* or *MyoR*) encodes a basic helix−loop−helix transcription factor that has been shown to inhibit the ability of MyoD to activate myogenesis in 10T1/2 fibroblasts[64] and rhabdomyosarcoma cells[65]. In C2C12 murine myoblasts, *MSC* facilitates the inhibition of myogenesis by NOTCH (although it appears to exhibit functional redundancy in this role)[66]. To test whether human *MSC* also has a role in repressing *PAX7*+ progenitor maturation, we knocked down *MSC* in primary human embryonic limb myoblasts. Quantitative PCR with reverse transcription (RT−qPCR) results showed upregulation of late myocyte genes, suggesting that it has a key role in maintaining limb muscle progenitor identity (Fig. 4g).

## Spatially resolved microenvironments exhibit distinct patterns of cell−cell communication

We next looked for stage-specific ligand−receptor interactions in co-located cell populations (see Methods). This highlighted the role

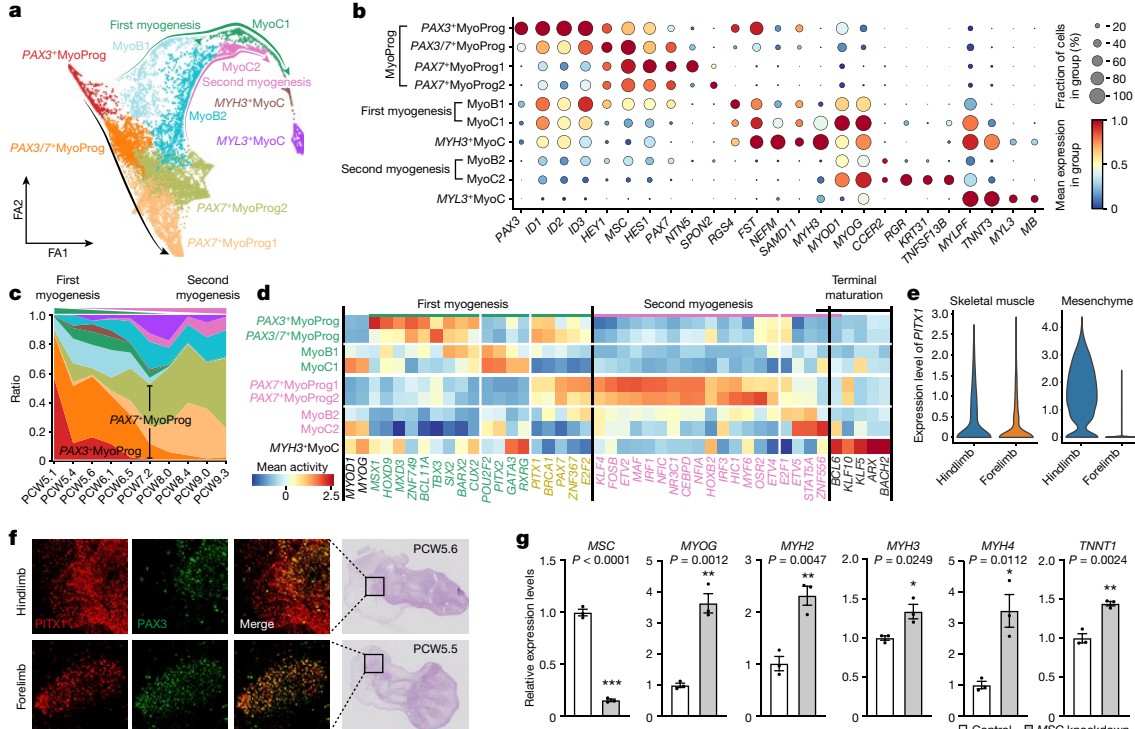

**Fig. 4 | Cell trajectory and transcription factors of embryonic and fetal limb myogenesis. a**, Force-directed graph layout of cells associated with the myogenesis, coloured by cell clusters. The green and pink arrows indicate the direction of first and second myogenesis, separately. MyoB, myoblast; MyoC, myocyte; MyoProg, myogenic progenitor. **b**, Dot plot showing expression pattern of selected marker genes. The colour bar indicates the average expression level in linearly scaled values. **c**, Fraction of cell type per timepoint. **d**, Heatmap illustrating the vertically normalized mean activity of filtered genes encoding transcription factors for each cell cluster. **e**, Violin plot

showing the expression level of *PITX1* in human forelimb and hindlimb at PCW5.6. **f**, Immunofluorescence co-staining (scale bar, 50 μm) of PITX1 and PAX3 on hindlimb (top panels) and forelimb (bottom panels) sections (scale bar, 200 μm). Hindlimb, $n = 4$; forelimb, $n = 2$. **g**, RT–qPCR analysis of the fold-enrichment myocyte genes upon knockdown of *MSC* in human primary embryonic myoblasts. Data are presented as mean ± s.e.m. *P* values are from two-sided Student's *t*-tests. $n = 2$ embryos and 3 independent experiments with similar results.

of the WNT signalling pathway in early limb morphogenesis. *WNT5A* exhibited a proximal–distal expression gradient, peaking in the distal mesenchyme (Fig. 5a–e). Its receptor *FZD10* was expressed in the distal ectoderm of the limb at PCW6, with weak mesenchymal expression, although at this comparatively late stage, it appears to be no longer restricted to posterior regions, as has been reported in early limb development in the mouse[67] (Fig. 5c). Furthermore, our single-cell data revealed high expression of *FZD4* in the mesenchymal condensate, a finding supported by RNA-ISH (Fig. 5a,d,e). This supports the suggestion from in vitro studies that *FZD4* has a role in initiating chondrogenesis when mesenchymal condensate reaches a critical mass[68].

In the early (PCW5.6) limb, NOTCH signalling was predicted to occur in its distal posterior aspect through the *SHH*-induced[11,69] canonical ligand *JAG1* (Fig. 5a,f) and was confirmed with RNA-ISH (Fig. 5g). This interaction occurs between adjacent cells, triggering proteolytic cleavage of the intracellular domain of NOTCH[70]. Through colocalization analysis (see Methods), we observed that *NOTCH1* expression closely follows *JAG1* with a probability of co-existence in each pixel (0.14 × 0.14 μm) of 0.75 ($2.63 × 10^7$ dual-positive pixels out of $3.51 × 10^7$ total pixels containing *JAG1*). Analysis of single-cell data showed that *JAG1* and *NOTCH1* were expressed by several mesenchymal populations within the early limb (Fig. 5a). This finding sheds further light on limb morphogenesis and has implications for conditions in which this signalling axis is disrupted, such as the posterior digit absence of Adams–Oliver syndrome and fifth finger clinodactyly of Alagille syndrome[71,72].

We captured weak but reproducible signals of *FGF8* in the apical ectodermal ridge across various timepoints, whereas *FGF10* was detected in

the adjacent distal mesenchyme (Fig. 5h–i). *FGF8* and *FGF10* have been shown to be expressed in the limb ectoderm and distal mesenchyme, respectively, and to form a feedback loop through *FGFR2* that is essential for limb induction[73]. Ectodermal *FGF8* expression was confirmed via RNA-ISH (Fig. 5j). *FGF10* expression was notably restricted in the foot plate adjacent to the forming phalanges and was excluded in the IDS and *RDH10*+ distal mesenchyme (Fig. 5h,j). This is consistent with expression in mouse, where conditional knockdown leads to short, webbed digits[74]. *FGF10*, which induces chondrogenesis via *FGFR2*, was expressed in the chondrocyte progenitors (Fig. 5h). RNA-ISH confirmed this throughout the skeleton of the forming limb (Fig. 5j). *FGFR2* colocalized with *FGF8* (probability = 0.62; $7.31 × 10^5$ dual positive; $1.17 × 10^6$ total *FGF8*) and similarly, *FGFR2* colocalized with *FGF10* (probability = 0.89; $1.56 × 10^6$ dual positive; $1.75 × 10^6$ total *FGF10*). Finally, the co-expression of *FGF10* and *FGF8* in the same pixel was infrequent (probability = 0.05; $9.68 × 10^4$ dual positive; $1.17 × 10^6$ total *FGF8*). The importance of this receptor in skeletal development is highlighted by the limb phenotypes observed in the *FGFR2*-related craniosynostoses[75].

## Homology and divergence between human and mouse limb development

Limb development has long been studied in model organisms, whereas studies of human samples are few. To explore differences between mice and humans, we collected 13 mouse limb samples for scRNA-seq and combined our newly generated data with 18 high-quality limb datasets from 3 published studies[76–78] (Fig. 6a, Supplementary Table 5 and Extended Data Fig. 9a–c). We used matched orthologues to align

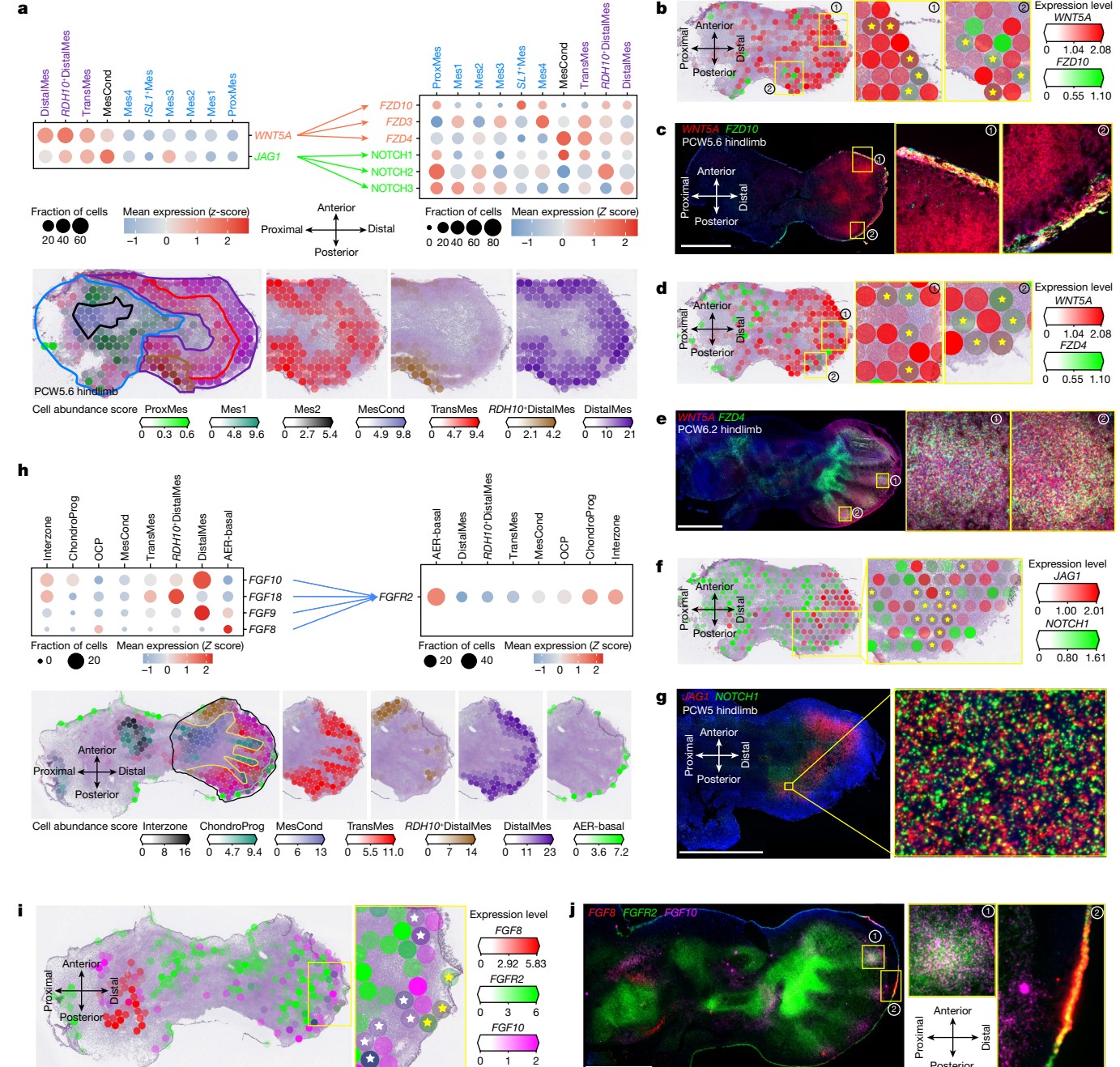

**Fig. 5 | Spatially resolved cell–cell communication. a**, Dot plots showing expression (*Z* score) of ligands and cognate receptors in cell clusters (top), and heatmaps showing predicted cell-type abundance (bottom). **b,d,f**, Visium heatmaps of the hindlimb at PCW5.6 showing expression (normalized and log-transformed) of *WNT5A* (**b,d**), *JAG1* (**f**) and their cognate receptors *FZD10* (**b**), *FZD4* (**d**) and *NOTCH1* (**f**). The yellow stars indicate both the ligand and its receptor expressed. **c,e,g**, RNA-ISH expression of *WNT5A* (**c,e**), *JAG1* (**g**) and their cognate receptors *FZD10* (**c**), *FZD4* (**e**) and *NOTCH1* (**g**) in situ. Scale bars, 1 mm.

**h**, Dot plots of *FGFR2* expression and its ligands (top), and Visium heatmaps of a PCW6.2 human hindlimb showing spatially resolved selected mesenchymal cell cluster (separated by colour) signatures (bottom). **i**, Visium heatmaps of a PCW6.2 human hindlimb showing expression of *FGF8*, *FGF10* and *FGFR2*. The yellow stars indicate that both *FGF8* and *FGFR2* are expressed. The white stars denote that both *FGF10* and *FGFR2* are expressed. **j**, RNA-ISH of *FGF8*, *FGF10* and *FGFR2* expression in the PCW6.2 hindlimb.

the two species and included non-orthologous genes for embedding (see Methods) (Fig. 6b). The resulting integrated atlas showed highly conserved cell composition between humans and mice, with similar developmental trajectories of the skeletal muscle and LPM (Fig. 6c,d and Extended Data Figs. 8d–g, 9d and 10a–c).

Some species differences were most likely technical, such as the greater abundance of *PAX3*⁺ myogenic progenitors in the mouse and the presence of two mouse-enriched mesenchymal populations,

'early DistMes' and 'early ProxMes', which originated from samples before embryonic day 12 (E12), the equivalent stage to human PCW5 (Extended Data Figs. 8d–e,g and 10a–c). Similarly, the lack of *Wt1* expression in the mouse proximal mesenchyme, despite previous description (Extended Data Fig. 10d), is probably due to dissection not including the trunk[34].

However, we also identified biological features. Mouse limbs contained a higher percentage of epithelial and immune cells (Fig. 6d),

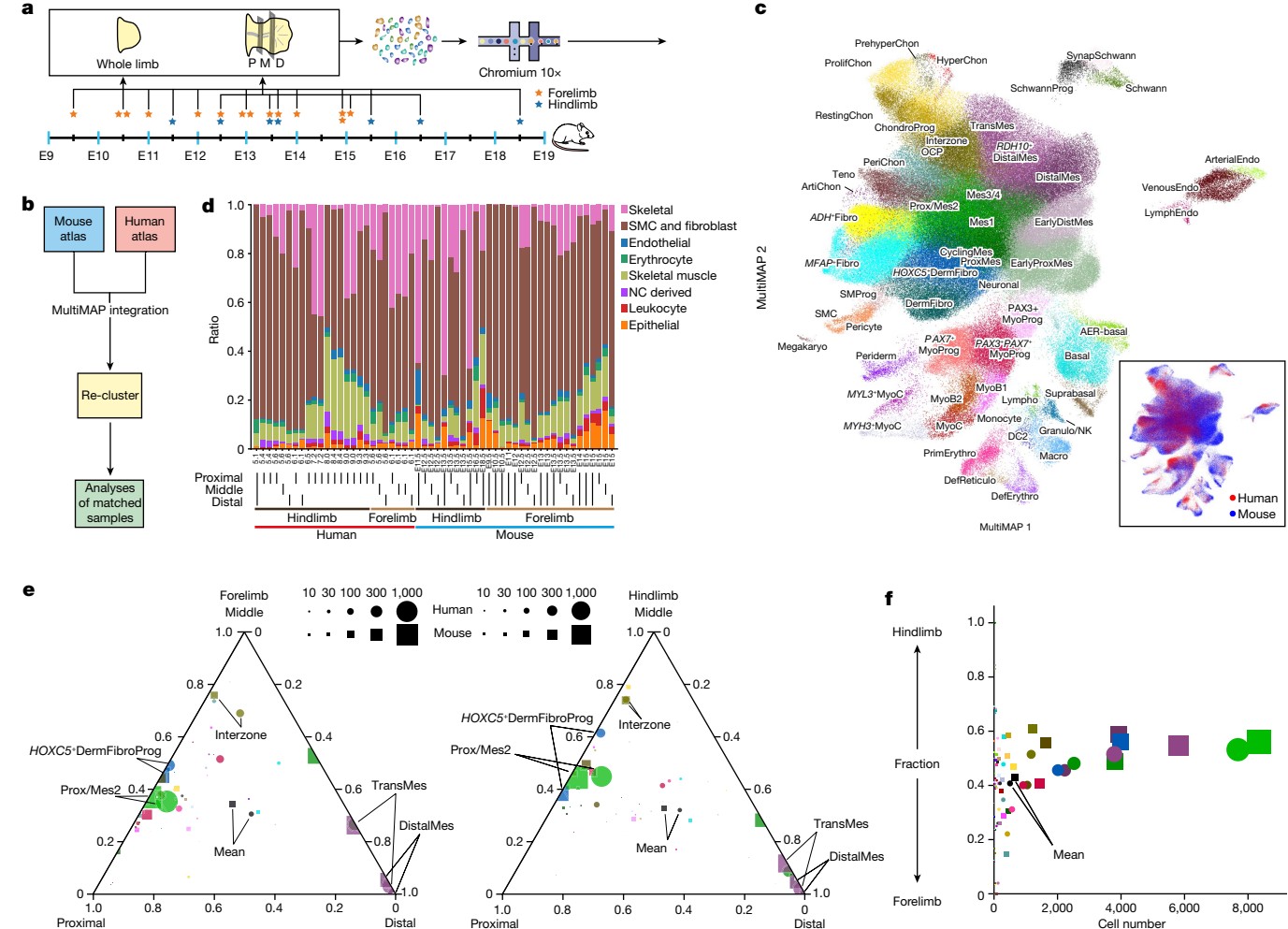

**Fig. 6 | Comparison of a single-cell atlas between human and mouse limb.**
**a**, Overview of mouse sampling and the experimental scheme. The stars indicate timepoints. **b**, Overview of the analysis pipeline to integrate human and mouse scRNA-seq data. **c**, MultiMAP layout of integrated cells, coloured by integrated cell-type annotation or species (bottom right). Cluster abbreviations are the same as in Fig. 1. **d**, Broad cell-type proportions of each scRNA-seq library, with dissection region, location and species labelled at the bottom. NC, neural crest. **e**,**f**, Triangular diagrams (**e**) showing the cell-type proportion biases towards the proximal, middle or distal region of the human and mouse forelimb (left) and hindlimb (right), and a scatter plot (**f**) showing the fraction of hindlimb representation of each cell type. Each cell type or mean is represented by a circle (human) and a square (mouse), with size (square of diameter) denoting the average number of cells per segment (proximal, middle or distal).

possibly due to faster maturation of these systems in the mouse. In skeletal muscle, *PAX3⁺PAX7⁺* myogenic progenitor cells were more abundant in humans than in mice (Extended Data Fig. 8d–f). Although the *PAX3⁺* pools were transcriptomically similar, the mouse data showed low expression of promyogenic factors *Fst* and *Uchl1*, although this may again be due to differences in sample stage[79,80] (Extended Data Fig. 8h). The mesenchymal compartments of both species were highly analogous (Extended Data Fig. 10d). Notably, in both species *FGF8* was expressed in the proximal mesenchyme; its presence in mesenchyme has been reported only in urodeles[81] (Extended Data Figs. 2c–e and 10e,f).

To explore the proximal–distal axis and forelimb versus hindlimb identities across species, we dissected forelimbs and hindlimbs from each species into proximal, middle and distal segments (Figs. 1a and 6a). Overall cluster compositions along the proximal–distal axis were highly similar, with proximal and distal/transitional mesenchymal cells enriched in the proximal and distal samples, respectively, and interzone enriched in the middle sections (Fig. 6e). Forelimb and hindlimb composition was also highly similar in both species (Fig. 6f). Finally, we took cells from the 34 LPM-derived states and compared orthologue expression signatures between proximal versus distal segments and forelimb versus hindlimb samples in mice and humans. Both species recapitulated known proximal–distal-biased genes such as *MEIS2* (proximal), *LHX2* (distal) and HOX genes (Extended Data Fig. 9e). Known forelimb/hindlimb-biased genes were also captured, such as *TBX5* specific to the forelimb and *TBX4*, *PITX1* and *ISL1* specific to the hindlimb (Extended Data Fig. 9f). Overall, we showed that the spatial expression patterns of genes controlling forelimb/hindlimb and proximal–distal identity are highly similar between mice and humans.

## Discussion

Our developmental limb atlas combines single-cell and spatial transcriptomics to form the first detailed characterization of human limb development across space and time, identifying and placing 67 clusters of cells into anatomical context[76]. In doing so, we uncovered several new cell states. We identified neural fibroblasts surrounding the sciatic nerve and two clusters in the tendon lineage mapped to the tendon and perimysium. In the autopod, we described three clusters of cells with subtly different transcriptomes mapped to distinct regions: distal (*LHX2⁺MSX1⁺SP9⁺*), *RDH10⁺* distal (*RDH10⁺LHX2⁺MSX1⁺*) and transitional

(*IRX1*+*MSX1*+) mesenchyme. Through spatial transcriptomic analysis of the foot plate, we connected physiological gene expression patterns to genetic conditions with a digit phenotype, demonstrating the clinical relevance of developmental cell atlas projects. Our study also presents a refined model of the overlapping processes of primary and secondary myogenesis in the limb, while identifying and validating *MSC* as a key player in muscle stem cell maintenance. We further maximized the utility of this study by presenting an integrated cross-species atlas with unified annotations. It is important to acknowledge that our findings are limited by the lack of earliest-stage limb samples, the 3′ end bias of our transcriptomic assays, differences in cell-type coverages between scRNA-seq and spatial transcriptomics, as well as the low spatial resolution of spatial transcriptomics (see Supplementary Information). In summary, our work presents one of the first cell atlases of an entire human embryonic tissue, presents detailed spatiotemporal models of its development and provides rich resources for single-cell, spatial and developmental biology communities.

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

## Methods

### Human tissue sample collection

First trimester human embryonic tissue was collected from elective termination of pregnancy procedures at Addenbrookes Hospital (Cambridge, UK) under full ethical approval from the East of England–Cambridge Central Research Ethics Committee (REC-96/085; for scRNA-seq and Visium), and at Guangzhou Women and Children's Medical Center (China) under approval of the Research Ethics Committee of Zhongshan School of Medicine (ZSSOM), Sun Yat-sen University (ZSSOM-2019-075) and Guangzhou Women and Children's Medical Center (2022-050A01, for in situ hybridization and immunohistochemistry). Consent was obtained after the decision was made to terminate pregnancy, in advance of the procedure. Experiments also followed the 2021 International Society for Stem Cell Research (ISSCR) guidelines in working on human embryos. Informed written consent was obtained from all donors before termination of pregnancy and tissue collection. No developmental abnormalities were visible or known in any of the embryos collected. All human data generated from China were registered at the China National Center for Bioinformation (PRJCA012474) and have been approved by the Chinese Ministry of Science and Technology for the Review and the Approval of Human Genetic Resources (2023BAT0445). For light-sheet fluorescence microscopy, tissues were obtained through INSERM's HuDeCA Biobank and made available in accordance with the French bylaw. Permission to use human tissues was obtained from the French agency for biomedical research (Agence de la Biomédecine, Saint-Denis La Plaine, France; no. PFS19-012) and the INSERM Ethics Committee (IRB00003888). Written, informed consent was given for tissue collection by all patients. Embryonic age (PCW) was estimated using the independent measurement of the crown rump length, using the formula PCW (days) = 0.9022 × crown rump length (mm) + 27.372. PCW was recorded as week and day, separated by a decimal point; for example, PCW5.6 translates to 5 weeks and 6 days.

### Human tissue processing and scRNA-seq data generation

Embryonic limbs were dissected from the trunk under a microscope using sterile microsurgical instruments. To capture cells of the interzone, four samples (a hindlimb and a forelimb from both PCW5.6 and PCW6.1) were then further dissected into proximal, middle (containing undisturbed interzone) and distal thirds before dissociation. For the PCW5.1 sample, no further dissection was performed due to small size and the limb was dissociated as a whole. For all other samples, the limb was dissected into proximal and distal halves before dissociation.

Dissected tissues were mechanically chopped into a mash, and then were digested in Liberase TH solution (50 μg ml⁻¹; 05401135001, Roche) at 37 °C for 30–40 min till no tissue piece was visible. Digested tissues were filtered through 40-μm cell strainers followed by centrifugation at 750g for 5 min at 4 °C. Cell pellets were resuspended with 2% FBS in PBS if the embryos were younger than PCW8, otherwise red blood cell lysis (00-4300, eBioscience) was performed. The single-cell suspensions derived from each sample were then loaded onto separate channels of a Chromium 10x Genomics single-cell 3′ version 2 library chip as per the manufacturer's protocol (PN-120233, 10x Genomics). cDNA sequencing libraries were prepared as per the manufacturer's protocol and sequenced using an Illumina Hi-seq 4000 with 2 x 150-bp paired-end reads.

### Mouse tissue sample collection and scRNA-seq data generation

Timed pregnant C57BL/6J wild-type mice were ordered from Jackson Laboratories. On arrival, timed pregnant mice were housed singly and maintained in solid-bottom Zyfone individually ventilated microisolator caging ((Lab Products) 7″ wide × 12″ length × 6″ height). All cages were sanitized in a cagewash facility with a final rinse temperature of at least 180 F° before use. Each cage contained autoclaved hardwood chip bedding (Aspen Chip Bedding, Northeastern Products)

and two sheets of tissue paper for nest building enrichment. All mice were fed irradiated standard rodent diet (PicoLab Rodent Diet 5053, PMI Nutrition International), and provided with ad libitum reverse osmosis water via water pouches (Hydropac, Lab Products) on arrival, before the start of any experimental manipulation. Animal rooms were maintained on a 14:10 h light–dark cycle with an hour-long dawn–dusk period with humidity ranging from 30% to 70% and temperatures ranging from 71 °F to 75 °F in compliance with the Guide for the Care and Use of Laboratory Animals. Animals were checked daily by the animal care staff to check for health and the availability of food, water and cage conditions. Embryos were collected at E12.5, E13.5 and E16.5. Only right-side forelimbs and hindlimbs were used in this study: $n = 5$ at the E12.5 timepoint, $n = 5$ at E13.5 and $n = 2$ at E16.5. No randomization, blinding or sample size choice were done. The sex of the embryos was not known or selected. Hindlimbs and forelimbs were pooled separately in ice-cold HBSS (14175-095, Gibco), and dissected into proximal, middle and distal limb regions, which were again separately pooled in 200 μl of HBSS placed in a drop in the centre of a 6-cm culture plate. Tissues were then minced with a razor blade and incubated with an addition of 120 μl of diluted DNase solution (04716728001, Roche) at 37 °C for 15 min. The DNase solution consisted of 1 ml UltraPure water (10977-015, Invitrogen), 110 μl 10× DNase buffer and 70 μl DNase stock solution. Of diluted Liberase TH (05401151001, Roche), 2 ml was then added to the plate, and the minced tissue suspension was pipetted into a 15-ml conical centrifuge tube. The culture plate was rinsed with 2 ml, and again with 1 ml of fresh Liberase TH, which was serially collected and added to the cell suspension. The suspension was incubated at 37 °C for 15 min, triturated with a P1000 tip and incubated for an additional 15 min at 37 °C. For the Liberase TH solution, 50X stock was prepared by adding 2 ml PBS to 5 mg of Liberase TH. Working solution was made by adding 100 μl 50X stock to 4.9 ml PBS. After a final gentle trituration of the tissue with a P1000 tip, the suspension was spun at 380g in a swinging bucket rotor at 4 °C for 5 min. After removing the supernatant, cells were resuspended in 5 ml of 2% FBS in PBS, and filtered through a pre-wetted 40-μm filter (352340, Falcon). After spinning again at 380g at 4 °C for 5 min, the supernatant was removed and cells were resuspended in 200 μl 2% FBS in PBS. A small aliquot was diluted 1:10 in 2% FBS/PBS and mixed with an equal volume of Trypan Blue for counting on a haemocytometer. The full suspension was diluted to 1.2 million cells per millilitre for processing on the 10x Genomics Chromium Controller, with a target of 8,000 cells per library. Libraries were processed according to the manufacturer's protocol, using the v3 Chromium reagents. All animal procedures were performed according to protocols approved by the Institutional Animal Care and Use Committee at the California Institute of Technology. Animals were housed in an AAALAC-accredited facility in accordance with the Guide for the Care and Use of Laboratory Animals.

### Visium spatial transcriptomic experiments of human tissue

Whole embryonic limb samples at PCW6–PCW8 were embedded in OCT within cryo wells and flash-frozen using an isopentane and dry ice slurry. Ten-micron-thick cryosections were then cut in the desired plane and transferred onto Visium slides before haematoxylin and eosin staining and imaged at ×20 magnification on a Hamamatsu Nanozoomer 2.0 HT Brightfield. These slides were then further processed according to the 10x Genomics Visium protocol, using a permeabilization time of 18 min for the PCW6 samples and 24 min for older samples. Images were exported as tiled tiffs for analysis. Dual-indexed libraries were prepared as in the 10x Genomics protocol, pooled at 2.25 nM and sequenced four samples per Illumina Novaseq SP flow cell with read lengths of 28 bp R1, 10 bp i7 index, 10 bp i5 index and 90 bp R2.

### Digit region analysis of Visium data

The Visium data were clustered by the Louvain algorithm after filtering genes that were expressed in less than one spot and performing

normalization and logarithmization. After that, the spot clusters of interest were annotated based on haematoxylin and eosin histology and marker genes. The differential expression testing was performed by Wilcoxon test using Scanpy (sc.tl.rank_gene_group).

## Alignment, quantification and quality control of scRNA-seq data

Droplet-based (10x) sequencing data were aligned and quantified using the Cell Ranger Single-Cell Software Suite (v3.0.2, 10x Genomics). The human reference is the hg38 genome refdata-cellranger-GRCh38-3.0.0, available at: http://cf.10xgenomics.com/supp/cell-exp/refdata-cellranger-GRCh38-3.0.0.tar.gz. The mouse reference is the mm10 reference genome refdata-gex-mm10-2020-A, available at: https://cf.10xgenomics.com/supp/cell-exp/refdata-gex-mm10-2020-A.tar.gz. Published mouse scRNA-seq FASTQ files were downloaded from ENCODE's portal and the Gene Expression Omnibus[76–78]. The following quality control steps were performed: (1) cells that expressed fewer than 200 genes (low quality) were excluded; (2) genes expressed by less than five cells were removed; and (3) cells in which over 10% of unique molecular identifiers were derived from the mitochondrial genome were removed.

## Alignment and quantification of human Visium data

Raw FASTQ files and histology images were processed, aligned and quantified by sample using the Space Ranger software v.1.1.0, which uses STAR v.2.5.1b52 for genome alignment, against the Cell Ranger hg38 reference genome refdata-cellranger-GRCh38-3.0.0, available at: http://cf.10xgenomics.com/supp/cell-exp/refdata-cellranger-GRCh38-3.0.0.tar.gz.

## Doublet detection of scRNA-seq data

Doublets were detected with an approach adapted from a previous study[82]. In the first step of the process, each 10x lane was processed independently using the Scrublet to obtain per-cell doublet scores. In the second step of the process, the standard Scanpy processing pipeline was performed up to the clustering stage, using default parameters[83]. Each cluster was subsequently separately clustered again, yielding an over-clustered manifold, and each of the resulting clusters had its Scrublet scores replaced by the median of the observed values. The resulting scores were assessed for statistical significance, with $P$ values computed using a right tailed test from a normal distribution centred on the score median and a median absolute deviation-derived standard deviation estimate. The median absolute deviation was computed from above-median values to circumvent zero truncation. The $P$ values were corrected for false discovery rate with the Benjamini–Hochberg procedure and were used to assess doublet level. The clusters from batch-corrected overall clustering across all the samples that have median scores lower than 0.1 and are supported by an absence of exclusive marker genes or literature were manually curated and removed (1,450 doublets were removed in human data and 958 in mouse data).

## Data preprocessing and integration of scRNA-seq data

Preprocessing included data normalization (pp.normalize_per_cell with 10,000 counts per cell after normalization), logarithmization (pp.log1p), highly variable genes detection (pp.highly_variable_genes and select for highly correlated ones as previously described[76]) per batch and merging, data feature scaling (pp.scale), cell cycle and technical variance regressing (tl.score_gene_cell_cycle and pp.regress_out(adata,['S_score', 'G2M_score', 'n_counts', 'percent_mito'])), and principal component analysis (tl.pca with 100 components) performed using the Python package Scanpy (v.1.8.2). bbknn (v.1.5.1) was used to correct for batch effect between sample identities with the following parameters (n_pcs = 100, metric = 'Euclidean', neighbors_within_batch = 3, trim = 299, approx = false). Following this, further dimension reduction was performed using uniform manifold approximation and projection (UMAP) (scanpy tl.umap with default parameters) based on the corrected neighbourhood graph of bbknn.

## Clustering and annotation of scRNA-seq data

We first applied Leiden graph-based clustering (scanpy tl.leiden with default parameters) to perform unsupervised cell classification. Each cluster was then subclustered if heterogeneity was still observed and was manually annotated (see Supplementary Table 1 for marker genes) and curated as previously described[84]. To make sure all the curated Leiden clusters could clearly be mapped onto their UMAP embedding coordinates, we performed the partition-based graph abstraction (PAGA) (tl.paga with the Leiden clusters) and reran UMAP with the initial position from the PAGA.

## Deconvolution of human Visium data using cell2location

To map clusters of cells identified by scRNA-seq in the profiled spatial transcriptomics slides, we used the cell2location method[85]. In brief, this involved first training a negative binomial regression model to estimate reference transcriptomic profiles for all the scRNA-seq clusters in the developing limb. Next, lowly expressed genes were excluded as per recommendations for use of cell2location, leaving 13,763 genes for downstream analysis. Next, we estimated the abundance of each cluster in the spatial transcriptomics slides using the reference transcriptomic profiles of different clusters. This was applied to all slides simultaneously, using the sample ID as the batch_key and categorical_covariate_keys. To identify microenvironments of colocalizing cell clusters, we used non-negative matrix factorization implementation in scikit-learn, utilizing the wrapper in the cell2location package[86]. A cell type was considered part of a microenvironment if the fraction of that cell type in said environment was over 0.2.

## Alignment and merging of multiple Visium sections using VisiumStitcher

To analyse the whole PCW8.1 human hindlimb, we took three consecutive 10-µm sections from different regions and placed them on different capture areas of the same Visium library preparation slide. The first section spanned the distal femur, knee joint and proximal tibia (sample C42A1), the second the proximal thigh (sample C42B1) and the third the distal tibia, ankle and foot (sample C42C1).

The images from these three Visium capture areas were then aligned using the TrackEM plugin (Fiji)[87]. Following affine transformations of C42B1 and C42C1 to C42A1, the transformation matrices were exported to an in-house pipeline (https://github.com/Teichlab/limbcellatlas) for complementary alignment of the spot positions from the SpaceRanger output to the reconstructed space. In addition, we arbitrarily decided that overlapping regions would keep the image from the centre portion (see Extended Data Fig. 6a) while keeping all the spots in the data matrix. Next, we merged the three library files and matched the reconstructed image to the unified AnnData object.

## Trajectory analysis of human scRNA-seq data

Development trajectories were inferred by combining diffusion maps, PAGA and force-directed graph. The first step of this process was to perform the first nonlinear dimensionality reduction using diffusion maps (scanpy tl.diffmap with 15 components) and recompute the neighbourhood graph (scanpy pp.neighbors) based on the 15 components of diffusion maps. In the second step of this process, PAGA (scanpy tl.paga) was performed to generate an abstracted graph of partitions. Finally, force-directed graph was performed with the initial position from PAGA (scanpt tl.draw_graph) to visualize the development trajectories.

## RNA velocity calculations for mesenchymal compartment

The scVelo version 0.24 package for Python was used to calculate a ratio of spliced-to-unspliced mRNA abundances in the dataset[88]. The data were subclustered to the mesenchymal compartment for a

single sample (PCW7.2). The data were then processed using default parameters following preprocessing as described in Scanpy scVelo implementation. The samples were preprocessed using functions for detection of minimum number of counts, filtering and normalization using scv.pp.filter_and_normalize and followed by scv.pp.moments function using default parameters. The gene-specific velocities were then calculated using scv.tl.velocity with mode set to stochastic and scv.tl.velocity_graph functions, and visualized using scv.pl.velocity_graph function.

### Cell–cell communication analysis of human scRNA-seq data
Cell–cell communication analysis was performed using CellPhoneDB. org (v.2.1.4) for each dataset at the same stage of development[89,90]. The stage-matched Visium data were used to validate the spatial distance and expression pattern of significant ($P < 0.05$) ligand–receptor interactions.

### Regulon analysis of transcription factors
To carry out transcription factor network inference, analysis was performed as previously described[91] using the pySCENIC Python package (v.0.10.3). For the input data, we filtered out the genes that were expressed in less than 10% of the cells in each cell cluster. Then, we performed the standard procedure including deriving co-expression modules (pyscenic grn), finding enriched motifs (pyscenic ctx) and quantifying activity (pyscenic aucell).

### Integration of human and mouse scRNA-seq data
Mouse orthologues were first 'translated' to human genes using MGI homology database (https://www.informatics.jax.org/homology.shtml). Processed human and mouse data were then merged together using outer join of all the genes. The matched dataset was then integrated by MultiMAP[92] (the MultiMAP_Integration() function), using separately pre-calculated principal components and the union set of previously calculated mouse and human feature genes (including both orthologues and non-orthologues) to maximize biological variance. Downstream clustering and embedding were performed in the same way as previously described and cell-type annotation was based on marker genes. Cell-type composition of proximal, middle and distal segments of the same limb was visualized using plotly.express. scatter_ternary() function. To capture the differential expression of sparsely captured genes, the odds ratio of the percentages of non-zero cells between groups of cells was used to select for proximal/distal or forelimb/hindlimb biased genes with a cut-off at 30-fold and 3-fold, respectively.

### Immunohistochemistry
The limb samples were post-fixed in 4% paraformaldehyde for 24 h at 4 °C followed by paraffin embedding. A thickness of 4-μm sections were boiled in 0.01 M citrate buffer (pH 6.0) after dewaxing. Immunofluorescence staining was then carried out as previously described[93]. Primary antibodies for RUNX2 (1:50; sc-390715, Santa Cruz), THBS2 (1:100; PA5-76418, Thermo Fisher), COL2A1 (1:200; sc-52658, Santa Cruz), PITX1 (1:30; Ab244308, Abcam), PAX3 (1:1; AB_528426 supernatant, DSHB), ALDH1A3 (1:50; 25167-1-AP, Proteintech) and MYH3 (1:3; AB_528358 supernatant, DSHB) and anti-KERA (1:1,000; HPA039321, Sigma-Aldrich) were incubated overnight at 4 °C. After washing, sections were incubated with appropriate secondary antibodies Alexa Flour 488 goat anti-mouse IgG1 (1:400; A-21121, Invitrogen), Alexa Flour 647 goat anti-mouse IgG2b (1:400; A-21242, Invitrogen), Alexa Flour 488 goat anti-mouse IgG (H + L) (1:400; A-11029, Invitrogen) and Alexa Flour 546 goat anti-rabbit IgG (H + L) (1:400; A-11035, Invitrogen) at room temperature for 1 h, and were mounted using FluorSave Reagent (345789, Calbiochem). For 3,3-diaminobenzidine staining, we used a streptavidin–peroxidase broad spectrum kit (SP-0022, Bioss) and 3,3-diaminobenzidine solution (ZLI-9017, ZSGB-BIO)

following the manuals from the manufacturers. The primary antibodies PI16 (1:500; HPA043763, Sigma-Aldrich), FGF19 (1:500; DF2651, Affinity) and NEFH (1:1,000; 2836, Cell Signaling) were applied. Single-plane images were acquired using an inverted microscope (DMi8, Leica).

### RNA-ISH
Fresh tissue samples were embedded in OCT and frozen at −80 °C until analysis. Cryosections were cut at a thickness of 10 μm or 12 μm using a cryostat (Leica CM1950 or CM3050). Before staining, tissue sections were post-fixed in 4% paraformaldehyde for 15 min at 4 °C. After a series of 50%, 70%, 100% and 100% ethanol dehydration for 5 min each, tissue sections were treated with hydrogen peroxide for 10 min. Next, the sections were digested with protease IV (322336, ACD) for 20–30 min at room temperature; alternatively, they were digested with protease III (322337, ACD) for 15 min after heat-induced epitope retrieval. RNA-ISH was then carried out manually or using BOND RX (Leica) by using the RNAscope Multiplex Fluorescent Reagent Kit v2 Assay (323110, ACD) or the PinpoRNA multiplex fluorescent RNA in situ hybridization kit (PIF2000, GD Pinpoease) according to the instructions by the manufacturers. To visualize targeted RNAs from individual channels, different tyramide signal amplification (TSA) fluorescent substrates were incubated. Two sets of fluorophores TSA520, TSA570 and TSA650 (PANOVUE) and Opal 520, Opal 570 and Opal 650 (Akoya Biosciences) were used and consistent results were obtained. For the staining of four probes, the RNAscope 4-plex Ancillary Kit (323120, ACD) was applied additionally, and a combination of fluorophores TSA520, TSA570, Opal620 and Opal690 were used. The stained sections were imaged with either AxioScan.Z1 (Zeiss) or the Opera Phenix High-Content Screening System (PerkinElmer).

### RNA-ISH colocalization analysis
Colocalization analysis was performed by first identifying the expressed genes on raw images through the utilization of a pixel classifier trained with the software ilastik[94]. Subsequently, the predicted mask image was subjected to analysis, with the probability of co-occurrence determined by tallying the instances in which one gene coexists with another at the same $0.14 \times 0.14$ μm pixel, and dividing this by the total number of pixels in which the gene of interest was expressed, regardless of the presence of the other gene.

### Light-sheet fluorescence microscopy
Embryonic and fetal limbs were dissected from morphologically normal specimens collected from PCW5 to PCW6.5. Candidate antibodies were screened by immunofluorescence on cryosections obtained from OCT-embedded specimens as previously described[95]. Whole-mount immunostaining of the limbs was performed as previously described, with primary antibody incubation at 37 °C reduced to 3 days followed by 1 day in secondary antibodies. Samples were embedded in 1.5% agarose and optically cleared with solvents using the iDisco+ method. Cleared samples were imaged with a Blaze light-sheet microscope (Miltenyi Biotec) equipped with a 5.5MP sCMOS camera controlled by Imspector Pro 7.5.3 acquisition software. A ×12 objective with ×0.6 or ×1 magnification (MI plan NA 0.53) was used. Imaris (v10.0, BitPlane) was used for image conversion, processing and video production.

### The antibodies used for light-sheet fluorescence microscopy
IRX1 Sigma-Aldrich cat. no. HPA043160, RRID: AB_10794771 (1/200e); MSX1 R&D Systems cat. no. AF5045, RRID: AB_2148804 (1/500e); LHX2 Abcam cat. no. ab184337, RRID: AB_2916270 (1/1,000e); SOX9 Abcam cat. no. ab196184, RRID: AB_2813853 (1/500e); MAFB Abcam cat. no. ab223744, RRID: AB_2894837 (1/500e); donkey anti-rabbit IgG H&L (Alexa Fluor 555) Abcam cat. no. ab150062, RRID: AB_2801638 (1/800e); and donkey anti-goat IgG H&L (Alexa Fluor 750) Abcam cat. no. ab175745, RRID: AB_2924800 (1/300e).

## MSC knockdown in human primary myoblasts

**Isolation of human primary myoblast cells.** The thighs from human embryos were processed as previously described[96], except that the dissociated cells were not treated with erythrocyte lysis solution, and were incubated with anti-human CD31 (12-0319-41, eBioscience), CD45 (12-0459-41, eBioscience) and CD184 (17-9999-41, eBioscience) antibodies for cell sorting. Fluorescent activated cell sorting (BD, influx) sorted CD31⁻CD45⁻CD184⁺ cells were cultured in complete growth medium DMEM supplemented with 20% FCS and 1% penicillin–streptomycin (15140122, Gibco).

**Small interfering RNA transfection.** Human primary myoblasts were seeded into a six-well plate one night before transfection. When the cell density reached approximately 50% confluence, oligos of small interfering RNA against *MSC* and negative control were transfected using Lipofectamine 3000 reagent (L3000015, Invitrogen) at a final concentration of 37.5 nM. After incubation for 16 h, the growth medium was replaced with differentiation medium containing 2% horse serum and 1% penicillin–streptomycin in DMEM. After culturing for an additional 6–8 h, the cells were collected for RNA extraction. Initially, three siRNA oligos (9242-1, 9242-2 and 9242-3, Bioneer) were tested, and the third one with sense sequences 5′-GAAGUUUCCGCAGCCAACA-3′ were used in this study.

**RNA extraction and qPCR.** Total cell RNA was extracted with the EZ-press RNA purification kit (B0004D, EZBioscience), and the cDNA was synthetized using the PrimeScript RT Master Mix Kit (RR036A, Takara). The qPCR was performed using PerfectStart Green qPCR Super Mix (AQ601, TransGen Biotech) on a real-time PCR detection system (LightCycle480 II, Roche). *RPLPO* served as an internal control, and the fold enrichment was calculated using the formula $2^{-\Delta\Delta Ct}$. The following primers (5′–3′) were used:

*RPLPO* forward: ATGCAGCAGATCCGCATGT, reverse: TTGCG CATCATGGTGTTCTT; *MSC* forward: CAGGAGGACCGCTATGAGAA, reverse: GCGGTGGTTCCACATAGTCT; *MYOG* forward: AGTGCCATC CAGTACATCGAGC, reverse: AGGCGCTGTGAGAGCTGCATTC; *MYH2* forward: GGAGGACAAAGTCAACACCCTG, reverse: GCCCTTTCTAG GTCCATGCGAA; *MYH3* forward: CTGGAGGATGAATGCTCAGAGC, reverse: CCCAGAGAGTTCCTCAGTAAGG; *MYH4* forward: CGGGAG GTTCACACAAAAGTCATA, reverse: CCTTGATATACAGGACAGTGACAA; *TNNT1* forward: AACGCGAACGTCAGGCTAAGCT, reverse: CTTGAC CAGGTAGCCGCCAAAA.

## Ethics statement

The work done in the UK was supported by the National Institute for Health and Care Research Cambridge Biomedical Research Centre (NIHR203312) and provided by the Cambridge Biorepository for Translational Medicine (https://www.cbtm.group.cam.ac.uk). Human fetal samples were provided by the National Institute for Health and Care Research Cambridge Biomedical Research Centre and collected under the Research Ethics Committee-approved study 96/085. The work done in China was approved by the Research Ethics Committee of Zhongshan School of Medicine (ZSSOM), Sun Yat-sen University (ZSSOM-2019-075) and Guangzhou Women and Children's Medical Center (2022-050A01). At both centres, consent was obtained from the patient following the decision to terminate the pregnancy and in advance of the procedure. All animal procedures were performed according to protocols approved by the Institutional Animal Care and Use Committee at the California Institute of Technology. Details of human tissue sample collection are in the Methods section of this article. The views expressed are those of the authors and not necessarily those of the National Institute for Health and Care Research or the Department of Health and Social Care.

## Reporting summary

Further information on research design is available in the Nature Portfolio Reporting Summary linked to this article.

## Data availability

All of our newly generated raw data are publicly available on ArrayExpress (mouse scRNA-seq, E-MTAB-10514; human Visium, E-MTAB-10367; and human scRNA-seq, E-MTAB-8813). Previously published raw data can be found from the ENCODE portal (ENCSR713GIS) and the Gene Expression Omnibus (GSE137335 and GSE142425). Processed data can be downloaded and visualized at our data portal (https://limb-dev. cellgeni.sanger.ac.uk/). The data deposited and made public are compliant with the regulations of Ministry of Science and Technology of the People's Republic of China. Source data are provided with this paper.

## Code availability

All in-house code can be found on GitHub (https://github.com/Teichlab/limbcellatlas/), which is central to our conclusions.

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

**Acknowledgements** We thank M. Thomson's laboratory for help with mouse 10x loading; members of the Teichmann laboratory, Zhang laboratory, Marioni laboraory, Haniffa laboratory and Behjati laboratory for discussion and feedback; O. Pourquié for feedback on muscle development; A. Pyle, H. Xi and J. Langerman for sharing their raw data with us; and K. To for proofreading the manuscript. This work was supported by the Wellcome Trust grants 206194, 108413/A/15/D and 211276/Z/18/Z, as well as the National Key Research and Development Program (grants 2019YFA0801703 and 2022YFA1104904), the National Natural Science Foundation of China (grants 31871370 and 82203952), the Science and Technology Program of Guangzhou (grant 202002030429) (to H.Z.); and the China Postdoctoral Science Foundation (grant 2021M700936) and the Natural Science Foundation of Guangdong (grant 2019A1515011342) (to S.W.). P.H. holds a non-stipendiary research fellowship at St Edmund's College, University of Cambridge. J.E.G.L. is funded by the Wellcome Trust under the clinical PhD programme (grant 222902/Z/21/Z) and supported by Darwin College, Cambridge through a Geoffrey Fisk Studentship. A.C. is funded by the INSERM cross-cutting programme HuDeCA 2018. R.B. was a recipient of a fellowship from the 'Fondation pour la Recherche Médicale' (FRM). This publication is part of the Human Cell Atlas (www.humancellatlas.org/publications/#).

**Author contributions** S.A.T. and H.Z. supervised the project. S.A.T. initiated and designed the project. X.H. and Y.F. carried out human tissue collection. B.A.W. carried out mouse tissue collection. B.A.W., L.M., L.B., R.E. and E.S.F. performed scRNA-seq. E.T. performed the Visium spatial experiments. S.W., K.R. and E.T. carried out the in situ staining and functional experiments. R.B. performed the light-sheet fluorescence microscopy. H.Y., C.L. and H.Z. provided experimental support. B.Z., P.H. and J.E.G.L. analysed the sequencing data and generated figures. V.K., K.P., M.P., N.Y. and T.L. provided computational support. D.R.F., H.F.,

A.D., M.A.S. and B.J.W. contributed to interpretation of the results. B.Z., P.H., J.E.G.L., S.W., H.Z. and S.A.T. wrote the manuscript. A.C., O.A.B., J.C.M., R.A.B., H.Z. and S.A.T. supervised team members. All authors contributed to the discussion and editing of the manuscript.

**Competing interests** In the past 3 years, S.A.T. has consulted for or been a member of scientific advisory boards at Qiagen, Sanofi, GlaxoSmithKline and ForeSite Labs. She is a consultant and equity holder for TransitionBio and EnsoCell. J.C.M has been an employee of Genentech, Inc. since September 2022. The remaining authors declare no competing interests.

**Additional information**
**Correspondence and requests for materials** should be addressed to Hongbo Zhang or Sarah A. Teichmann.

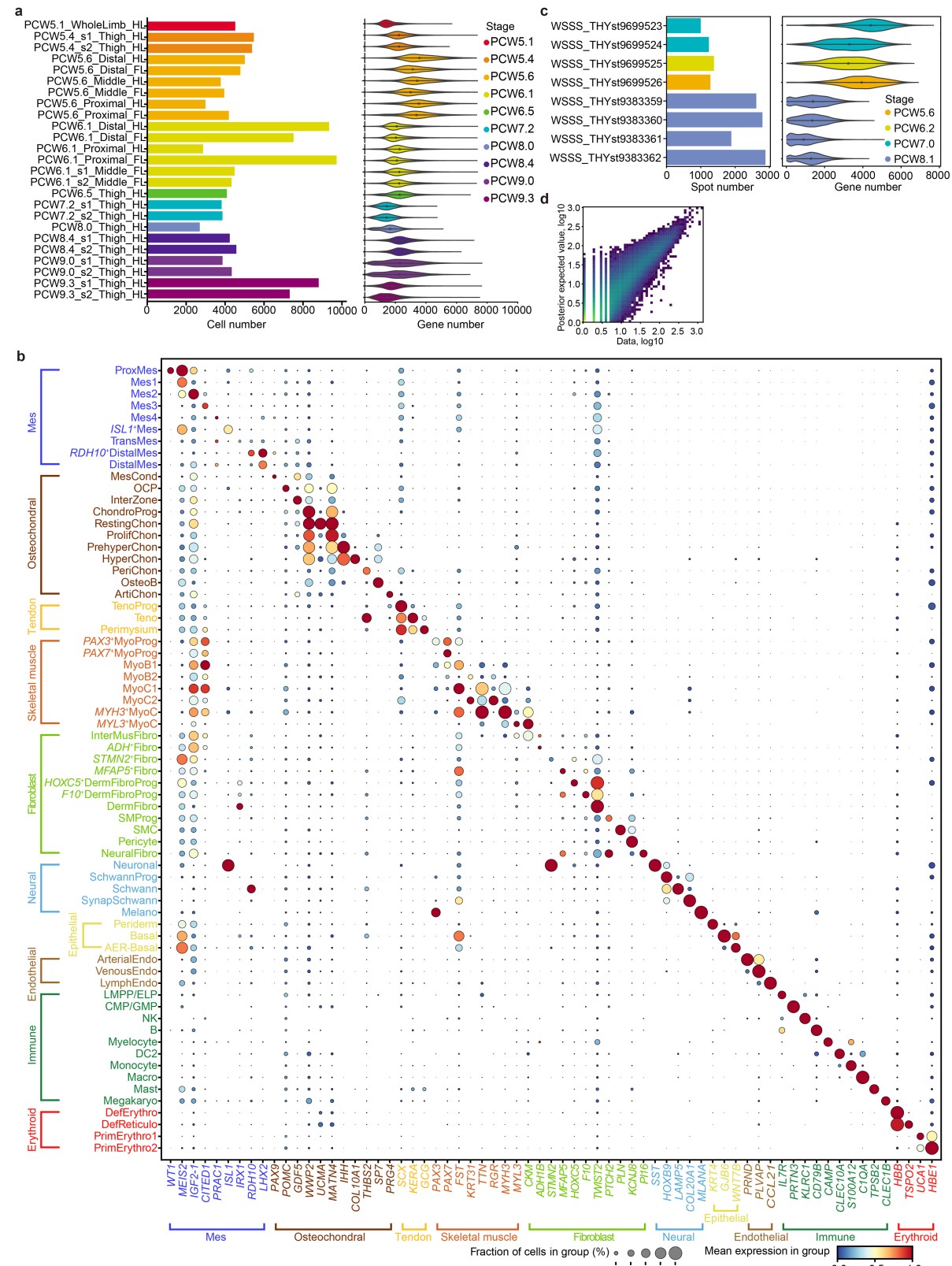

**Extended Data Fig. 1 | Data quality and preprocessing of human scRNA-seq and spatial visium data. a**, Bar plot and violin plot showing the sample size and per-cell quality of each library, separately, coloured by stage. **b**, Dot plot showing the expression level of marker genes for each cell cluster. The colour bar indicates the linearly scaled mean of expression level. Cluster abbreviations same as Fig. 1. **c**, Bar plot and violin plot showing the number of voxels and per-voxel quality of each 10x Visium library, separately, coloured by stage (n = 1 library per bar/violin). **d**, Scatter plot showing the reconstruction accuracy of cell2location.

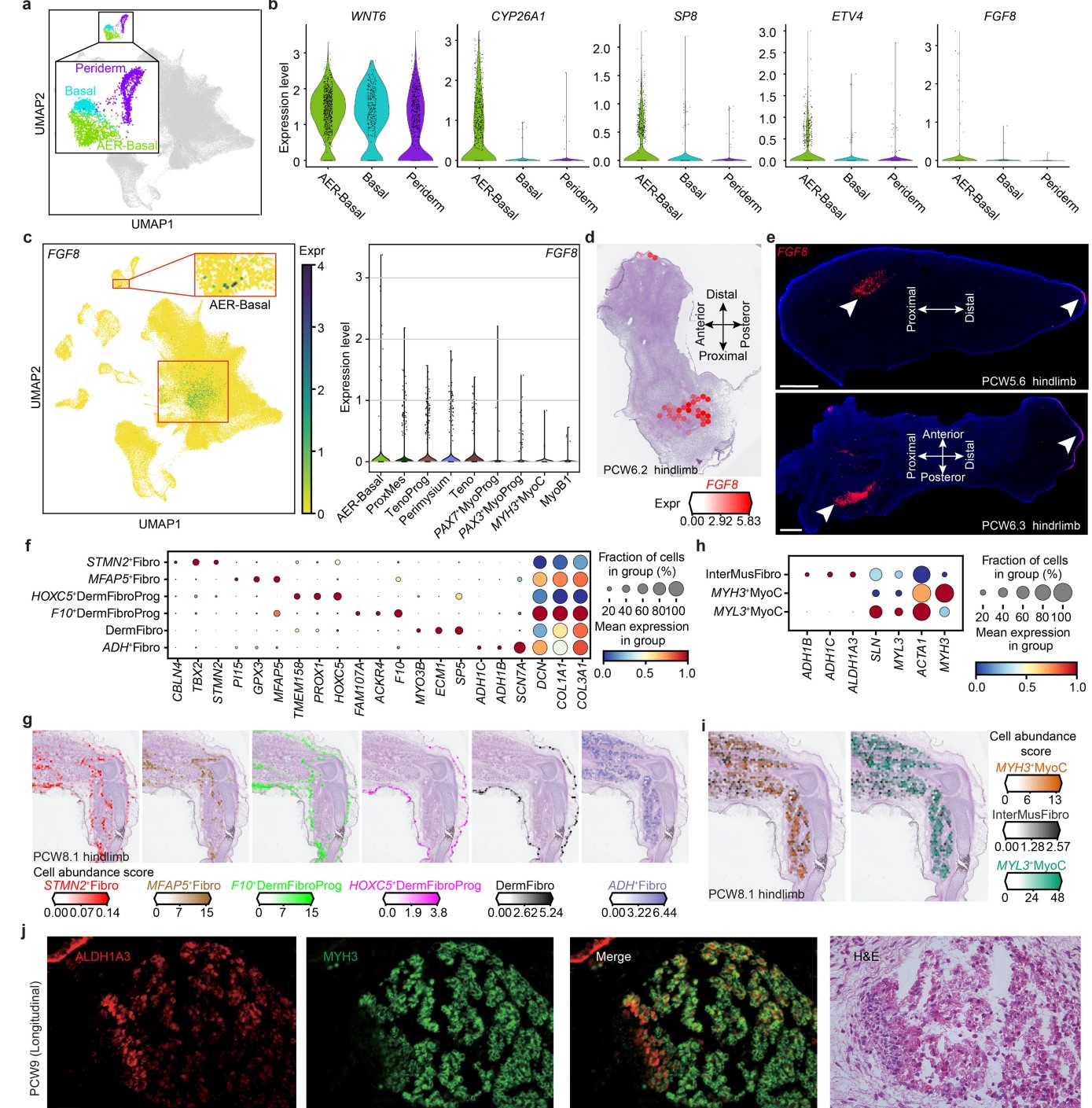

**Extended Data Fig. 2 | The heterogeneity of epidermis and fibroblast.**
**a**, Uniform manifold approximation and projection (UMAP) visualization of AER-basal, basal and periderm cells. **b**, Violin plot showing the normalised and log-transformed expression level of *WNT6*, *CYP26A1*, *SP8*, *ETV4* and *FGF8* in AER-basal, basal and periderm cells. **c**, UMAP (left panel) and violin plot (right panel) showing the normalised and log-transformed expression level of *FGF8* in the human limb. ProxMes, proximal mesenchyme; TenoProg, tendon progenitor; MyoProg, myogenic progenitor; MyoB, myoblast; MyoC, Myocyte. **d**, Heatmap across tissue section from PCW6.2 (post conception week 6 plus 2 days) human hindlimb showing *FGF8* expression. **e**, RNA-ISH of tissue sections

from human hind limb showing the expression pattern of *FGF8*. Scale bar, 500 μm. **f**, **h**, Dot plot showing the expression level of marker genes for selected fibroblast (Fibro) clusters (**f**) and muscle interstitial fibroblast (InterMusFibro) (**h**). The colour bar indicates the linearly scaled values of expression level. DermFibro, dermal Fibro; DermFibroProg, DermFibro progenitor. **g**, **i**, Heatmaps across tissue sections from PCW8.1 human hindlimb showing inferred abundance of each fibroblast cluster (**g**) and InterMusFibro (**i**). **j**, Immunofluorescence staining of MYH3 and ALDH1A3 on the skeletal muscle tissue (as also shown by H&E staining) from a PCW9 longitudinal section. n = 2. Scale bar, 50 μm.

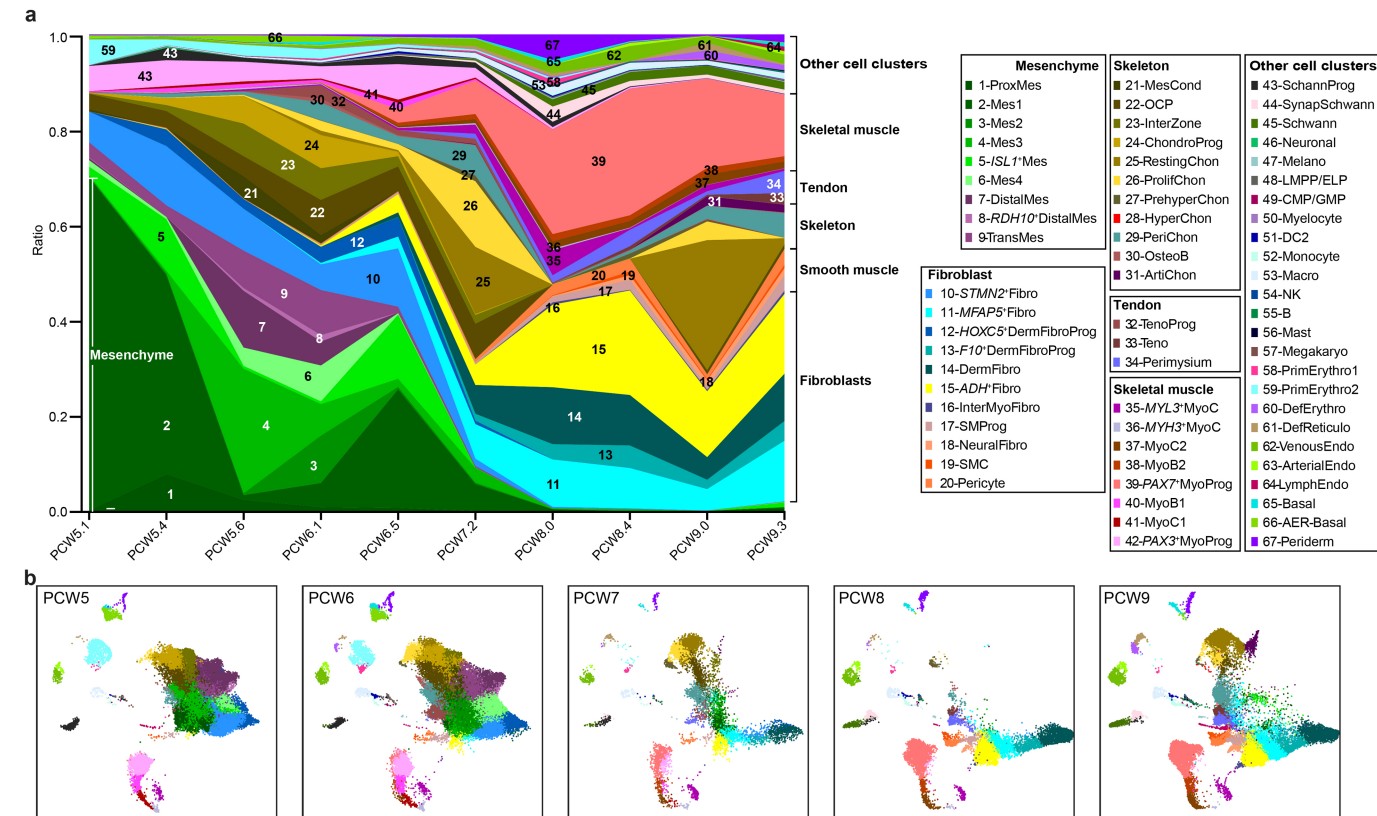

**Extended Data Fig. 3 | Dynamic changes in cell clusters of the human embryonic limb over developmental time. a**, Fraction of cell cluster per time point, coloured by cell clusters and grouped by cell compartment. Cluster abbreviations same as Fig. 1. **b**, Uniform manifold approximation and projection (UMAP) visualisation of cells per post conception week (PCW), coloured by cell cluster in **a**.

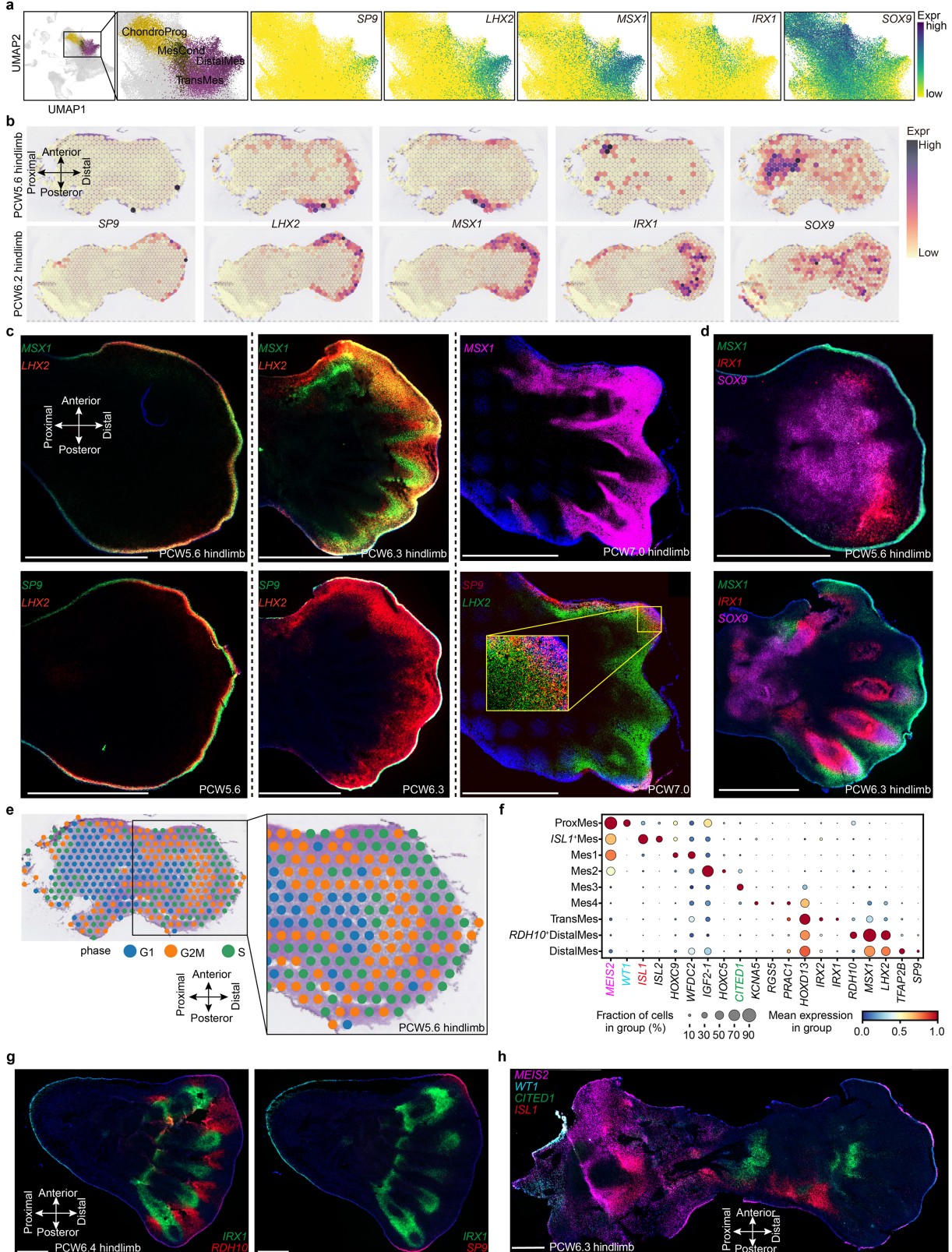

**Extended Data Fig. 4** | See next page for caption.

**Extended Data Fig. 4 | The heterogeneity of Mesenchyme. a**, Uniform manifold approximation and projection (UMAP) plot showing the cell clusters of Chondrogenic progenitor (ChondroProg), mesenchymal condensate cell (MesCond), transitional mesenchyme (TransMes) and distal mesenchyme (DistalMes) (left panel) as well as the expression level of *SP9*, *LHX2*, *MSX1*, *IRX1* and *SOX9*. The colour bar indicates the normalised and log-transformed expression values. **b**, Heatmaps across tissue sections from human hindlimb at stage of PCW5.6 (post conception week 5 plus 6 days) and PCW6.2 showing the spatial expression pattern of *SP9*, *LHX2*, *MSX1*, *IRX1* and *SOX9*. The colour bar indicates the normalised log-transformed expression values. **c**, **d**, RNA-ISH of tissue sections from human hindlimb showing the spatial expression pattern of *SP9*, *LHX2* and *MSX1* (**c**), as well as *MSX1*, *IRX1* and *SOX9* (**d**) at different stage. Scale bar, 1 mm. **e**, Heatmaps across tissue sections from human hindlimb at stage of PCW5.6 showing the cell cycle of G1, G2M and S phase. **f**, Dot plot showing the expression level of marker genes for different cell clusters of mesenchyme. The colour bar indicates the linearly scaled mean of expression level. ProxMes, proximal Mes. **g**, **h**, RNA-ISH of tissue sections from human hindlimb showing the spatial expression pattern of *IRX1*, *SP9* and *RDH10* (**g**), as well as *MEIS2*, *WT1*, *CITED1* and *ISL1* (**h**) n = 2-4 for RNA-ISH. Scale bar, 1 mm.

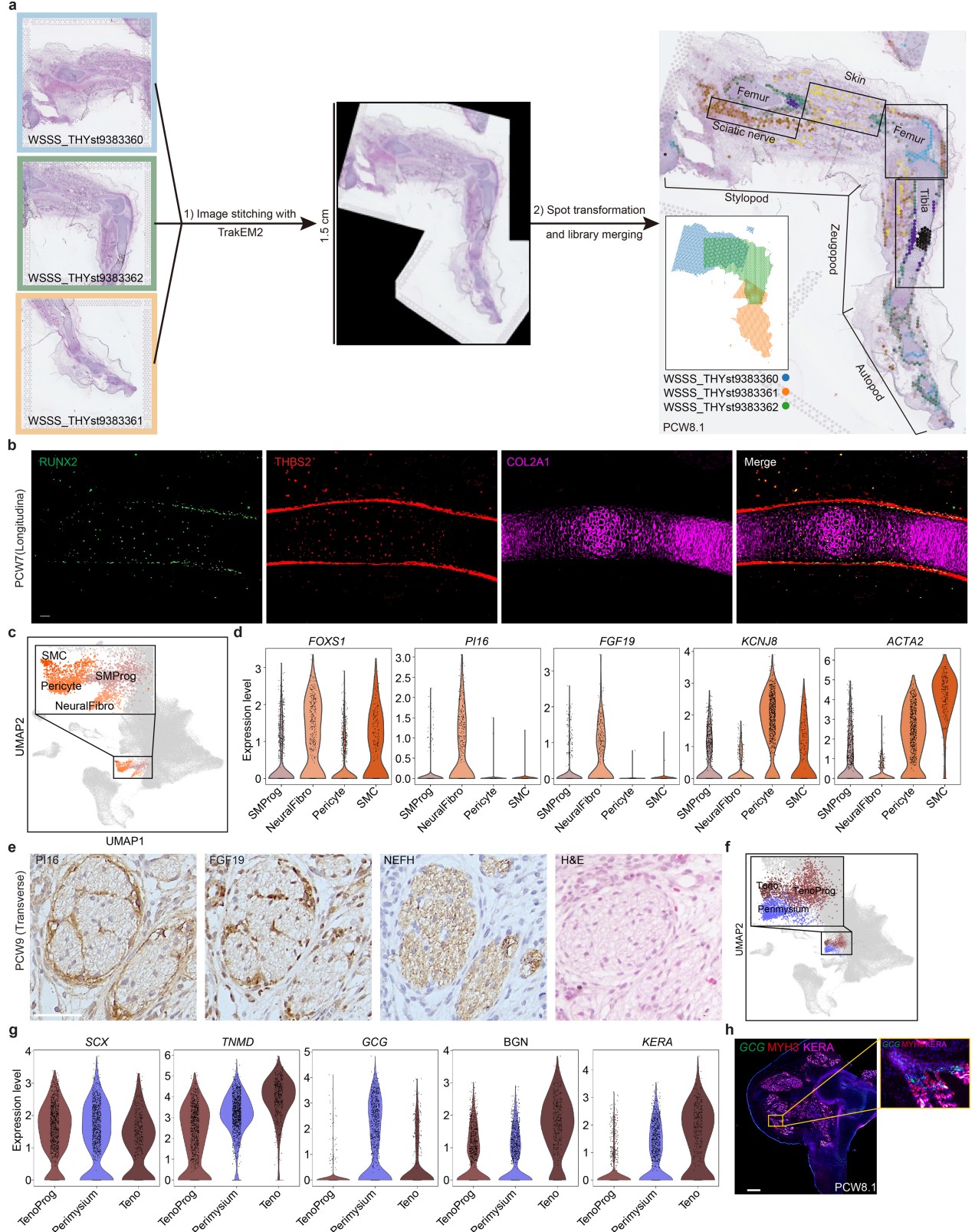

**Extended Data Fig. 5** | See next page for caption.

**Extended Data Fig. 5 | Identification of novel cell types at spatial and single-cell level. a**, The VisiumStitcher workflow for merging the 10x Visium spatial data of the human limb. **b**, Immunofuorescence staining of RUNX2, THBS2 and COL2A1 on the longitudinal section of the tibia from a PCW7 embryo. n = 3. Scale bar, 50 μm. **c**, Uniform manifold approximation and projection (UMAP) visualization of smooth muscle progenitor (SMProg), neural fibroblasts (NeuralFibro), pericyte and smooth muscle (SMC). **d**, Violin plot showing the expression level of *FOXS1*, *PI16*, *FGF19*, *KCNJ8* and *ACTA2* in SMProg, NeuralFibro, pericyte and SMC, using normalised and log-transformed values. **e**, Immunohistochemical staining of PI16 and FGF19 showing the NeuralFibro in the sciatic nerve at PCW9. The neurofilament was stained with NEFH antibody. A neighbouring section stained with H&E solution is also shown. n = 4. Scale bar, 50 μm. **f**, UMAP visualization of tendon progenitor (TenoProg), tenocytes (Teno) and perimysium cells. **g**, Violin plot showing the expression level of *SCX*, *TNMD*, *GCG*, *BGN* and *KERA* in TenoProg, Perimysium, and Teno. The expression level of genes is the normalised and log-transformed values. **h**, RNA-ISH (*GCG*) combined with immunohistochemistry (MYH3 and KERA) of tissue sections from human hind limb showing the spatial expression pattern of *GCG*, MYH3 and KERA. n = 2. Scale bar, 1 mm.

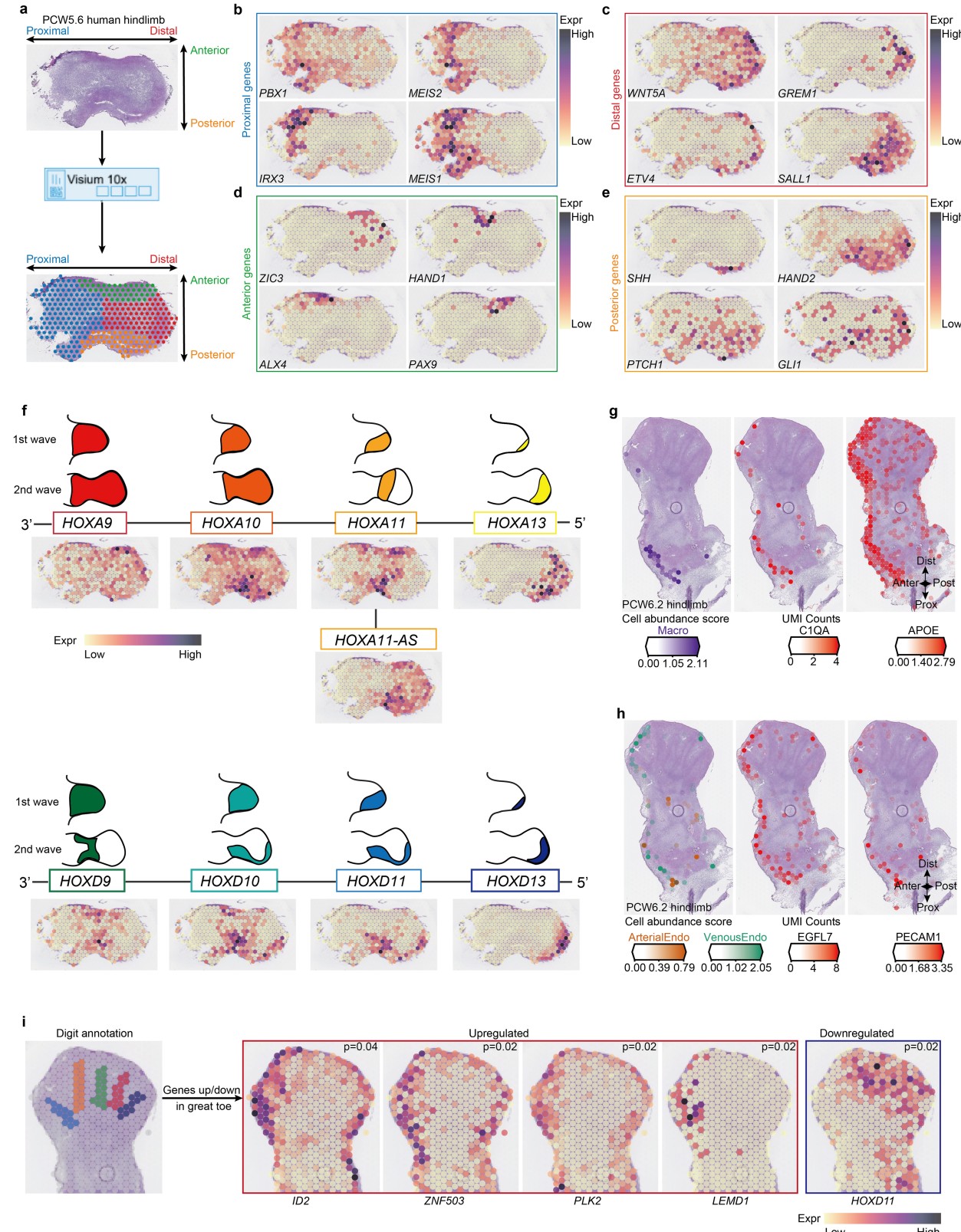

**Extended Data Fig. 6** | See next page for caption.

**Extended Data Fig. 6 | Spatial expression patterns of genes that determine human limb axis formation and morphogenesis. a**, Overview of analysis workflow to identify genes specific to spatial location. **b-e**, Heatmaps across tissue section from the human hindlimb at stage of PCW5.6 showing spatial expression pattern of genes specific to proximal (**b**), distal (**c**), anterior (**d**) or posterior (**e**) regions. The expression level of genes is the normalised and log-transformed values. **f**, Heatmaps across tissue section from human hindlimb at stage of PCW5.6 showing spatial expression pattern of homeobox (HOX) A (top panel) and D (bottom panel) family genes. **g**, **h**, Heatmaps across tissue sections from the human hindlimb at stage PCW6.2 showing inferred abundance of macrophage (**g**) and endothelial cells (vein endothelial cells (VeinEndo) and arterial endothelial cells (ArterialEndo), **h**) as well as expression of maker genes. Anter, anterior; Post, posterior; Prox, proximal; Dist, distal. The expression level of genes is the normalised and logarithmic value of raw counts. **i**, Spatially resolved heatmaps across tissue section from the human hindlimb at stage of PCW6.2 showing spatial expression pattern of digit-associated genes in normalized and log-transformed values.

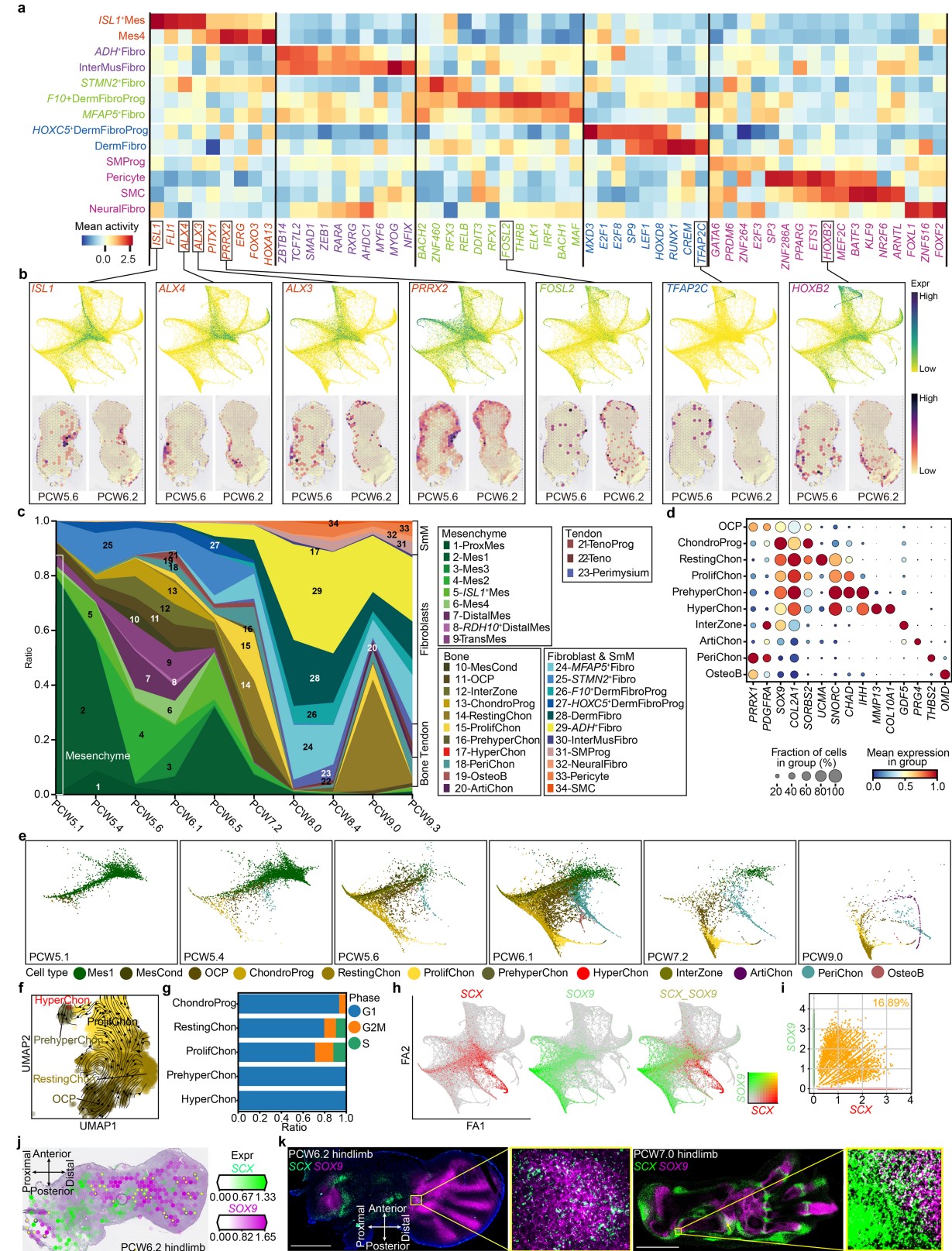

**Extended Data Fig. 7** | See next page for caption.

**Extended Data Fig. 7 | The transcriptional regulation of LPM differentiation in the human limb. a**, Heatmap illustrating the vertically normalised mean activity of selected transcription factors for each cell type from soft connective lineage of LPM. **b**, Force-directed graphs (top panel) and heatmaps across tissue section from the human hindlimb at PCW5.6 and PCW6.2 (bottom panel) showing the expression pattern of representative transcription factors in normalised and log-transformed values. **c**, Stacked bar chart showing the fraction of cell cluster per time point, coloured by cell type and grouped by tissue type. PCW, post conception week. SmM, smooth muscle group; other abbreviations as per Fig. 1. **d**, Dot plot showing the expression level of marker genes of osteochondral cell clusters. The colour bar indicates the mean normalised expression level. **e**, Force-directed graph of cells per time point, coloured by cell type in **a**. **f**, Uniform manifold approximation and projection (UMAP) visualization of the chondrocyte lineage with arrows representing inferred differentiation directions (See Methods). **g**, Stacked bar chart showing the fraction of phase of cell cycle per osteochondral cell cluster. **h**, Force directed graph showing the expression level of *SCX* and *SOX9*. The colour bar indicates the normalised and log-transformed expression values. **i**, Scatter plots showing the expression level of *SCX* and *SOX9* expression in all LPM-derived cells in normalised and log-transformed values. The percentages of double positive cells are given. **j**, Heatmap across tissue section from the human hindlimb at PCW6.2 showing *SCX* and *SOX9* expression in normalised and log-transformed values. The voxels marked with yellow asterisks express both *SCX* and *SOX9*. **k**, RNA-ISH of tissue sections from the human hindlimb showing the expression of *SCX* and *SOX9* in situ.

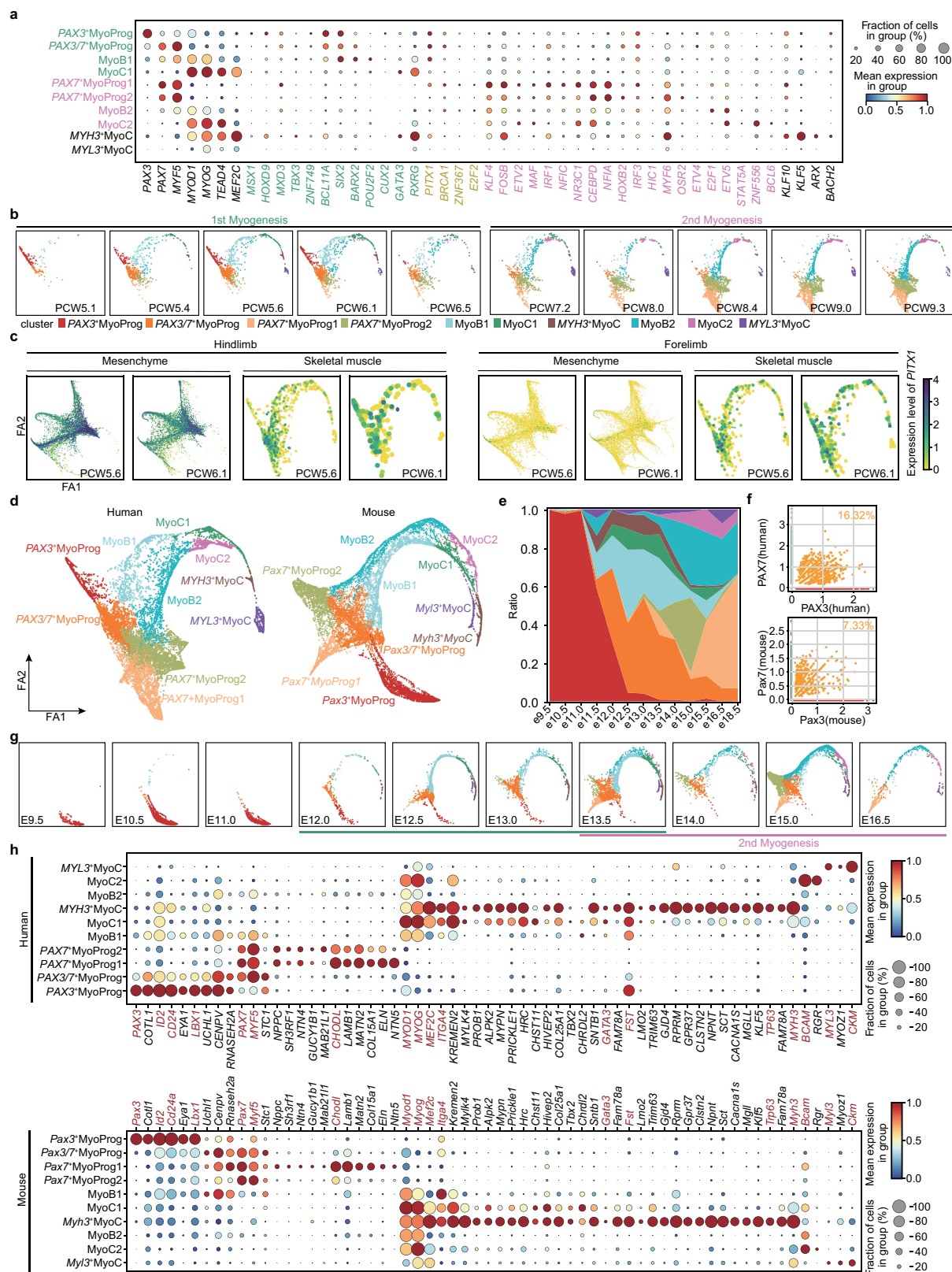

**Extended Data Fig. 8** | See next page for caption.

**Extended Data Fig. 8 | The transcriptional regulation of myogenesis in human and mouse. a**, Dot plot showing expression level of transcription factors per cell cluster in humans, coloured by group (green, first myogenesis; pink, second myogenesis; yellow, both). The colour bar indicates the linearly scaled mean of expression level. MyoProg, myogenic progenitor; MyoB, myoblast; MyoC, myocyte. **b**, Force-directed graph showing muscle populations per time point. **c**, Force-directed graph showing the expression of *PITX1* between forelimb (left panel) and hindlimb (right panel) in cells derived from mesenchyme and skeletal muscle lineage. The expression level of genes is the normalised and logarithmic value of raw counts. **d**, Force-directed graph of human and mouse skeletal muscle cells, coloured by cell clusters. **e**, Stacked bar chart showing the fraction of mouse cell clusters per time point, followed by the colour code of mouse cell clusters in d. **f**, Scatter plots of *PAX3* and *PAX7* expression (normalised and log-transformed) in mouse and human skeletal muscle cells. The percentages of double positive cells are given. **g**, Force-directed graph of mouse cells per time point, coloured by cell cluster in **d. h**, Dotplots of selected genes expressed in humans and mice. The colour bar indicates linearly scaled average expression levels.

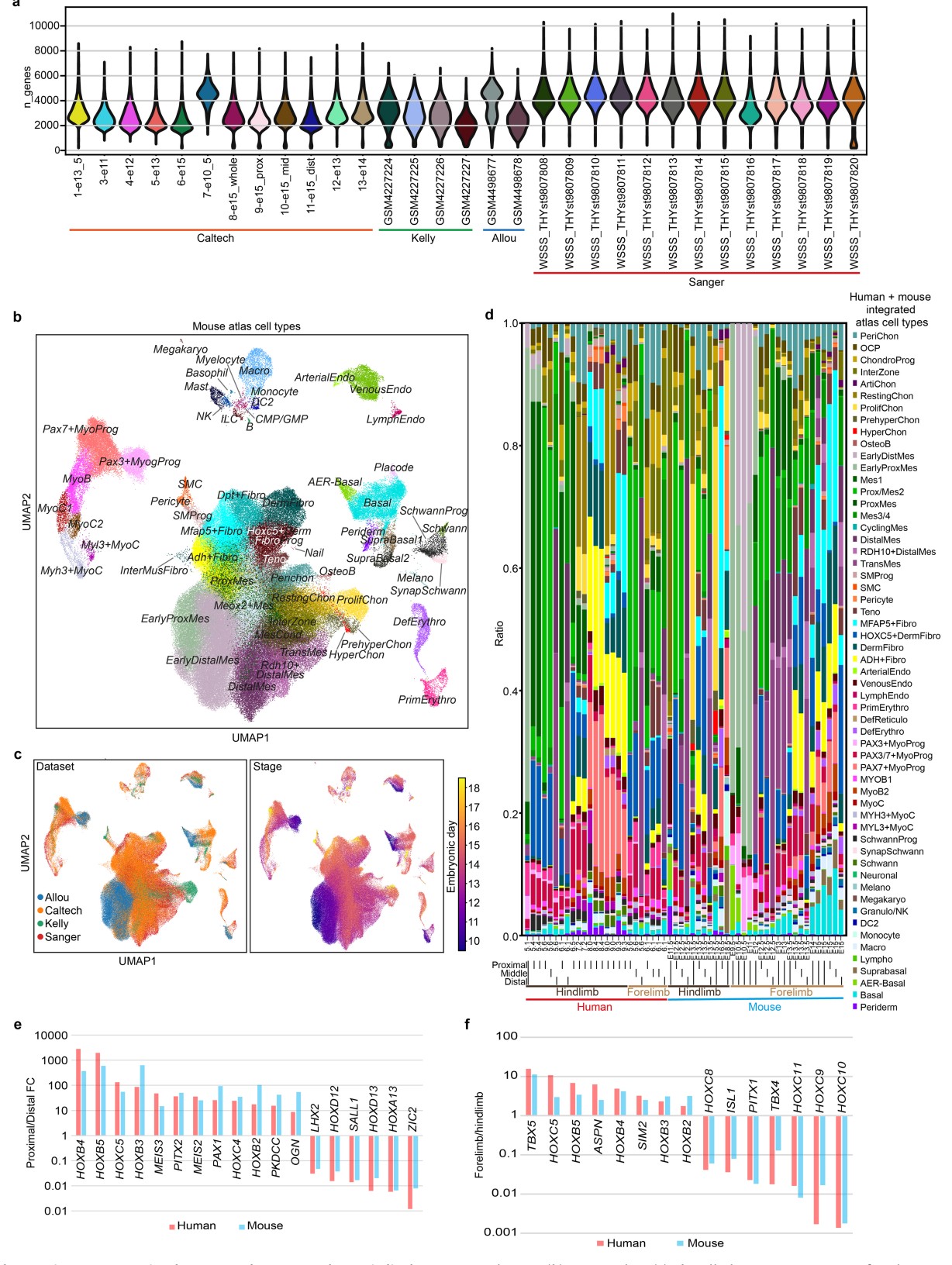

**Extended Data Fig. 9 | Comparing human and mouse embryonic limbs.**
**a**, Violin plots of sample quality for all the scRNA-seq data in the integrated atlas, coloured by library ID and group by dataset at the bottom. **b**, **c**, The integrated mouse scRNA-seq data projected on a shared UMAP plane, coloured by cell clusters (**b**) or metadata (**c**). **d**, Cell-cluster proportions of each scRNA-seq library with dissection region, location and species labelled at the bottom. **e**, Genes enriched in proximal or distal segments in human and mouse. **f**, Genes enriched in the forelimb or hindlimb in human and mouse.

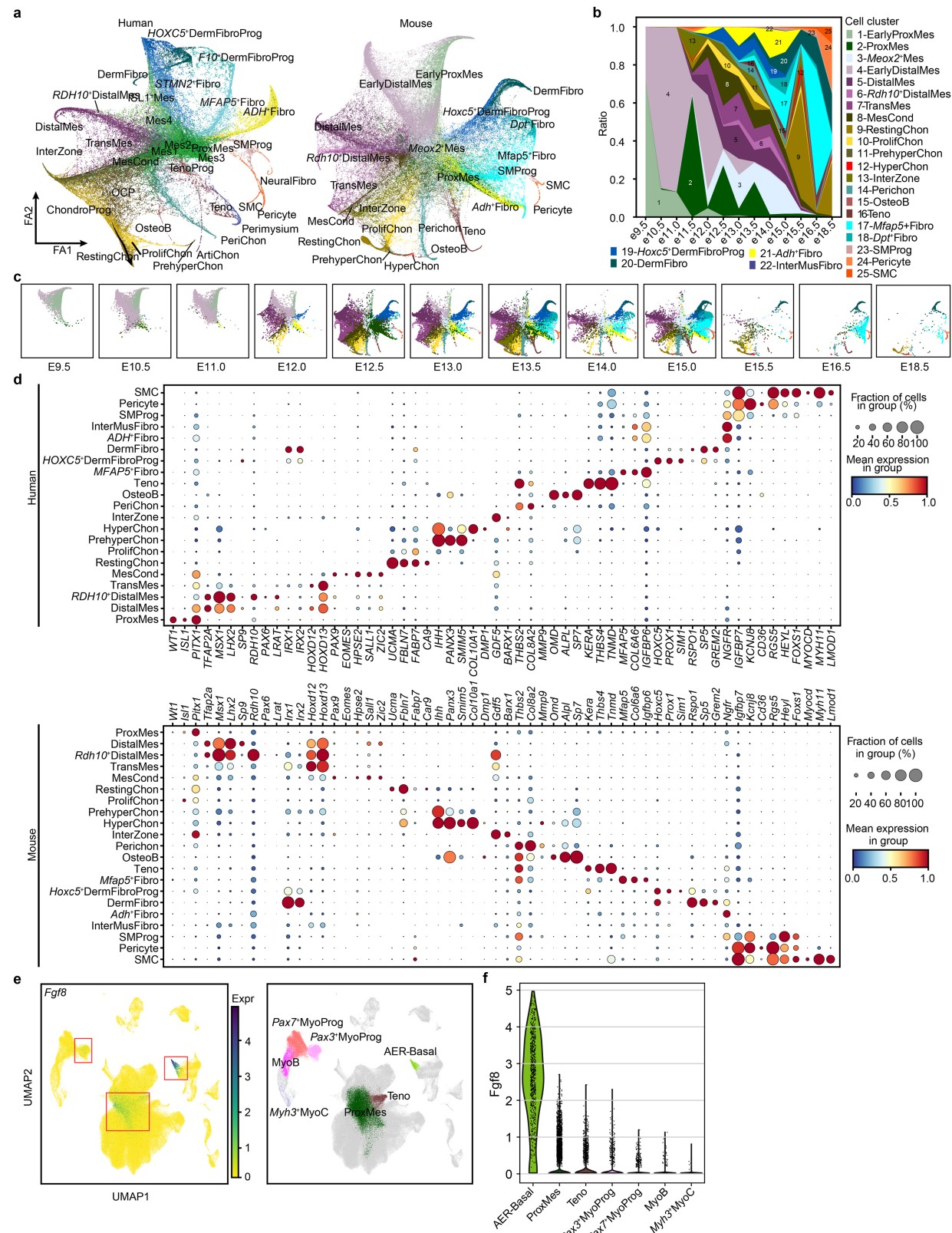

**Extended Data Fig. 10 | The LPM lineage in human and mouse. a**, Force-directed graph of human and mouse LPM-derived cells, coloured by cell clusters. **b**, Stacked bar chart showing the fraction of mouse cell clusters per time point, followed by the colour code of mouse cell clusters in **a**. **c**, Force-directed graph of mouse cells per time point, coloured by cell cluster in **a**. **d**, Dotplot of selected genes expressed in human and mouse. The colour bar indicates the linearly scaled average expression levels. **e**, UMAP plot showing the expression of *Fgf8* in mouse limb cell atlas. The expression level of gene on the left is the normalised and logarithmic value of raw counts. **f**, Violin plot showing the normalised and log-transformed expression level of *Fgf8* in cell clusters of mouse.

# Reporting Summary

## Statistics

For all statistical analyses, confirm that the following items are present in the figure legend, table legend, main text, or Methods section.

| n/a | Confirmed | |
|---|---|---|
| ☐ | ☒ | The exact sample size (*n*) for each experimental group/condition, given as a discrete number and unit of measurement |
| ☐ | ☒ | A statement on whether measurements were taken from distinct samples or whether the same sample was measured repeatedly |
| ☐ | ☒ | The statistical test(s) used AND whether they are one- or two-sided<br>*Only common tests should be described solely by name; describe more complex techniques in the Methods section.* |
| ☐ | ☒ | A description of all covariates tested |
| ☐ | ☒ | A description of any assumptions or corrections, such as tests of normality and adjustment for multiple comparisons |
| ☐ | ☒ | A full description of the statistical parameters including central tendency (e.g. means) or other basic estimates (e.g. regression coefficient) AND variation (e.g. standard deviation) or associated estimates of uncertainty (e.g. confidence intervals) |
| ☐ | ☒ | For null hypothesis testing, the test statistic (e.g. *F*, *t*, *r*) with confidence intervals, effect sizes, degrees of freedom and *P* value noted<br>*Give P values as exact values whenever suitable.* |
| ☒ | ☐ | For Bayesian analysis, information on the choice of priors and Markov chain Monte Carlo settings |
| ☐ | ☒ | For hierarchical and complex designs, identification of the appropriate level for tests and full reporting of outcomes |
| ☒ | ☐ | Estimates of effect sizes (e.g. Cohen's *d*, Pearson's *r*), indicating how they were calculated |

*Our web collection on statistics for biologists contains articles on many of the points above.*

## Software and code

Policy information about availability of computer code

| Data collection | Software used for data alignment and mapping include: STAR (v2.5.1b52) , 10X Space Ranger software (v.1.1.0) |
|---|---|
| Data analysis | Single cell data analysis was mostly performed using Python (v3.7.4) and scanpy (v1.8.2).<br>scRNA-seq data were aligned by Cellranger (v3.0.2) with hg38 (GRCh38-3.0.0) and mm10 (mm10-2020-A).<br>Visium data were aligned using Space Ranger (v1.1.0)<br>Doublets were removed using Scrublet (v0.2.1).<br>Batch correction was performed using bbknn (v1.5.1).<br>RNA velocity analysis was done by scVelo(v0.24)<br>data integration of human and mouse was performed using MultiMAP (v1.0).<br>Cell-cell communication analysis was performed using CellPhoneDB (v2.1.4).<br>Enrichment analysis of transcription factors was performed using pySCENIC (v0.10.3).<br>Alignment and merging of multiple visium sections was performed using Fiji.<br>Deconvolution of human Visium data was performed using cell2location (v0.1).<br>RNA Velocity analysis was performed using scVelo (v0.24).<br>Colocalisation analysis was done using ilastik (v1.3)<br>Light Sheet Fluorescence Microscopy was supported by Imspector Pro (v7.5.3) and Imaris (v10.0).<br>Flow cytometry analysis was done using FlowJo (v10)<br>MSC knock-down data were plotted using Prism (v9) |

For manuscripts utilizing custom algorithms or software that are central to the research but not yet described in published literature, software must be made available to editors and reviewers. We strongly encourage code deposition in a community repository (e.g. GitHub). See the Nature Portfolio guidelines for submitting code & software for further information.

## Data

Policy information about [availability of data](availability of data)

All manuscripts must include a [data availability statement](data availability statement). This statement should provide the following information, where applicable:

- Accession codes, unique identifiers, or web links for publicly available datasets
- A description of any restrictions on data availability
- For clinical datasets or third party data, please ensure that the statement adheres to our [policy](policy)

All of our newly generated raw data are publicly available on ArrayExpress (mouse scRNA-seq, E-MTAB-10514; human Visium, E-MTAB-10367; human scRNA-seq, E-MTAB-8813). Previously published raw data can be found from ENCODE portal (ENCSR713GIS, https://www.encodeproject.org/publication-data/ENCSR713GIS/) and GEO (GSE137335 https://www.ncbi.nlm.nih.gov/geo/query/acc.cgi?acc=GSE137335 and GSE142425 https://www.ncbi.nlm.nih.gov/geo/query/acc.cgi?acc=GSE142425). Processed data and be downloaded and visualisedvisualized at our data portal (https://limb-dev.cellgeni.sanger.ac.uk/). All the source data for figures can be found in supplementary Excel files.

## Research involving human participants, their data, or biological material

Policy information about studies with [human participants or human data](human participants or human data). See also policy information about [sex, gender (identity/presentation), and sexual orientation](sex, gender (identity/presentation)) and [race, ethnicity and racism](race, ethnicity and racism).

| | |
|---|---|
| Reporting on sex and gender | This study focuses on the limb development of human embryo, and we included data from medical aborted female and male embryos. From the stage of PCW5-9 we have analysed, we do not observe sex and gender difference between females and males. |
| Reporting on race, ethnicity, or other socially relevant groupings | *Please specify the socially constructed or socially relevant categorization variable(s) used in your manuscript and explain why they were used. Please note that such variables should not be used as proxies for other socially constructed/relevant variables (for example, race or ethnicity should not be used as a proxy for socioeconomic status).*<br>*Provide clear definitions of the relevant terms used, how they were provided (by the participants/respondents, the researchers, or third parties), and the method(s) used to classify people into the different categories (e.g. self-report, census or administrative data, social media data, etc.)*<br>*Please provide details about how you controlled for confounding variables in your analyses.* |
| Population characteristics | We used first trimester embryos (age, post conception weeks 5-9) from voluntary medical abortion. Tissue samples used for human scRNA-seq/Visum and validation experiments were obtained from donors of British and Chinese, respectively. No developmental abnormalities were visible or known in any of the embryos collected. |
| Recruitment | Medical aborted embryos were collected with the agreement of the pregnant female. The termination time point were decided by the female and doctor. |
| Ethics oversight | First trimester human embryonic tissue was collected from elective termination of pregnancy procedures at Addenbrookes Hospital, Cambridge, UK under full ethical approval (REC-96/085; for scRNA-seq and Visium), at Guangzhou Women and Children's Medical Center, China under approval of the Research Ethics Committee of Sun Yat-sen University (ZSSOM-2019-075) and Guangzhou Women and Children's Medical Centre (2022-050A01, for In-situ hybridisation and immunohistochemistry). Experiments were also followed the 2021 International Society for Stem Cell Research (ISSCR) guidelines in working on human embryos. Informed written consent was obtained from all donors before abortion and tissue collection. No developmental abnormalities were visible or known in any of the embryos collected. All human data generated from China was registered at China National Center for Bioinformation (PRJCA012474) and has been approved by the Chinese Ministry of Science and Technology for the Review and the Approval of Human Genetic Resources (2023BAT0445). For light-sheet fluorescence microscopy, tissues were obtained through INSERM's HuDeCA Biobank and made available in accordance with the French bylaw. Permission to use human tissues was obtained from the French agency for biomedical research (Agence de la Biomédecine, Saint-Denis La Plaine, France; N° PFS19-012) and INSERM Ethics Committee (IRB00003888). Written, informed consent was given for tissue collection by all patients. Embryonic age (post conception weeks, PCW) was estimated using the independent measurement of the crown rump length (CRL), using the formula PCW (days) = 0.9022 × CRL (mm) + 27.372. |

Note that full information on the approval of the study protocol must also be provided in the manuscript.

# Field-specific reporting

Please select the one below that is the best fit for your research. If you are not sure, read the appropriate sections before making your selection.

☒ Life sciences          ☐ Behavioural & social sciences          ☐ Ecological, evolutionary & environmental sciences

For a reference copy of the document with all sections, see [nature.com/documents/nr-reporting-summary-flat.pdf](nature.com/documents/nr-reporting-summary-flat.pdf)

# Life sciences study design

All studies must disclose on these points even when the disclosure is negative.

| | |
|---|---|
| Sample size | For human scRNA seq, n = 1 at PCW5.1; n = 2 at PCW5.4; n = 6 at PCW5.6; n = 6 at PCW6.1; n = 1 at PCW6.5; n = 2 at PCW7.2; n = 1 at PCW8.0; n = 2 at PCW8.4; n = 2 at PCW9.0; n = 2 at PCW9.3;. <br> For human Visum, n = 2 at PCW5.6; n = 3 at PCW6.2; n = 2 at PCW7.0; n = 4 at PCW8.1; <br> For mouse scRNA seq, n = 5 at E12.5; n = 5 at E13.5; n = 2 at E16.5. <br> For experimental validation: <br> RNA In situ hybridization: n = 2-4 for each staining at indicated stage mentioned in the manuscript; <br> Light Sheet Fluorescence Microscopy for MSX1 IRX1 SOX9 staining: n = 2; <br> RUNX2, THBS2, COL2A1 immunofluorescence staining: n = 3; <br> PITX1, PAX3 immunofluorescence staining: hindlimb, n = 4; forelimb, n = 2; <br> ALDH1A3, MYH3 immunofluorescence staining: n = 2; <br> PI16, FGF19, NEFH Immunohistochemistry staining: n = 4; <br> Myoblast isolation and culture: n = 2. <br>  GCG, MYH3, KERA staining: n = 2 <br> Sample size depends on availability of human tissues. We try to include at least two replicates when available. No sample size calculation was performed. |
| Data exclusions | We excluded cells based on the QC thresholds summarized in Methods section. We also removed cell doublets. |
| Replication | We used 1-2 biological and 1-2 technical replicates for human scRNA-seq and Visum; <br> We used 1-6 biological and 1-2 technical replicates for mouse scRNA-seq. <br> We used 2-4 biological and 1-3 technical replicates for experimental validations. <br> All attempts at replication were successful. |
| Randomization | Intentional randomization was not performed. Samples were allocated based on their ages. |
| Blinding | All human specimens were de-identified before analyses. However, selected attributes such as (developmental stage and dissected region) were available to all investigators.  Blinding was not performed during tissue sample collection, analysis of scRNA-seq and Visium, as well as experimental validations, although our initial computational processing used unbiased approaches for all the sequencing samples. A majority of the downstream analyses did not adopt blinding as key sample attributes were needed for accurate cell cluster annotation and downstream analyses to create the atlas. |

# Reporting for specific materials, systems and methods

We require information from authors about some types of materials, experimental systems and methods used in many studies. Here, indicate whether each material, system or method listed is relevant to your study. If you are not sure if a list item applies to your research, read the appropriate section before selecting a response.

## Materials & experimental systems

| n/a | Involved in the study |
|---|---|
| ☐ | ☒ Antibodies |
| ☒ | ☐ Eukaryotic cell lines |
| ☒ | ☐ Palaeontology and archaeology |
| ☐ | ☒ Animals and other organisms |
| ☒ | ☐ Clinical data |
| ☒ | ☐ Dual use research of concern |
| ☒ | ☐ Plants |

## Methods

| n/a | Involved in the study |
|---|---|
| ☒ | ☐ ChIP-seq |
| ☐ | ☒ Flow cytometry |
| ☒ | ☐ MRI-based neuroimaging |

## Antibodies

| | |
|---|---|
| Antibodies used | 1. Anti-RUNX2 Antibody (C-12) (1:50, Santa Cruz, sc-390715) <br> https://www.scbt.com/p/runx2-antibody-c-12 <br> 2. Anti-THBS2 (Thrombospondin 2) Polyclonal Antibody (1:100, Invitrogen, PA5-76418) <br> https://www.thermofisher.cn/cn/zh/antibody/product/Thrombospondin-2-Antibody-Polyclonal/PA5-76418 <br> 3. Anti-COL2A1 (M2139) antibody (1:200, Santa Cruz, sc-52658) <br> https://www.scbt.com/p/col2a1-antibody-m2139/ <br> 4. Anti-PITX1/BFT antibody (1:30, Abcam, ab244308) <br> https://www.abcam.com/pitx1bft-antibody-ab244308.html <br> 5. Anti-PAX3 antibody (1:1, DSHB, AB_528426 supernatant) <br> https://dshb.biology.uiowa.edu/Pax3 <br> 6. Anti-ALDH1A3 antibody (1:50, Proteintech, 25167-1-AP) <br> https://www.ptglab.com/Products/ALDH1A3-Antibody-25167-1-AP.htm <br> 7. Anti-MYH3 (F1.652) antibody (1:3, DSHB, AB_528358 supernatant) <br> 8. Anti-PI16 antibody (1:500, Sigma-Aldrich, HPA043763) |

https://www.sigmaaldrich.com/catalog/product/sigma/hpa043763
9. Anti-FGF19 antibody (1:500, Affinity, DF2651)
http://www.affbiotech.com/goods-6732-DF2651-FGF19_Antibody.html
10. Anti-NEFH (Neurofilament-H) (RMdO 20)antibody (1:1000, Cell Signaling, 2836)
https://www.cellsignal.com/products/primary-antibodies/neurofilament-h-rmdo-20-mouse-mab/2836
11. Alexa Flour 488 goat anti-mouse IgG1 (1:400, Invitrogen, A-21121)
https://www.thermofisher.cn/cn/zh/antibody/product/Goat-anti-Mouse-IgG1-Cross-Adsorbed-Secondary-Antibody-Polyclonal/A-21121
12. Alexa Flour 647 goat anti-mouse IgG2b (1:400, Invitrogen, A-21242)
https://www.thermofisher.cn/cn/zh/antibody/product/Goat-anti-Mouse-IgG2b-Cross-Adsorbed-Secondary-Antibody-Polyclonal/A-21242
13. Alexa Flour 488 goat anti-mouse IgG (H+L) (1:400, Invitrogen, A-11029)
https://www.citeab.com/antibodies/2401117-a-11029-goat-anti-mouse-igg-h-l-highly-cross-adsor
14. Alexa Flour 546 goat anti-rabbit IgG (H+L) (1:400, Invitrogen, A-11035)
https://www.thermofisher.cn/cn/zh/antibody/product/Goat-anti-Rabbit-IgG-H-L-Highly-Cross-Adsorbed-Secondary-Antibody-Polyclonal/A-11035
15. Streptavidin-Peroxidase broad spectrum Immunohistochemical staining kit (Bioss, SP-0022)
http://www.bioss.com.cn/prolook_03.asp?id=AF08169606008548&pro37=9
16. IRX1 (1:200, Sigma-Aldrich, HPA043160)
https://www.sigmaaldrich.com/US/en/product/sigma/hpa043160
17. MSX1 (1:500, R&D Systems, AF5045)
https://www.rndsystems.com/cn/products/human-mouse-msx1-antibody_af5045
18. LHX2 (1:1000, Abcam, ab184337)
https://securedrtest.abcam.com/products/primary-antibodies/lhx2lh2-antibody-epr20449-ab184337.html
19. SOX9 (1:500, Abcam, ab196184)
https://securedrtest.abcam.com/products/primary-antibodies/alexa-fluor-647-sox9-antibody-epr14335-ab196184.html
20. MAFB (1:500, Abcam, ab223744)
https://securedrtest.abcam.com/products/primary-antibodies/mafb-antibody-ab223744.html
21. Donkey Anti-Rabbit IgG H&L (1:800, Alexa Fluor® 555) (Abcam, ab150062)
https://securedrtest.abcam.com/products/secondary-antibodies/donkey-rabbit-igg-hl-alexa-fluor-555-preadsorbed-ab150062.html
22. Donkey Anti-Goat IgG H&L (Alexa Fluor® 790) (1:300, Abcam, ab175745)
https://securedrtest.abcam.com/products/secondary-antibodies/donkey-goat-igg-hl-alexa-fluor-750-preadsorbed-ab175745.html
23. anti-KERA antibody (1:1000, Sigma-Aldrich, HPA039321)
https://www.sigmaaldrich.cn/CN/en/product/sigma/hpa039321
24. CD31 (PECAM-1) Monoclonal Antibody (WM-59 (WM59)), PE (5 µL/Test, eBioscience, 12-0319-41)
https://www.thermofisher.cn/cn/zh/antibody/product/CD31-PECAM-1-Antibody-clone-WM-59-WM59-Monoclonal/12-0319-41
25. CD45 Monoclonal Antibody (HI30), PE (5 µL/Test, eBioscience, 12-0459-41)
https://www.thermofisher.cn/cn/zh/antibody/product/CD45-Antibody-clone-HI30-Monoclonal/12-0459-41
26. CD184 (CXCR4) Monoclonal Antibody (12G5), APC (5 µL/Test, eBioscience, 17-9999-41)
https://www.thermofisher.cn/cn/zh/antibody/product/CD184-CXCR4-Antibody-clone-12G5-Monoclonal/17-9999-41

Validation | For all the above antibodies used in the manuscript, they relevant applications (i.e. FACS or Immunohistochemistry staining) were validated by the manufactures and indicated on the individual website.

# Animals and other research organisms

Policy information about studies involving animals; ARRIVE guidelines recommended for reporting animal research, and Sex and Gender in Research

Laboratory animals | C57BL/6N wild type embryos; All animal procedures were performed according to protocols approved by the Institutional Animal Care and Use Committee at the California Institute of Technology. Animals were housed in an AAALAC accredited facility in accordance with the Guide for the Care and Use of Laboratory Animals. Animal rooms were maintained on a 14:10 h light:dark cycle with an hour-long dawn/dusk period with humidity ranging from 30% to 70% and temperatures ranging from 71 to 75 °F.Age: n = 5 at E12.5; n = 5 at E13.5; n = 2 at E16.5

Wild animals | No wild animals were used in this study

Reporting on sex | Sex information was not recorded at the time of collection due to sample processing time requirements.

Field-collected samples | No field collected samples were used in the study

Ethics oversight | approved by The Institutional Animal Care and Use Committee at the California Institute of Technology

Note that full information on the approval of the study protocol must also be provided in the manuscript.

# Flow Cytometry

## Plots

Confirm that:

☒ The axis labels state the marker and fluorochrome used (e.g. CD4-FITC).

☒ The axis scales are clearly visible. Include numbers along axes only for bottom left plot of group (a 'group' is an analysis of identical markers).

☒ All plots are contour plots with outliers or pseudocolor plots.

☒ A numerical value for number of cells or percentage (with statistics) is provided.

## Methodology

| | |
|---|---|
| Sample preparation | The thighs from human embryos were processed as described (Lapan AD, et al. Methods Mol Biol. 2012), except that the dissociated cells were not treated with erythrocyte lysis solution, and were incubated with anti-human CD31 (eBioscience, 12-0319-41), CD45 (eBioscience, 12-0459-41) and CD184 (eBioscience, 17-9999-41) antibodies for cell sorting. Fluorescent activated cell sorting (FACS, BD, influx) sorted CD31-CD45-CD184+ cells were cultured in complete growth medium DMEM supplemented with 20% FCS and 1% penicillin/streptomycin (Gibco, 15140122). |
| Instrument | BD Influx Cell Sorter |
| Software | Exported raw file was opened with FlowJo (version 10) to analyze cell populations. |
| Cell population abundance | The number of CD31-CD45- cells accounts for approximately 97.24% of embryonic thigh dissociated cells, and the abundance of CD184+ cells is about 0.56%. The percentage for CD31-CD45- CD184+ cells is about 0.54%. |
| Gating strategy | Gating for CD31-CD45- CD184+ to isolate myoblasts: 1) gate on FSC vs SSC to exclude cell debris or small particles and include all cell populations.; 2) gate on PE channel to sort CD31-CD45- cells; 2) gate on APC channel to sort CD31-CD45- CD184+ cells as primary myoblasts. |

☒ Tick this box to confirm that a figure exemplifying the gating strategy is provided in the Supplementary Information.

