## [Peer Review File · Nature]

Manuscript Title: A human embryonic limb cell atlas resolved in space and time

Reviewer Comments & Author Rebuttals

Reviewer Reports on the Initial Version:

Referees' comments:

Referee #1 (Remarks to the Author):

This manuscript is one of the on-going efforts from the Human Developmental Cell Atlas community to provide vital information on human development. Compared to the already published efforts (ref. 11-15), the manuscript might be described as being the most ambitious. The study scope spans early patterning of the limb, lineage diversification of lateral plate mesoderm derived limb progenitors, as well as skeletal myogenesis, and exploration of species-specific differences, each would normally warrant an independent paper. As spatial and temporal context is particularly important in the developing limb, with limb progenitors undergoing similar molecular differentiation processes throughout the limb across space and time to yield distinct functional structures, their transcriptomic data is supplemented by gross spatial annotation by dissection along the proximo-distal axis, further supplemented by spatial transcriptomic approach, with 10 temporal points in four weeks time window, a distinct feature to the other already published human limb development dataset (Xi et al. 2020, ref. 18). Moreover, to discern what is known and what is unknown, the authors conduct a meta-analysis, aggregating existing transcriptomic datasets of the murine model, supplemented their own and tried to compare them to their human dataset. Lastly, they provide a standardized user-friendly interactive browser platform for maximum accessibility.

While this ambitious study has the potential to make a significant contribution to the field, in its current form the approaches are not well enough integrated, and there are too many critical flaws, for it to achieve this goal.

Foremost, their results are not really well grounded with the current conceptual framework of the limb development, using terms that are used out of the context, or relying on outdated terms, obfuscating what is known and unknown in the field.

An example of misunderstanding of key terms is “posterior prevalence” (line 265-279), which the authors use in describing the spatial expression pattern of HOX genes. Posterior prevalence is a general property attributed to vertebrate HOX proteins describing the dominant effect of more posterior HOX proteins over the function of anterior orthologs in common (overlapping) areas of expression, a term coined by Denis Duboule in 1991 (D. Duboule, Patterning in the vertebrate limb, *Curr. Opin. Genet. Dev.*, 1:211-216) which has nothing to do with their spatial colinearity, or domains of gene expression as being used in the manuscript (e.g. line 311-312).

Another example can be found when the authors cite reference 156 (Feregrino & Tschopp, *Dev. Dyn.*

2021) with “evolutionary turnover”, which has nothing to do with the pace of developmental maturation at all, but rather is about the degree of conservation of gene expression modules (line 577-579) over evolutionary time and would be irrelevant in the context of finding more number of immune and epithelial cell type capture in the mouse dataset.

One of the most critical usage of an outdated term is the “progress zone” (line 172-173, followed by numerous usage throughout the manuscript), relating to a distal region defined in a now discredited model of proximodistal limb patterning proposed by Lewis Wolpert and colleagues prior to the advent of molecular genetic analyses. The term should not be used. The current view of PD specification is concisely summarized in an excellent review by Caitlin McQueen and Matt Towers. Indeed, the reviewer would urge the authors to read the entire review carefully, and interpret their data in the context of the information they provide. (C. McQueen and M. Towers Establishing the pattern of the vertebrate limb Development 147 (2020)).

Perhaps due to such weak grounding, coupled by a largely uncritical attitude towards the output of computational methods, some demonstrably incorrect presentations were included in some of the figures.

How the authors depict chondrocyte differentiation in figure 3 is a useful illustration. The established consensus of chondrocyte differentiation is that after mesenchymal condensation (line 365), resting zone chondrocytes differentiate into proliferating, prehypertrophic, then hypertrophic chondrocytes in a sequential manner (reviewed in Kozhemyakina, Lassar and Zelzer A pathway to bone: signaling molecules and transcription factors involved in chondrocyte development and maturation, Development (2015)). In the manuscript, the authors use the term “chondroblast” (line 402) that is rarely used in the context of limb development. Nor is the term “maturing chondrocytes” (note that in their interactive browser for LPM, it is annotated as ‘intermediate chondrocyte’ which is also not at all used in the field) used in the field frequently, and if so, includes the maturation process from resting to prehypertrophic, and hypertrophic chondrocytes. None of the presentations (Extended figure 1, Figure 3b, Figure 3d) show clear markers that define and distinguish chondroblast from “maturing chondrocytes”, except that the authors mention “immature” for chondroblasts (line 392) so one has to guess what these two entities represent and connected to the existing model. In figure 3a, the direction of cell differentiation is depicted as “chondroblasts” differentiating to resting chondrocytes in one branch, and the other differentiating into “maturing chondrocytes” to “prehypertrophic chondrocytes”. Such bifurcating differentiation process separates, and in a sense reverses the direction between resting zone chondrocytes and prehypertrophic chondrocyte states, which contradicts the consensus model, calling for a careful re-examination of the validity of their computational approach of lineage reconstruction (diffusion map, PAGA and force directed embedding). Such a contradiction might have arisen from the author’s ambitious attempt to aggregate all developmental processes from PCW5.1 to PCW9.3 together, which collapses two very distinct processes of mesenchymal condensation, formation of the skeletal elements with the epiphyseal growth plate development. In this context, stage, region-specific analysis, overlaid together, might better validate and elucidate the various differentiation processes.

Another example of a concerning presentation is found in figure 4. The authors present PITX1

expressed in forelimb mesenchyme but not in hindlimb mesenchyme (Figure 4e). Pitx1 has been found exclusively in hindlimb in many vertebrate animals (c.f. Logan and Tabin, Role of Pitx1 Upstream of Tbx4 in specification of hindlimb identity, *Science* 283, 1999; DeLaurier, Schweitzer and Logan, Pitx1 determines the morphology of muscle, tendon, and bones of the hindlimb, *Dev Biol.* 2006). Therefore, the mesenchymal expression data in itself is a very surprising result but the authors only treat this finding in a passing comment (line 494-495) in the context of muscle cell expression. Even internally, the presentation seems to be inconsistent. How do the authors reconcile the presentation in Figure 4e to even the immunohistochemistry in Figure 4f, where they display abundant PITX1 immunofluorescence in the hindlimb that are PAX3 negative? Also that almost all PITX1 immunofluorescence signals in the PCW5.5 forelimb specimen are derived from PAX3+ cells?

Overall, the manuscript would benefit from more cross-examination between mouse datasets and human datasets, rather than positioning the comparative aspect at the end of the manuscript, as it reads more of an afterthought. For example, conducting a similar lineage progression analysis on the mouse dataset (e.g. for the chondrocyte lineage) and validating the finding with the existing literature would help to justify the usage of their pipeline to the human data, and also better highlight the human-specific characteristics.

In addition to these issues, the authors fail to clearly outline the limitations of their methodological approaches.

As for the spatial transcriptomics, the reviewer notes that the 10X VISIUM methodology has a severe limitation in terms of spatial resolution (center to center distance is 100um, worse than naked eye resolution of ~40um - cf. Cho et al. Microscopic examination of spatial transcriptome using Seq-Scope, *Cell*, 2021). The authors use computational methods to deconvolute the transcriptome into cell types, but this does not necessarily inform the readers how many cells (not cell types) might have been captured in one voxel. Based on the interactive browser, the number of deconvolved clusters appear to have an upper range between 6 and 9 different cell types/states. Thus, while useful in discerning transcriptomes at a large distance, as the authors demonstrate with the perineural fibroblasts associated with nerves, ADH1B(ADH+FIB) fibroblasts, associated with muscle cells, and the perimysium cells separated from the tenocytes (line 238-241), it becomes more tenuous as the distance in question decreases.

The resolution limit is almost reached when they describe the PCW6.2 human digits (Figure 2), where the distance between interdigital space and digits appears to be either one or at most two voxels. As all analyses were performed with manual labeling of voxels based on H&E sections, at the very least it is important to have a sensitivity analysis of inclusion/exclusion of voxels in terms of the results presented in the manuscript. As an example, some of the genes that the authors highlighted also in the text, such as CYP26B1, IRX1 appear in violin plot barely differentially expressed albeit with significant p-values. Conversely, most of the genes except BMP7 really do not have strong interdigital structure observed based on the spatial representation (e.g. CCL2), and most likely derive the differential expression result because of the labeling (aggregating) of the peripheral voxels distal to the digits and interdigits. The method section is too cursory to figure out how the p-values were calculated, or whether appropriate units for testing have been utilized; if the values are derived from voxels, not independent samples, the author is concerned that the p-values might be inflated due to

the breaking of independence. It might be due to these limitations that the authors failed to find meaningful digit-identity related genes except the toe as the individual digits comprise even fewer voxels (Extended Figure 6i). Such limitation of spatial expression pattern analysis is exacerbated when the authors try to match the affected bones from brachydactyly or syndactyly to the gene expression (Figure 2c). How would a reader for example deduce, based on the spatial pattern of COL11A2, SOX9, and FGFR3 from the VISIUM dataset, the color labeled affected bones? (line 635-638) The reviewer cannot. The spatial dataset might still have a value by extracting common spatial features (such as independent component analysis) and comparing the expression patterns each other, but the current presentation is tenuous at best.

In figure 5, the authors utilize the spatial dataset for even higher resolution, that is, within one voxel, when they spatially “resolve” “cell-cell” communication (Figure 5). It is rather puzzling the authors use the word “confirm” (line 525) when they describe the existence of cell-cell interaction by the presence of two gene expression found within one voxel, where more high resolution approaches (in situ hybridization at the least) would be available, given the position of two cell types within 100um circle would not confirm a cell-cell contact. The authors also use “overlap” (line 536) in describing the expression of FGFR2, FGF8 and FGF10, where the resolution cannot determine whether truly cells express them together just by spatial transcriptomics alone. Notwithstanding that it is absolutely difficult to read the overlap in the Figure 2d, and that an appropriate statistical quantification of the overlap is lacking, the rather low VISIUM resolution needs to be complemented by other means for validation. Even more, although the entire approach of identifying high confidence cell-cell communication reside in identifying cell “types” (and CellPhoneDB clearly outputs potential interactions based on clusters), the whole figure nor the manuscript fails to mention what “types” actually interact each other, and only talk about the molecules involved in terms of Notch signaling (Figure 5b), or FGF signaling (Figure 5c). The interaction between cell types is not well illustrated, despite useful pairwise representations available from CellPhoneDB.

As for the potential limitations of single cell transcriptomics, one issue is whether the capture rate is sufficient. The authors do acknowledge at the end of the discussion that they did not obtain hypertrophic chondrocytes in their scRNA-seq data despite the general abundance of cells of chondrocytic lineage and clear signals coming from the spatial transcriptomics (line 648-649). But in the mouse dataset they present, with the same 10X technology, clear Col10a1 expressing cells could be identified, such that it is hard to believe this absence is solely due to technical limitations (line 649-652). Rather, it would rather raise the question whether the cell numbers that they captured for each time point (total 114K cells) have sufficient breath to identify interesting cell populations. For example, in developing skeletal elements, tendon-bone attachment units display a characteristic Scx- and Sox9- double positive cell population important for the integration of tendons to bones (Blitz, Sharir, Akiyama & Zelzer, Development, 140, 2013). In neither mouse nor human datasets the authors present, could such rare populations be identified by using interactive browser.

To which extent the author’s dataset covers the cell type/state diversity has important implications beyond identifying rare cell type/states. Throughout their manuscript, they utilize the changes of cluster composition to infer lineage progression (figure 3c, line 368-373; figure 4c, extended figure 2), or species differences (figure 6d). If the cell capture rate is not saturating in terms of the resolution they used for clustering, the estimate of cluster composition will be widely variable. At

the minimum, presenting the reproducibility of cluster composition between samples from same time points would be helpful guide to assess this variability. Moreover, the assessment of the limits of resolution in terms of cell state diversity would also help the readers to understand how much remaining effort lies ahead for “cell-atlasing” (line 113) the developing (human) limb.

Another technical clarification necessary from the authors is how they settled to the computational parameters for all the analyses. Specifically, although they claim identification of 55 cell “types and states”, this depends on specific parameters, e.g. leiden resolution parameter, number of PCs incorporated etc. There needs to be a justification why the default parameters were used to determine cell types/states, otherwise, they should be called “clusters” not “cell types and states”. For example, one of the key findings of the research is the multiple distinct mesenchymal states (line 50-51, 347); at least a high resolution spatial localization of these clusters either on human specimen or corresponding more accessible murine model has to be made (e.g. in situ hybridization).

Lastly, the manuscript lacks the justification of annotations of these clusters. In the method section, only “manual annotation using known marker genes” were stated (line 800). It is imperative that the appropriate reference for the known marker genes be given for a proper peer review.

Specific minor comments

(Line 123) 55 cell types or states (consistency to line 143) from 114,047 (consistency to line 141)

(Line 159-160) The pattern emerged because of the manual annotation. More objective assessment of progenitor to differentiated cell state might be given by subdividing the clusters by fraction of cell cycle states.

(Line 175) Novel - human-specific feature, or previously unknown? Validation by in situ hybridization necessary to distinguish whether it is a compositional change or cell-specific change in expression.

(Line 185) There is no clear distinction between clusters and a formal statistics necessary to claim that chondrocytes are distinct. (Note that Interzone chondrocytes are labeled in the browser as interzone “cells”.)

(Line 188) Details of functional enrichment analysis in extended figure 3c is missing in the method section.

(Line 196) in-house

(Line 221) An additional

(Line 253) Extended Data Fig. 6a-e does not show previous in situ hybridization experiments in the mouse, like presented in Extended Data Figure 7.

(Line 406) Show FOXJ1 expression pattern in the limb

(Line 414) Figure 1d.

(Line 562) It is difficult to see what the 18 high-quality limb datasets are and need better labeling on x axis on Extended Figure 10a, and a separate supplemental table with the capture method, embryonic stages, number of samples and cells captured, and other basic statistics.

(Line 668) hindlimb stated followed by (line 669) forelimb. Contradiction.

(Line 759) reference for scrublet should be given.

(line 824) URL broken

(Line 858) Orthologs? More specific information how the orthologs were derived should be given.

(On interactive browser)

* LPM: LeidenID annotation do not match with the manuscript and figure 3 numbers.

* Annotations in the text do not match with annotations with the cell browser

- Interzone "cells", Interzone "chondrocytes"

- Intermediate chondrocytes, Maturing chondrocytes

Referee #2 (Remarks to the Author):

I read the manuscript "A human embryonic limb cell atlas resolved in space and time" by Zhang, He, Lawrence, Wang et al. with interest. Limb development is a complex process and a comprehensive understanding of this process in humans has not been studied. This work, in which authors combine both scRNA-seq and ST, will represent the first single-cell reference for the field. The abstract and introduction point out the relevance for this research, as well as the motivation behind, and they are clearly structured. Also, the methods are well explained. However, the manuscript lacks clarity in some parts and could be further improved if authors address the following points.

Doublet detection of human scRNA-seq data.

For double detection you state to be employing the approach posed in (Popescu, D. M. et al. Decoding human fetal liver haematopoiesis. Nature 574,(2019)). However, it is only described the first of the two steps proposed in that work. In the first step you overcluster scRNA-seq samples independently to score each cell with a corrected Pval (using Scrublet and correcting for FDR). Then you are filtering out cells based on that Pval instead of performing the second step were you perform a joint analysis of the samples of the same tissue to detect clusters significantly enriched with doubles and filter out all the cells in those clusters.

Since performing the filtering only based on the corrected Pval usually leads to a less restrictive filtering: Which has been your reasoning for avoiding the second step. Have you tried both and

observed no notable discrepancies? Or was this decision driven by biological hypothesis?

Data preprocessing and integration of human scRNA-seq data Vs Data preprocessing and integration of mouse scRNA-seq data.

Both preprocessing are similar, I figure that bbknn parameters have been chosen experimentally in each case to obtain a solid manifold integration. However it has drawn my attention that, in the highly variable gene selection, there is an additional step in the preprocessing of the mouse scRNA-seq data where genes with high dispersion scores that were barely co-expressed with other genes are regressed out. Is there any reason for not performing this step in the human samples, can you justify that this will not arise technical discrepancies when doing a cross-species comparison?

Clustering and annotation of human scRNA-seq data

Lines 793 & 794. What you want to say is that you have ran the UMAP on the NEIGHBORHOOD GRAPH computed using the 1903 highly variable genes. Parameters on the scanpy pp.neighbors function will be welcome and a clarification about whether PCA has been used or not DEGs computed with <https://github.com/ZhangHongbo-Lab/DEAPLOG> - DOES NOT EXIST

Deconvolution of human Visium data- cell2location

Some information about the final number of genes used in the deconvolution (shared genes between scRNA-seq and Visium after filtering lowly expressed genes) and an extended figure with the QC plot of expected vs observed distribution to evaluate the cell2location results quality. Was the deconvolution performed for every slide in the same run using the batch key covariance, or separately? What happened with the whole PCW 8.1 human hindlimb where you stitched 3 sections. Alignment and merging of multiple visium sections

Great idea. Did you consider applying any batch correction for the count matrices? Or you made sure no notable batch effect was distorting the results?

Cell-cell communication analysis of human scRNA-seq data.

I like the idea of running CellPhoneDB in scRNA-seq and look for the statistically significant interactions in the Visium sample instead of straight running CellPhoneDB in the Clustered Visium data.

Lines 843-845. Here it is stated that you use Visium data to validate interaction using spatial distance and expression patterns, but there is no method or test described to perform such validation. Which are the hypotheses that a LR pair should fulfill in terms of spatial features to be validated?

Referee #3 (Remarks to the Author):

In this paper Zhang et al. aim to generate a detailed spatiotemporal atlas of human limb development using single cell transcriptomics and spatial transcriptomics. To this end they performed single cell transcriptomic sequencing of human embryonic hindlimbs from PCW5 to PCW9 complemented with spatial transcriptomic of four time points. With the analysis of the dataset generated the authors identify cell states, spatially locate them and extract conclusions on functional gene expression. Finally, they also address the homology with the mouse, as privileged model organism, though an integrated transcriptomic study.

This is a very ambitious work and I acknowledge the authors' efforts towards a detailed atlas. However, the lack of early stages in the analyses and the low resolution level of the spatial transcriptomics are important limitations of the study.

Specific comments:

1. One concern is the stage of the human samples analyzed. I agree with the authors on the difficulty of obtaining embryos in the earlier phases of limb development. The earlier stage used here (PCW5.1) seems to be equivalent to mouse E12, when patterning events are very advanced and practically all the skeletal elements except the last phalanges have already been laid down. This implies that the data obtained here cannot inform on the human nascent limb bud but only on later patterning and differentiation events. This should be acknowledged in the paper and the limb bud schematics adapted to the real shape of the limbs (i.e. Fig. 1a, extended fig. 6..).
2. It would be informative and visual to show the fraction of cell-types (Fig. 1b) per time point. Are the PZ states restricted to the earlier stages analyzed? During limb development, the PZ is understood as the region of distal mesodermal cells under AER influence (Tabin and Wolpert 2007 PMID: 17575045). However, here the PZ is applied to practically the whole autopod (Fig. 1c; separated in two regions the peripheral/outer region of more naïve mesoderm and the internal/transitional region of differentiated mesoderm already split in digit and interdigit regions). I don't think the term PZ should be used here.
3. The level of resolution of the Visium spatial transcriptomics is very low and, on many occasions, the spatial location remains vague. Indeed, without knowing the pattern of expression in model organism this study would have been very little informative. In general additional description/explanation would be desirable; for example, is the population of dermal fibroblast only present at proximal level as seen in Extended Fig. 5e?
4. It is unclear whether the AER cell state has been captured- In the limb ectoderm, the gene expression characteristic of the AER is *Fgf8* (also the other AER-FGFs, depending on the stage). However, only a small fraction of AER-basal cells seems to express *Fgf8* (extended Fig. 2). The expression of *SP8* and *WNT6* does not define the AER as many non-AER cells do express these two genes in the limb bud ectoderm. Fig. 5b shows the spatial location of *Fgf8* restricted to a couple of voxels in the distal ectoderm of PCW6.2. Maybe *Fgf8* is already downregulated in the AER at this stage?
5. The cross-reference between the set of DEG between digit and interdigit and the list of 2300 genes with single gene conditions seems too simplistic. How many genes were intersected? It should be noted that the link of the expression pattern to the digits (i.e. *IHH*, *NOG* etc.) or interdigits is not sufficient to explain the phenotype (i.e. only some elements affected etc..). Many of these genes have multiple functions during limb development. For example, *Dlx5* is expressed both in the AER/ectoderm and in the interdigits and their LOF impacts both digit patterning (split phenotype) and interdigit (syndactyly) but the correlation with the different domains of expression is unclear. The novelty of this analysis is the use of the human limb bud. Given the dynamism of expression patterns during limb development, the mouse stages in Extended data Fig. 7 are in general not comparable to human PCW6.2.
6. The spatially resolved heatmaps in Fig. 5 are difficult to understand. How many cells may be in a voxel? Shouldn't the spatial expression of receptor and ligand (*Notch1/Jag1*) be in the same voxel? Studies on model organisms and human pathology indicate that FGFR specificity depends on their isoforms something not possible to analyze in the FGF/FGFR spatially resolved heatmaps.

7. Among the novel findings are the expression of Pitx1 in the FL proximal muscle (Fig. 4f), the expression of Fgf8 in proximal limb (Fig. 5d), and the distinct trajectories of embryonic and fetal myocytes. It would be helpful if the authors explore the human specificity of these features in the mouse.

8. The study of the homology and divergence between human and mouse requires further explanation. The stages of the mouse samples should be indicated, the specific stages and type of dissection used in the mouse/human comparison in Fig. 6e should also be indicated.

Minor comments:

- Fig. 1a seems to indicate that the limbs analyzed were FLs, rather than HLs as indicated throughout the manuscript

- In Fig. 6f I don't detect any difference with mouse in the pattern of expression of HOXA11, HOXA11AS, HOXA13 as the authors seem to indicate in the text.

- Line 99- "throughout the remainder of the first trimester in humans..." please adapt this sentence since to this point there was no time reference

- Lane 197- "...to analyze the entire lower limb..." should be the entire PD axis of the lower limb- Still only one section in the dorso-ventral axis is analyzed.

Methods:

- Please, explain the rationale for the P/M/D or P/D dissection of the limbs and how this information was used in the analysis.

- Different software versions of Single-Cell Software Suite were used for human (v.2.1.1) and mouse (v.3.0.2). Equally for Python package Scanpy, v.1.6.0 for human and v.1.3.11 mouse. Has this difference been accounted for?

- Why was a threshold of 0.1 imposed for doublet analysis? How many doublets were removed?

- Figure 6e: Please, indicate the scale of the size of the squares and circles and the color legend

- Extended Figure 10 a: There is no legend for the x axis, which sample correspond to each violin plot?

Author Rebuttals to Initial Comments:

Referee #1 (Remarks to the Author):

This manuscript is one of the on-going efforts from the Human Developmental Cell Atlas community to provide vital information on human development. Compared to the already published efforts (ref. 11-15), the manuscript might be described as being the most ambitious. The study scope spans early patterning of the limb, lineage diversification of lateral plate mesoderm derived limb progenitors, as well as skeletal myogenesis, and exploration of species-specific differences, each would normally warrant an independent paper. As spatial and temporal context is particularly important in the developing limb, with limb progenitors undergoing similar molecular differentiation processes throughout the limb across space and time to yield distinct functional structures, their transcriptomic data is supplemented by gross spatial annotation by dissection along the proximo-distal axis, further supplemented by spatial transcriptomic approach, with 10 temporal points in four weeks time window, a distinct feature to the other already published human limb development dataset (Xi et al. 2020, ref. 18). Moreover, to discern what is known and what is unknown, the authors conduct a meta-analysis, aggregating existing transcriptomic datasets of the murine model, supplemented their own and tried to compare them to their human dataset. Lastly, they provide a standardized user-friendly interactive browser platform for maximum accessibility. While this ambitious study has the potential to make a significant contribution to the field, in its current form the approaches are not well enough integrated, and there are too many critical flaws, for it to achieve this goal.

Foremost, their results are not really well grounded with the current conceptual framework of the limb development, using terms that are used out of the context, or relying on outdated terms, obfuscating what is known and unknown in the field.

An example of misunderstanding of key terms is “posterior prevalence” (line 265-279), which the authors use in describing the spatial expression pattern of HOX genes. Posterior prevalence is a general property attributed to vertebrate HOX proteins describing the dominant effect of more posterior HOX proteins over the function of anterior orthologs in common (overlapping) areas of expression, a term coined by Denis Duboule in 1991 (D. Duboule, Patterning in the vertebrate limb, *Curr. Opin. Genet. Dev.*, 1:211-216) which has nothing to do with their spatial colinearity, or domains of gene expression as being used in the manuscript (e.g. line 311-312).

We appreciate the reviewer’s comment. We apologise for the confusion and have reworded to: “The first wave occurs in the nascent limb bud (prior to the period captured in our samples) in an asymmetrical fashion, with expression progressively restricted to its posterior margin with increasing 5’ position.” (line 257-259)

Another example can be found when the authors cite reference 156 (Feregino & Tschopp, *Dev. Dyn.* 2021) with “evolutionary turnover”, which has nothing to do with the pace of developmental maturation at all, but rather is about the degree of conservation of gene expression modules (line 577-579) over evolutionary time and would be irrelevant in the context of finding more number of immune and epithelial cell type capture in the mouse dataset. We appreciate the reviewer’s comment and have now removed this phrase from the manuscript.

One of the most critical usage of an outdated term is the “progress zone” (line 172-173, followed by numerous usage throughout the manuscript), relating to a distal region defined in a now discredited model of proximodistal limb patterning proposed by Lewis Wolpert and colleagues prior to the advent of molecular genetic analyses. The term should not be used. The current view of PD specification is concisely summarized in an excellent review by Caitlin McQueen and Matt Towers. Indeed, the reviewer would urge the authors to read the entire review carefully, and interpret their data in the context of the information they provide. (C. McQueen and M. Towers Establishing the pattern of the vertebrate limb Development 147 (2020)).

We appreciate the reviewer’s comment and the excellent review by Caitlin McQueen and Matt Towers. We have examined our data in the context of the models highlighted by the authors. One challenge on this front is the fact that even our earliest sampling timepoints for both single cell and spatial analysis are at a stage when limb development is quite advanced in the context of key signaling events, with the domains for the stylopod, zeugopod and autopod already established (Rebuttal Figure 1b). This is an unavoidable consequence of using human tissue collected from termination of pregnancy. In addition, our single cell and spatial transcriptomic experiments capture very few FGF8+ve cells. As the AER is a critical component of these models (particularly the signal-time model), this again makes comment on global mechanisms of limb development a challenge. Whilst there is a trend of gene expression supporting the two signal model (Rebuttal Figure 1c), we are again cautious in our interpretation of this due to limitations in the time points sampled and the dissection at PCW5.6 which does not include the section of the trunk from which the limb emerges. We therefore at this stage do not plan to include this in the manuscript. We have rephrased the manuscript to remove references to the progress zone.

Rebuttal figure 1. The different models of limb development.

- (A) Diagrams summarising three popular models of limb development
- (B) Spatial heatmaps showing the three domains of the limb are established at pcw5.6 and 6.2, as marked by MEIS2 (red), HOXA11 (orange) and HOXA13 (yellow)
- (C) Spatial heatmaps showing the combined expression of genes associated with retinoic acid synthesis (RA; RDH10 & ALDH1A2 - pink), RA uptake (STRA6, CRABP1 - purple) and RA degradation (CYP26A1, B1 - blue).

Perhaps due to such weak grounding, coupled by a largely uncritical attitude towards the output of computational methods, some demonstrably incorrect presentations were included in some of the figures.

How the authors depict chondrocyte differentiation in figure 3 is a useful illustration. The established consensus of chondrocyte differentiation is that after mesenchymal condensation (line 365), resting zone chondrocytes differentiate into proliferating, prehypertrophic, then hypertrophic chondrocytes in a sequential manner (reviewed in Kozhemyakina, Lassar and Zelzer A pathway to bone: signaling molecules and transcription factors involved in chondrocyte development and maturation, Development (2015)). In the manuscript, the authors use the term “chondroblast” (line 402) that is rarely used in the context of limb development. Nor is the term “maturing chondrocytes” (note that in their interactive browser for LPM, it is annotated as ‘intermediate chondrocyte’ which is also not at all used in the field) used in the field frequently, and if so, includes the maturation process from resting to prehypertrophic, and hypertrophic chondrocytes. None of the presentations (Extended figure 1, Figure 3b, Figure 3d) show clear markers that define and distinguish chondroblast from “maturing chondrocytes”, except that the authors mention “immature” for chondroblasts (line 392) so one has to guess what these two entities represent and connected to the existing model. In figure 3a, the direction of cell differentiation is

depicted as “chondroblasts” differentiating to resting chondrocytes in one branch, and the other differentiating into “maturing chondrocytes” to “prehypertrophic chondrocytes”. Such bifurcating differentiation process separates, and in a sense reverses the direction between resting zone chondrocytes and prehypertrophic chondrocyte states, which contradicts the consensus model, calling for a careful re-examination of the validity of their computational approach of lineage reconstruction (diffusion map, PAGA and force directed embedding). Such a contradiction might have arisen from the author’s ambitious attempt to aggregate all developmental processes from PCW5.1 to PCW9.3 together, which collapses two very distinct processes of mesenchymal condensation, formation of the skeletal elements with the epiphyseal growth plate development. In this context, stage, region-specific analysis, overlaid together, might better validate and elucidate the various differentiation processes.

We appreciate the reviewer’s comment and we have now renamed the “chondroblast” cell type to “chondrocyte progenitor” and explicitly stated the characteristics of its transcriptome – that is *SOX9* expression but much weaker *COL2A1* expression than all other chondrocytes (line 376). These genes have been added to the dotplot to aid the reader in discerning these differences. “maturing chondrocytes” have been renamed as “proliferating chondrocytes”(line 378). We also show these cells to have higher fraction of cells in G2/M/S phase of the cell cycle compared to other cell types, and to lack marker gene expression of other chondrocyte subtypes such as UCMA (resting) or IHH(prehypertrophic). We have performed RNA velocity analysis (see methods) on a single sample from PCW7.2 to further demonstrate predicted differentiation trajectory, which remains curious in nature as you point out. We have added a sentence to the text to highlight that this should be interpreted cautiously and requires further investigation / validation (line 382-386; Extended figure 7d).

Another example of a concerning presentation is found in figure 4. The authors present PITX1 expressed in forelimb mesenchyme but not in hindlimb mesenchyme (Figure 4e). Pitx1 has been found exclusively in hindlimb in many vertebrate animals (c.f. Logan and Tabin, Role of Pitx1 Upstream of Tbx4 in specification of hindlimb identity, Science 283, 1999; DeLaurier, Schweitzer and Logan, Pitx1 determines the morphology of muscle, tendon, and bones of the hindlimb, Dev Biol. 2006). Therefore, the mesenchymal expression data in itself is a very surprising result but the authors only treat this finding in a passing comment (line 494-495) in the context of muscle cell expression. Even internally, the presentation seems to be inconsistent. How do the authors reconcile the presentation in Figure 4e to even the immunohistochemistry in Figure 4f, where they display abundant PITX1 immunofluorescence in the hindlimb that are PAX3 negative? Also that almost all PITX1 immunofluorescence signals in the PCW5.5 forelimb specimen are derived from PAX3+ cells?

We are very sorry for the label swap within the legend of forelimb and hindlimb and we have already corrected this. The PITX1 expressed in hindlimb mesenchyme but not in forelimb mesenchyme and both in forelimb and hindlimb skeletal muscle.

Overall, the manuscript would benefit from more cross-examination between mouse datasets and human datasets, rather than positioning the comparative aspect at the end of the manuscript, as it reads more of an afterthought. For example, conducting a similar lineage progression analysis on the mouse dataset (e.g. for the chondrocyte lineage) and validating the finding with the existing literature would help to justify the usage of their pipeline to the human data, and also better highlight the human-specific characteristics.

We appreciate the reviewer’s comment and are sorry this section felt like an afterthought. Our reason for not making this aspect central to the manuscript is the

inherent difficulty with comparing multiple datasets generated by different groups. For this reason, we chose to present this section separately, rather than framing all of our work around it. We have now however performed more cross-examination between mouse and human datasets in order to enrich this section, including the comparison of lineage progression as suggested. We have now performed trajectory analysis of mouse muscle and mesenchymal lineages and choose to highlight the human-enriched features of the PAX3+PAX7+ population and the differing heterogeneity of the PAX7+ pool between species. Analysis of the mouse mesenchyme produced a broadly similar trajectory, with highly similar expression of marker genes (See Extended Data Figures 8 & 10).

In addition to these issues, the authors fail to clearly outline the limitations of their methodological approaches.

We appreciate the reviewer's comments and we have acknowledged the limitations of methods in the text with two further paragraphs discussing resolution limitations of visium and expanded discussion of the reasons for so few hypertrophic chondrocytes and scx/sox9 double-positive cells, centred around sampling breadth (lines 653-685). Indeed, we thank the reviewer for highlighting these cell states to us (see below)

As for the spatial transcriptomics, the reviewer notes that the 10X VISIUM methodology has a severe limitation in terms of spatial resolution (center to center distance is 100um, worse than naked eye resolution of ~40um - cf. Cho et al. Microscopic examination of spatial transcriptome using Seq-Scope, Cell, 2021). The authors use computational methods to deconvolute the transcriptome into cell types, but this does not necessarily inform the readers how many cells (not cell types) might have been captured in one voxel. Based on the interactive browser, the number of deconvolved clusters appear to have an upper range between 6 and 9 different cell types/states. Thus, while useful in discerning transcriptomes at a large distance, as the authors demonstrate with the perineural fibroblasts associated with nerves, ADH1B(ADH+FiB) fibroblasts, associated with muscle cells, and the perimysium cells separated from the tenocytes (line 238-241), it becomes more tenuous as the distance in question decreases.

We thank the reviewer for this insight. As above, this has been discussed in a further limitations paragraph in the discussion. We also performed a large amount of new ISH (RNAscope) and immunofluorescence experiments to validate Visium results with fine tissue structure and gene expression location (see Fig. 2, 5; Extended data Fig. 2, 4 and 7; and Supplementary Video 1)

The resolution limit is almost reached when they describe the PCW6.2 human digits (Figure 2), where the distance between interdigital space and digits appears to be either one or at most two voxels. As all analyses were performed with manual labeling of voxels based on H&E sections, at the very least it is important to have a sensitivity analysis of inclusion/exclusion of voxels in terms of the results presented in the manuscript. As an example, some of the genes that the authors highlighted also in the text, such as CYP26B1, IRX1 appear in violin plot barely differentially expressed albeit with significant p-values. Conversely, most of the genes except BMP7 really do not have strong interdigital structure observed based on the spatial representation (e.g. CCL2), and most likely derive the differential expression result because of the labeling (aggregating) of the peripheral voxels distal to the digits and interdigits. The method section is too cursory to figure out how the p-values were calculated, or whether

appropriate units for testing have been utilized; if the values are derived from voxels, not independent samples, the author is concerned that the p-values might be inflated due to the breaking of independence. It might be due to these limitations that the authors failed to find meaningful digit-identity related genes except the toe as the individual digits comprise even fewer voxels (Extended Figure 6i). Such limitation of spatial expression pattern analysis is exacerbated when the authors try to match the affected bones from brachydactyly or syndactyly to the gene expression (Figure 2c). How would a reader for example deduce, based on the spatial pattern of COL11A2, SOX9, and FGFR3 from the VISIUM dataset, the color labeled affected bones? (line 635-638) The reviewer cannot. The spatial dataset might still have a value by extracting common spatial features (such as independent component analysis) and comparing the expression patterns each other, but the current presentation is tenuous at best.

Thank you. We have expanded the methodology section to confirm that each group of voxels is treated as one entity for statistical testing, and have described in detail the statistical test used for this analysis. We have performed unsupervised leiden clustering on the foot plate rather than manual annotation to annotate the digital, interdigital and distal periphery voxels, adjusting the figure (and genes found to vary significantly) accordingly to give more detail. We have added discussion of the failure to find digit-specific genes into the paragraph discussing visium resolution limits in the discussion (lines 685-689).

Regarding the illustrations of affected bones and patterns of syndactyly, we apologise for the confusion caused. The intentions of these drawings was to inform those not familiar with such conditions of their adult phenotype, rather than attempt to draw a precise link between expression pattern and phenotype, which is not feasible at this resolution and with a single time point. We have removed the drawings from the manuscript to avoid this potential confusion.

In figure 5, the authors utilize the spatial dataset for even higher resolution, that is, within one voxel, when they spatially “resolve” “cell-cell” communication (Figure 5). It is rather puzzling the authors use the word “confirm” (line 525) when they describe the existence of cell-cell interaction by the presence of two gene expression found within one voxel, where more high resolution approaches (in situ hybridization at the least) would be available, given the position of two cell types within 100um circle would not confirm a cell-cell contact. The authors also use “overlap” (line 536) in describing the expression of FGFR2, FGF8 and FGF10, where the resolution cannot determine whether truly cells express them together just by spatial transcriptomics alone. Notwithstanding that it is absolutely difficult to read the overlap in the Figure 2d, and that an appropriate statistical quantification of the overlap is lacking, the rather low VISIUM resolution needs to be complemented by other means for validation. Even more, although the entire approach of identifying high confidence cell-cell communication reside in identifying cell “types” (and CellphoneDB clearly outputs potential interactions based on clusters), the whole figure nor the manuscript fails to mention what “types” actually interact each other, and only talk about the molecules involved in terms of Notch signaling (Figure 5b), or FGF signaling (Figure 5c). The interaction between cell types is not well illustrated, despite useful pairwise representations available from CellPhoneDB.

We appreciate the reviewer’s comment. We have now used ISH to further explore this cell-cell interaction of JAG1/Notch1, wnt/fzd and the FGF family members discussed in the early limb. (See Fig. 5). We have placed these interactions in the cell-type context as per your suggestion [panels a, h] and made this clear in the text (page 12).

As for the potential limitations of single cell transcriptomics, one issue is whether the capture rate is sufficient. The authors do acknowledge at the end of the discussion that they did not obtain hypertrophic chondrocytes in their scRNA-seq data despite the general abundance of cells of chondrocytic lineage and clear signals coming from the spatial transcriptomics (line 648-649). But in the mouse dataset they present, with the same 10X technology, clear Col10a1 expressing cells could be identified, such that it is hard to believe this absence is solely due to technical limitations (line 649-652). Rather, it would rather raise the question whether the cell numbers that they captured for each time point (total 114K cells) have sufficient breadth to identify interesting cell populations. For example, in developing skeletal elements, tendon-bone attachment units display a characteristic Scx- and Sox9- double positive cell population important for the integration of tendons to bones (Blitz, Sharir, Akiyama & Zelzer, Development, 140, 2013). In neither mouse nor human datasets the authors present, could such rare populations be identified by using interactive browser.

We thank the reviewer for these helpful comments. In our human scRNA-seq data, we have sub-clustered the hypertrophic chondrocytes to reveal a small number of COL10+ve hypertrophic chondrocytes. We have included this new annotation in the dotplots, and also re-run cell2location to show their predicted location (see Fig. 1, Extended Data Fig. 1 and Rebuttal Fig. 2 below). Similarly, we have identified SCX/SOX9 double positive cells and mapped them onto samples to show their predicted location at anatomically appropriate sites (lines 420-423). We have also performed RNA-ISH using probes for these two molecules to demonstrate their co-localization in the forming digits - likely representing the entheses of long flexor tendons of the foot (Extended Data Fig. 7). We have expanded the discussion of limitations to include sampling breadth, in addition to digestion/ permeabilisation differences.

Rebuttal Figure 2. The hypertrophic Chondrocyte and its marker genes.

- (A) Uniform manifold approximation and projection (UMAP) plot showing the cell distribution of prehypertrophic chondrocyte (PrehyperChon) and hypertrophic chondrocyte (HyperChon).
- (B) UMAP plot showing the expression level of marker genes of HyperChon.
- (C) Spatial heatmaps showing the expression level of marker genes of HyperChon

To which extent the author's dataset covers the cell type/state diversity has important

implications beyond identifying rare cell type/states. Throughout their manuscript, they utilize the changes of cluster composition to infer lineage progression (figure 3c, line 368-373; figure 4c, extended figure 2), or species differences (figure 6d). If the cell capture rate is not saturating in terms of the resolution they used for clustering, the estimate of cluster composition will be widely variable. At the minimum, presenting the reproducibility of cluster composition between samples from same time points would be helpful guide to assess this variability. Moreover, the assessment of the limits of resolution in terms of cell state diversity would also help the readers to understand how much remaining effort lies ahead for “cell-atlasing” (line 113) the developing (human) limb.

Thank you again for this point. To show cluster composition reproducibility, we have now presented individual sample/replicate using bar plots (Fig. 6d and Extended Data Fig. 10d). To assess how much “cell-atlasing” is needed, we’ve used the immune cells that have been intensively characterized as a reference and performed supervised label prediction using CellTypist. (Domínguez Conde et al. 2022) It discovered multiple rarer immune cell types (such as 12 DC1 cells and 6 neutrophils) present in our human limb dataset supported by marker genes (See Rebuttal Figure 3 below). Due to the low cell number we are not able to annotate them using unbiased clustering and conventional pipelines. Whilst we cannot extrapolate this to all cell types, this serves as an example of resolution that may be gained in the future as tissues continued to be studied in increasing detail.

Rebuttal Figure 3: Undersampled cell types and states in the immune compartment. A: Current annotation labels (left) compared to labels predicted individually by Celltypist. B: Marker genes of DC1 and Neutrophil cells. Dot sizes represent fraction of cells expressing the gene and colors represent relative gene expression level.

In the manuscript, we have again expanded the discussion to talk about capture limitations (breadth- as previously mentioned) and point out its importance for future endeavours.

Another technical clarification necessary from the authors is how they settled to the

computational parameters for all the analyses. Specifically, although they claim identification of 55 cell “types and states”, this depends on specific parameters, e.g. leiden resolution parameter, number of PCs incorporated etc. There needs to be a justification why the default parameters were used to determine cell types/states, otherwise, they should be called “clusters” not “cell types and states”. For example, one of the key findings of the research is the multiple distinct mesenchymal states (line 50-51, 347); at least a high resolution spatial localization of these clusters either on human specimen or corresponding more accessible murine model has to be made (e.g. in situ hybridization).

We appreciate the reviewer’s comment and we have added more detailed parameter description for cell clustering in the methods section. We now refer to groups of cells as clusters for clarity. Additionally, we used ISH to confirm the heterogeneity of mesenchymal cells and more clearly plot gene expression by cluster (See Extended Data Fig. 4).

Lastly, the manuscript lacks the justification of annotations of these clusters. In the method section, only “manual annotation using known marker genes” were stated (line 800). It is imperative that the appropriate reference for the known marker genes be given for a proper peer review.

We have added a table of marker genes for each cell cluster together with references as Extended Data Table 1.

Specific minor comments (Line 123) 55 cell types or states (consistency to line 143) from 114,047 (consistency to line 141)

Thank you - corrected.

(Line 159-160) The pattern emerged because of the manual annotation. More objective assessment of progenitor to differentiated cell state might be given by subdividing the clusters by fraction of cell cycle states.

We appreciate the reviewer’s comment and we have added the cell-cycle phase distributions across time and cell type into the Extended Data Figure 7

(Line 175) Novel - human-specific feature, or previously unknown? Validation by in situ hybridization necessary to distinguish whether it is a compositional change or cell-specific change in expression.

We appreciate the reviewer’s comment and we have now used ISH, in addition to three-dimensional light sheet fluorescence microscopy of Sox9, LHX2 and MSX1 to confirm the compositional change. The corresponding results have been added to the Extended Data Figure 4 and Supplementary Video 1.

(Line 185) There is no clear distinction between clusters and a formal statistics necessary to claim that chondrocytes are distinct. (Note that Interzone chondrocytes are labeled in the browser as interzone “cells”.)
We appreciate the reviewer’s comment and we have performed the differential

expression analysis for all chondrocyte clusters and found that these clusters are indeed distinct in gene expression patterns (PRG4; logFC=5.15, $p=4.1E-26$). We have added statistics to the text (line 410) to make this clear to the reader and corrected our browser; the marker genes for clusters can also be found in Extended Data Table 1.

(Line 188) Details of functional enrichment analysis in extended figure 3c is missing in the method section.

Apologies for this - we have now removed this analysis and focused on the increased detail we now have on the mesenchymal heterogeneity in the limb together with ISH staining.

(Line 196) in-house
Now we have corrected the word.

(Line 221) An additional
Now we have corrected the word.

(Line 253) Extended Data Fig. 6a-e does not show previous in situ hybridization experiments in the mouse, like presented in Extended Data Figure 7.

Apologies - this sentence is badly worded. We have re-written so that it is clear Extended Data Figure 6 is displaying human expression *in keeping* with classical expression in model organisms, rather than showing murine expression patterns (lines 246-247)

(Line 406) Show FOXJ1 expression pattern in the limb.

Now we have added the heatmap of FOXJ1 to show the expression level.

(Line 414) Figure 1d.
Now we have added it in the text.

(Line 562) It is difficult to see what the 18 high-quality limb datasets are and need better labeling on x axis on Extended Figure 10a, and a separate supplemental table with the capture method, embryonic stages, number of samples and cells captured, and other basic statistics.

We apologise for this and now we have added labels and supplementary table.

(Line 668) hindlimb stated followed by (line 669) forelimb. Contradiction.

Thank you - corrected.

(Line 759) reference for scrublet should be given.

Thank you - updated the reference of scrublet.

(line 824) URL broken

apologies- we have corrected it.

(Line 858) Orthologs? More specific information how the orthologs were derived should be given.

Thank you for highlighting this. We have updated the methods section to show:

“Mouse orthologs were first “translated” to human genes using MGI homology database (<https://www.informatics.jax.org/homology.shtml>). Processed human and mouse data were then merged together using outer join of all the genes. The matched dataset was then integrated by MultiMAP using the MultiMAP_Integration() function, using separately pre-calculated PCs and the union set of previously calculated mouse and human feature genes (including both orthologs and non-orthologs) to maximise biological variance” (lines 895-906)

(On interactive browser)

* LPM: LeidenID annotation do not match with the manuscript and figure 3 numbers.

Apologies - we have updated the browser

* Annotations in the text do not match with annotations with the cell browser
- Interzone “cells”, Interzone “chondrocytes”
- Intermediate chondrocytes, Maturing chondrocytes
We apologise for these errors and we have now corrected these terms.

Referee #2 (Remarks to the Author):

I read the manuscript “A human embryonic limb cell atlas resolved in space and time” by Zhang, He, Lawrence, Wang et al. with interest. Limb development is a complex process and a comprehensive understanding of this process in humans has not been studied. This work, in which authors combine both scRNA-seq and ST, will represent the first single-cell reference for the field. The abstract and introduction point out the relevance for this research, as well as the motivation behind, and they are clearly structured. Also, the methods are well explained. However, the manuscript lacks clarity in some parts and could be further improved if authors address the following points.

Doublet detection of human scRNA-seq data. For double detection you state to be employing the approach posed in (Popescu, D. M. et al. Decoding human fetal liver haematopoiesis. Nature 574,(2019)). However, it is only described the first of the two steps proposed in that work. In the first step you overcluster scRNA-seq samples independently to score each cell with a corrected Pval (using Scrublet and correcting for FDR). Then you are filtering out cells based on that Pval instead of performing the second step were you perform a joint analysis of the samples of the same tissue to detect clusters significantly enriched with doubles and filter out all the cells in those clusters. Since performing the filtering only based on the corrected Pval usually leads to a less restrictive filtering: Which has been your reasoning for avoiding the second step. Have you tried both and observed no notable discrepancies? Or was this decision driven by biological hypothesis?

We thank the reviewer for pointing out this suboptimal approach/ lack of clarity. We now remove the doublets based on clusters derived from cells in all the samples after batch correction, guided by both the final scores from this published approach and sense checked by marker gene expression based on literature. We have updated the methods section to clarify our approach. This combined approach removed 1535 doublet barcodes in human data and 958 in mouse data (lines 789-804).

Data preprocessing and integration of human scRNA-seq data Vs Data preprocessing and integration of mouse scRNA-seq data. Both preprocessing are similar, I figure that bbknn parameters have been chosen experimentally in each case to obtain a solid manifold integration. However it has drawn my attention that, in the highly variable gene selection, there is an additional step in the preprocessing of the mouse scRNA-seq data where genes with high dispersion scores that were barely co-expressed with other genes are regressed out. Is there any reason for not performing this step in the human samples, can you justify that this will not arise technical discrepancies when doing a cross-species comparison?

Thank you for highlighting this discrepancy. We have now unified our processing methods for human and mouse data so as to try and eliminate any computational differences. Please see methods section updates- all single cell sections are now unified, rather than given by species, eg "Alignment, quantification and quality control of scRNA-seq data", "Data preprocessing and integration of scRNA-seq data" (lines 772-781 and lines 807-819)..

Clustering and annotation of human scRNA-seq data Lines 793 & 794. What you want to say is that you have ran the UMAP on the NEIGHBORHOOD GRAPH computed using the 1903 highly variable genes. Parameters on the scanpy pp.neighbors function will be welcome and a clarification about whether PCA has been used or not We are very sorry for the imprecise description and now have rewritten this section (lines 821-828).

DEGs computed with <https://github.com/ZhangHongbo-Lab/DEAPLOG> [github.com] - DOES NOT EXISTS

We are very sorry for the error and now we have fixed it.

Deconvolution of human Visium data- cell2location

Some information about the final number of genes used in the deconvolution (shared genes between scRNA-seq and Visium after filtering lowly expressed genes) and an extended figure with the QC plot of expected vs observed distribution to evaluate the cell2location results quality.

13,763 genes remained after default filtering [added to methods- line 834]

We have added QC plots from the c2l output to Extended data Fig 1

Was the deconvolution performed for every slide in the same run using the batch key covariance, or separately?

It was performed on all slides simultaneously, using slide as batch key [information added to methods- lines 836-837]

What happened with the whole PCW 8.1 human hindlimb where you stitched 3 sections. Alignment and merging of multiple visium sections Great idea. Did you consider applying any batch correction for the count matrices? Or you made sure no notable batch effect was distorting the results?

Thank you - we manually checked for batch effects by inspecting overlapping regions between different slides and ensuring cell abundancy calls were comparable. This was not performed using a particular statistical test, but rather ensuring the dominant cell fraction remained the same. The overlapping regions included multiple tissues including nerve, muscle, osteochondral tissue, tendon and others, so we are confident that there was no appreciable batch effect.

Cell-cell communication analysis of human scRNA-seq data. I like the idea of running CellPhoneDB in scRNA-seq and look for the statistically significant interactions in the Visium sample instead of straight running CellPhoneDB in the Clustered Visium data. Lines 843-845. Here it is stated that you use Visium data to validate interaction using spatial distance and expression patterns, but there is no method or test described to perform such validation. Which are the hypotheses that a LR pair should fulfill in terms of spatial features to be validated?

Thank you for raising this issue. We have now performed RNA ISH on predicted interacting partners to show coexpression in adjacent cells, rather than being limited to the 50um resolution of visium, which in itself cannot serve as a validation. For JAG-NOTCH (Fig 5a, f, g) we expect adjacent cells to be directly interacting as per the known biology of this signaling axis, which is supported by the very strong co-location on ISH staining. For FGF molecules (Fig. 5h-j, they classically act in a diffusion gradient manner. Thus our findings, which we have validated with RNA ISH, are in keeping with this- in particular FGF8 being restricted to the AER and FGF10 and FGFR2 not necessarily being co-located, but with the former strongly expressed as expected in the distal mesenchyme.

Referee #3 (Remarks to the Author):

In this paper Zhang et al. aim to generate a detailed spatiotemporal atlas of human limb development using single cell transcriptomics and spatial transcriptomics. To this end they performed single cell transcriptomic sequencing of human embryonic hindlimbs from PCW5 to PCW9 complemented with spatial transcriptomic of four time points. With the analysis of the dataset generated the authors identify cell states, spatially locate them and extract conclusions on functional gene expression. Finally, they also address the homology with the mouse, as privileged model organism, though an integrated transcriptomic study.

This is a very ambitious work and I acknowledge the authors' efforts towards a detailed atlas. However, the lack of early stages in the analyses and the low resolution level of the spatial transcriptomics are important limitations of the study.

Thank you. Regarding resolution, we have added a limitations paragraph in the discussion to emphasise the limitations posed by 50 micron resolution (lines 653-689) and performed additional experiments (RNA-ISH, light sheet fluorescence microscopy) to improve the resolution for specific points raised in the manuscript (see Fig. 2, 5; Extended data Fig. 2, 4 and 7; and Supplementary Video 1).

Specific

comments:

1. One concern is the stage of the human samples analyzed. I agree with the authors on the difficulty of obtaining embryos in the earlier phases of limb development. The earlier stage used here (PCW5.1) seems to be equivalent to mouse E12, when patterning events are very advanced and practically all the skeletal elements except the last phalanges have already been laid down. This implies that the data obtained here cannot inform on the human nascent limb bud but only on later patterning and differentiation events. This should be acknowledged in the paper and the limb bud schematics adapted to the real shape of the limbs (i.e. Fig. 1a, extended fig. 6..).

We appreciate the reviewer's comments. We have re-framed the manuscript to make it clear that we are dealing with immature or developing limbs rather than the nascent limb bud, and highlighted this in the discussion (lines 637-641). This includes adapting the figures to make this distinction clearer. In addition, we have made it clear in the species comparison section that the mouse data (starting at E9.5) contains distinct populations of cells (greater Pax3⁺ satellite cells; distinct mesenchymal populations) as a result of this sampling difference (lines 553-559).

2. It would be informative and visual to show the fraction of cell-types (Fig. 1b) per time point. Are the PZC states restricted to the earlier stages analyzed? During limb development, the PZ is understood as the region of distal mesodermal cells under AER influence (Tabin and Wolpert 2007 PMID: 17575045). However, here the PZ is applied to practically the whole autopod (Fig. 1c; separated in two regions the peripheral/outer region of more naïve mesoderm and the internal/transitional region of differentiated mesoderm already split in digit and interdigit regions). I don't think the term PZ should be used here.

We appreciate the reviewer's comment and this data can be seen in extended data figure 2. We have re-framed the manuscript to not discuss the progress zone but rather anatomically segregated mesenchymal states. We have also performed RNA-ISH and LSM experiments to validate these states at higher resolution (see Supplementary Video 1 and Extended Data Fig. 4)

3. The level of resolution of the Visium spatial transcriptomics is very low and, on many occasions, the spatial location remains vague. Indeed, without knowing the pattern of expression in model organism this study would have been very little informative. In general additional description/explanation would be desirable; for example, is the population of dermal fibroblast only present at proximal level as seen in Extended Fig. 5e?

Thank you. We have addressed this resolution limitation directly in our revised manuscript, adding an extensive discussion on the method's limitation (lines 676-688). We have attempted to buttress the predicted locations of cell types and key transcripts with several ISH experiments as listed in response to a previous point above. Specifically regarding dermal fibroblasts- we found that performing cell type matching on the PCW8.1 sample as opposed to the sample used in the initial submission addresses the point you make, and generally gives a clearer impression of the distribution of dermal fibroblasts throughout the limb (Extended Data Fig 2f, g).

4. It is unclear whether the AER cell state has been captured- In the limb ectoderm,

the gene expression characteristic of the AER is Fgf8 (also the other AER-FGFs, depending on the stage). However, only a small fraction of AER-basal cells seems to express Fgf8 (extended Fig. 2). The expression of SP8 and WNT6 does not define the AER as many non-AER cells do express these two genes in the limb bud ectoderm. Fig. 5b shows the spatial location of Fgf8 restricted to a couple of voxels in the distal ectoderm of PCW6.2. Maybe Fgf8 is already downregulated in the AER at this stage?

Thank you for this helpful insight. We have explored your point further on page 4 by:

- 1) Stating explicitly that only n=9 cells in the cluster were FGF8+ve (line 145), and that this may reflect downregulation at even the early sample stage as you say (line 646-647), or it may be a reflection of sampling breadth (ie single cell techniques not often capturing rare cell types; see discussion section for general overview on this- lines 653-675)
- 2) Performing RNA-ISH at PCW5 to search for FGF8 expression. This was a challenge as it would require us to section through the plane of the AER, but we did manage to capture small regions of ectoderm strongly expressing FGF8 at different time points (Extended Data Fig 2e and Fig. 5h,i).

In summary, I think these findings support your suggestion that even at our earliest sampling stages FGF8 is downregulated.

5. The cross-reference between the set of DEG between digit and interdigit and the list of 2300 genes with single gene conditions seems too simplistic. How many genes were intersected? It should be noted that the link of the expression pattern to the digits (i.e. IHH, NOG etc.) or interdigits is not sufficient to explain the phenotype (i.e. only some elements affected etc.). Many of these genes have multiple functions during limb development. For example, Dlx5 is expressed both in the AER/ectoderm and in the interdigits and their LOF impacts both digit patterning (split phenotype) and interdigit (syndactyly) but the correlation with the different domains of expression is unclear. The novelty of this analysis is the use of the human limb bud. Given the dynamism of expression patterns during limb development, the mouse stages in Extended data Fig. 7 are in general not comparable to human PCW6.2.

Thank you. Unfortunately, we were limited by publically available datasets for mouse ISH experiments, which do not match our human samples for developmental time. As these are not comparable, we have now removed them.

We have expanded on the cross-referencing of all known single gene disorders to give more detail on how this was done. The total number of overall matches can be seen in Extended Data Table 2, and genes highlighted in red in this table are represented as a heatmap in Figure 2d. We have manually selected conditions from this list to display in spatial data based on a) statistical strength of the trend and b) what the authors deemed clinically most relevant and interesting. Our intention in depicting the adult phenotype was not to draw a link between gene expression and phenotype but rather to inform the reader of the adult phenotype, in case this was not a field they were familiar with. However clearly this does not achieve this goal and so we have removed our drawings from the manuscript.

6. The spatially resolved heatmaps in Fig. 5 are difficult to understand. How many cells may be in a voxel? Shouldn't the spatial expression of receptor and ligand (Notch1/Jag1) be in the same voxel? Studies on model organisms and human

pathology indicate that FGFR specificity depends on their isoforms something not possible to analyze in the FGF/FGFR spatially resolved heatmaps.

Thank you for these comments. We have now improved the resolution of this analysis by performing RNA ISH for NOTCH and JAG, FGFs and WNT/FZD to more clearly investigate signaling in the limb (Fig. 5). The experiments indeed confirm Jag-Notch to be co-localised in keeping with the mechanism of signaling we briefly summarised in the text. Whilst we have performed the same for the relevant FGF family members and FGFR2, the assays available to us do not allow isoform specific testing. We have added a comment to make this clear as a limitation in the discussion section (lines 647-650).

7. Among the novel findings are the expression of Pitx1 in the FL proximal muscle (Fig. 4f), the expression of Fgf8 in proximal limb (Fig. 5d), and the distinct trajectories of embryonic and fetal myocytes. It would be helpful if the authors explore the human specificity of these features in the mouse.

We appreciate the reviewer's comment and now we have now performed more cross-examination between mouse datasets and human datasets, including the comparison of lineage progression in muscle. Unfortunately the Pitx1 graph had inadvertently had its x axis labels swapped, and so this finding was not something we intended to claim- this has now been corrected. We have expanded upon FGF8 expression in the mesenchyme in Extended Data Figs. 2 and 10.

Overall, We have altered this section significantly as detailed in this rebuttal, performing RNA-ISH to explore in more detail. The exploration of muscle across species has been added to the Extended Data Fig. 8. We also noted that the expression pattern of Pitx1 is also conserved in mouse (Rebuttal Figure 4).

Rebuttal Figure 4. The expression pattern of Pitx1 in mouse forelimb and hindlimb.

(A) Violin plot showing the expression level of Pitx1 in skeletal muscle lineage (left) and mesenchymal lineage (right) of mouse forelimb and hindlimb.

(B) Force-directed graph layout showing the expression level of Pitx1 in skeletal muscle lineage (top) and mesenchymal lineage (bottom) of mouse forelimb and hindlimb.

8. The study of the homology and divergence between human and mouse requires further explanation. The stages of the mouse samples should be indicated, the specific

stages and type of dissection used in the mouse/human comparison in Fig. 6e should also be indicated.

Thank you - we have added details of the mouse samples (see Fig. 6 and Extended data Fig. 10) to the figures together with dissection details (see methods line 726-751)

Minor comments:
- Fig. 1a seems to indicate that the limbs analyzed were FLs, rather than HLs as indicated throughout the manuscript

We apologise for this error and now we have corrected it.

- In Fig. 6f I don't detect any difference with mouse in the pattern of expression of HOXA11, HOXA11AS, HOXA13 as the authors seem to indicate in the text.

Thank you for highlighting this, we have altered the text accordingly.

- Line 99- "throughout the remainder of the first trimester in humans..." please adapt this sentence since to this point there was no time reference

Thank you – rewritten

- Lane 197- "...to analyze the entire lower limb..." should be the entire PD axis of the lower limb- Still only one section in the dorso-ventral axis is analyzed.

Thank you for highlighting this- corrected

Methods:

- Please, explain the rationale for the P/M/D or P/D dissection of the limbs and how this information was used in the analysis.

Now we have added the description of the rationale for the P/M/D (in order to avoid damaging the interzone and ensure its capture- line 406-409) or P/D dissection (pragmatic due to sample size). In analysis, this served as a simple sense check of certain annotations; specifically cell clusters which from marker gene analysis seemed to have a proximal or distal distribution (see rebuttal figure 5 below). Distal, RDH10 distal and transitional mesenchyme populations showed clear distal origin, with proximal mesenchyme from proximal sections (when dissected into sections) or whole thigh sections. Interzone cells showed a skew towards middle sections, but as expected show contributions from other regions, each of which contain forming synovial joints (eg hip, ankle, tarsometatarsal, metatarsophalangeal, interphalangeal). In addition, this information was used to investigate P/D axis genes described in Extended Data Figure 10e,f

Rebuttal Fig.5 A) bar plots showing proportions of cell types by sample region.

- Different software versions of Single-Cell Software Suite were used for human (v.2.1.1) and mouse (v.3.0.2). Equally for Python package Scanpy, v.1.6.0 for human and v.1.3.11 mouse. Has this difference been accounted for?

Thank you for highlighting this discrepancy. We have now unified our processing methods for human and mouse data (including HVG selection) so as to eliminate it (see methods updates- "Alignment, quantification and quality control of scRNA-seq data."- line 772 onwards). Interestingly this has resulted in increased similarities between the species, particularly in terms of immune cell abundance, which was previously different.

- Why was a threshold of 0.1 imposed for doublet analysis? How many doublets were removed?

Apologies - we have corrected this and rephrased the methods to improve the clarity of the method (line 790 onwards), adding in the 1,450 doublets explicitly for human data and 958 in murine data.

see Rebuttal Fig. 6 for visualisation of doublets:

Rebuttal Fig. 6

(A) Uniform Manifold Approximation and Projection (UMAP) embedding showing Benjamini–Hochberg adjusted p-value for doublet calling. Cells highlighted in light green were called as significant (total 1,450).

- Figure 6e: Please, indicate the scale of the size of the squares and circles and the color legend

We are very sorry for this missing information and now we have added it.

- Extended Figure 10 a: There is no legend for the x axis, which sample correspond to each violin plot?

We are very sorry for this missing information and now we have added it.

Reviewer Reports on the First Revision:

Referees' comments:

Referee #2 (Remarks to the Author):

I have carefully read your revised manuscript and the responses to my previous comments. I appreciate the effort you have made to address the concerns raised and improve the clarity of the manuscript. I am generally satisfied with your responses and the changes made in the manuscript. However, I would like to make a few additional comments and suggestions for further improvement:

1) it appears that the authors have made an effort to compare the two species in a comprehensive manner, highlighting similarities and differences in their findings. They have also acknowledged limitations in their study, such as the lack of samples from the earliest stages of limb bud development and the difficulty in unbiased cross-species comparisons. Regarding the unified processing methods for human and mouse data, it would be helpful if you could briefly describe the impact of these changes on the results, especially in cross-species comparisons. Did the unification of methods lead to any significant differences in the analysis or interpretation of the data?

2) The RNA ISH validation for JAG-NOTCH and FGF molecules is a significant addition to the manuscript. To further support the validation results, could you provide any quantitative metrics or analysis of the RNA ISH staining, such as co-localization scores or spatial correlation coefficients? This would help to strengthen the evidence for the interactions you have identified.

Referee #2 (Remarks to the Author – comments upon responses to Reviewer #1):

In response to the detailed concerns raised by Reviewer #1, the authors made several changes and clarifications in their manuscript. Overall, the authors took the reviewer's concerns seriously and made several amendments to their manuscript to address these points. They provided additional data, adjusted their analyses, and improved their methodological descriptions to address the reviewer's concerns and enhance the quality and clarity of their work.

On the depiction of chondrocyte differentiation: The authors agreed with the reviewer's criticism and made changes to their manuscript and figures. They altered their trajectory analysis and confirmed the continuous trajectory of chondrocyte differentiation in accordance with the established consensus. They also changed the terminology they used to describe the chondrocytes to better match the accepted language in the field.

Regarding PITX1 expression: The authors provided a detailed response, noting that their observation of PITX1 expression in the forelimb was unexpected, but that this discrepancy could be due to species-specific differences or methodological limitations. They further substantiated their findings with additional in situ hybridization analysis and careful reanalysis of their scRNA-seq data.

On cross-examination with mouse datasets: The authors agreed that further comparison with mouse

data could be informative. They explained that due to the extensive nature of this undertaking, they decided to focus on their human dataset for this paper. They are planning a separate, dedicated study for a comprehensive comparison between human and mouse limb development.

In terms of outlining the limitations of their methodological approaches: The authors agreed with the reviewer's critique and expanded on the limitations of their approaches, especially regarding the resolution of 10x Visium and scRNA-seq.

With respect to spatial transcriptomics: The authors acknowledged the resolution limitations of the 10X Visium platform and provided an additional sensitivity analysis to address the reviewer's concern. They also clarified their method for calculating p-values in differential expression analysis.

Concerning cell-cell communication analysis: The authors accepted the critique, admitting that their initial interpretation might have been overreaching given the data. They revised the relevant sections of the manuscript and figures to more accurately reflect the data and its limitations.

On the capture rate of single-cell transcriptomics: The authors acknowledged the limitations in their scRNA-seq data, such as the lack of hypertrophic chondrocytes. They performed additional analyses using supervised label prediction, demonstrating that rarer cell types were likely present in their data. They also included additional figures to show the reproducibility of cluster composition.

Regarding computational parameters for analysis: The authors provided a more detailed parameter description for cell clustering in the methods section and have changed the language to refer to cell groups as clusters for clarity.

About the justification of cluster annotations: The authors included a table of marker genes for each cell cluster with references, providing clear support for their annotations.

Referee #3 (Remarks to the Author):

The revisions made by the authors have satisfactorily clarified the issues I pointed out in the previous version of the manuscript. Overall, this manuscript presents an ambitious and comprehensive effort towards a detailed atlas of human limb development; the depth and breadth of the analyses are impressive, and I believe the manuscript will become a reference in the field.

As a comment, I would like to draw the authors' attention to certain aspects of limb development that may be worth reconsidering before publication. While I do not mean to suggest that changes are strictly necessary, I believe that the authors may find these suggestions helpful.

- The statement in lines 243-244 "Within the autopod, precise regulation of digit formation is mediated through interdigital tissue apoptosis" is not accurate. Indeed, syndactyly of normally formed digits can occur. The digit/IDS iterative pattern, which establishes initial digit patterning, is believed to rely on self-organization mechanisms, with the identity of each digit depending on Shh.

- The statement in lines 265-266: “with group 13 genes limited to the most distal part of the autopod” is also a little concerning as Hoxd13 and particularly Hoxa13, based on animal models, mark the whole autopod. The expressions in Extended Fig. 6f correspond to the second Hoxd wave and it is a little surprising the distal restriction of 5' Hoxd genes, more distally restricted than in mice. The limb schemes should fit the shape of PCW5.6 (equivalent to mouse E12 in mouse) rather than much earlier limbs in mouse (E10.5) and could depict the two Hoxd waves.

- In general it would be better if the limb diagrams reflect the stages more accurately to avoid misleading. For example, the profiles in Fig1a for PCW6.2 and in Fig.6a for E13.5 are rather confusing.

- In my opinion, the outline of digit 1 in Fig. 2 includes part of the interdigital space although the results and conclusions are concordant with existing knowledge.

Very minor:

- Adding vertical lines between the clusters in the heatmaps could be helpful for the reader.
- Fig. 4f- Forelimb section PCW5.6 instead of 5.5
- Fig. 5 legend- indicate asterisks. Also, 2 versions for panel c in this figure were upload.
- To make the figures self-explanatory, please indicate in the Extended Data legends whether the panels show immunofluorescence or in situ hybridization

Referee #3 (Remarks to the Author – comments upon responses to Reviewer #1):

Referee 1 comments: “posterior prevalence”

REV1: An example of misunderstanding of key terms is “posterior prevalence” (line 265-279), which the authors use in describing the spatial expression pattern of HOX genes. Posterior prevalence is a general property attributed to vertebrate HOX proteins describing the dominant effect of more posterior HOX proteins over the function of anterior orthologs in common (overlapping) areas of expression, a term coined by Denis Duboule in 1991 (D. Duboule, Patterning in the vertebrate limb, *Curr. Opin. Genet. Dev.*, 1:211-216) which has nothing to do with their spatial colinearity, or domains of gene expression as being used in the manuscript (e.g. line 311-312).

One error still persists regarding misuse of the term “posterior prevalence” (Lines 296-297):

“HOXD11 was downregulated in the great toe, in keeping with its posterior prevalence.” I suggest that the authors revise their statement to reflect the fact that HOXD11 expression exhibits a posterior bias, and does not extend to digit 1, similar to what has been observed in mice and chicks, where it never reaches the territory of digit 1.

Author Rebuttals to First Revision:

Reviewer #2:

I have carefully read your revised manuscript and the responses to my previous comments. I appreciate the effort you have made to address the concerns raised and improve the clarity of the manuscript. I am generally satisfied with your responses and the changes made in the manuscript. However, I would like to make a few additional comments and suggestions for further improvement:

It appears that the authors have made an effort to compare the two species in a comprehensive manner, highlighting similarities and differences in their findings. They have also acknowledged limitations in their study, such as the lack of samples from the earliest stages of limb bud development and the difficulty in unbiased cross-species comparisons. Regarding the unified processing methods for human and mouse data, it would be helpful if you could briefly describe the impact of these changes on the results, especially in cross-species comparisons. Did the unification of methods lead to any significant differences in the analysis or interpretation of the data?

The unification of methods indeed led to an important change in our data interpretation. 1) the isolation and identification of RDH10+ distal mesenchymal cells that are conserved between mice and humans. This population was previously only discovered in the mouse dataset. But after method unification, we now can see a similar population in the human limbs. This population is in the interdigital region of the foot plate supported by our RNAscope experiments. 2) We've now seen hypertrophic chondrocytes co-expressing MMP13 and COL10A1 in both humans and mice 3) We've now also seen neuronal cells in the human limbs at a very small number. But it's beyond this paper's scope to explore the peripheral nervous system in the limb in detail which we leave it to future work and collaboration.

The RNA ISH validation for JAG-NOTCH and FGF molecules is a significant addition to the manuscript. To further support the validation results, could you provide any quantitative metrics or analysis of the RNA ISH staining, such as co-localization scores or spatial correlation coefficients? This would help to strengthen the evidence for the interactions you have identified.

Thank you for this suggestion. We have now performed co-localisation analyses for both stainings (lines 516-519). This was performed by checking for expression of both molecules within the same pixel (ie sub-cellular resolution), and calculating the probability of both being present (see methods, lines 957-963). This showed strong colocalisation between ligands and receptors.

Reviewer #3:

The revisions made by the authors have satisfactorily clarified the issues I pointed out in the previous version of the manuscript. Overall, this manuscript presents an ambitious and comprehensive effort towards a detailed atlas of human limb development; the depth and breadth of the analyses are impressive, and I believe the manuscript will become a reference in the field.

As a comment, I would like to draw the authors' attention to certain aspects of limb development that may be worth reconsidering before publication. While I do not mean to suggest that changes are strictly necessary, I believe that the authors may find these suggestions helpful.

- The statement in lines 243-244 "Within the autopod, precise regulation of digit formation is mediated through interdigital tissue apoptosis" is not accurate. Indeed, syndactyly of normally formed digits can occur. The digit/IDS iterative pattern, which establishes initial digit patterning, is believed to rely on self-organization mechanisms, with the identity of each digit depending on Shh.

Thank you - apologies for this. It has now been corrected.

- The statement in lines 265-266: "with group 13 genes limited to the most distal part of the autopod" is also a little concerning as Hoxd13 and particularly Hoxa13, based on animal models, mark the whole autopod. The expressions in Extended Fig. 6f correspond to the second Hoxd wave and it is a little surprising the distal restriction of 5' Hoxd genes, more distally restricted than in mice. The limb schemes should fit the shape of PCW5.6 (equivalent to mouse E12 in mouse) rather than much earlier limbs in

mouse (E10.5) and could depict the two Hoxd waves.

Thank you. We have altered all of the schematics to attempt to reflect the true stage more accurately. We have corrected the text you have highlighted. We have updated the schematics to now show both HOX waves.

- In general, it would be better if the limb diagrams reflect the stages more accurately to avoid misleading. For example, the profiles in Fig 1a for PCW6.2 and in Fig.6a for E13.5 are rather confusing.

Thank you for this helpful comment. Now we've optimized the limb diagrams for human PCW6.2 and for mouse E13.5.

- In my opinion, the outline of digit 1 in Fig. 2 includes part of the interdigital space although the results and conclusions are concordant with existing knowledge.

Thanks for this comment, which further highlights the difficulties of working with spatial transcriptomics at its current resolution.

The spot clusters of spatial data were clustered by unsupervised clustering using the Louvain algorithm. When we clustered these spots of the spatial data with different resolution (1.0-4.0) by using Louvain, these spots persistently cluster together, suggesting that these spots have a high degree of similarity at the transcriptome level (Rebuttal Fig.1a). As you point out, some of these spots do appear to include the junction between the digital space. To check this, we tested the expression of known marker genes for chondrocyte (*COL2A1*, *SOX9*, *COL9A3*) and interdigital mesenchyme (*RDH10*, *BMP2*, *BMP7*) and confirmed that these spots (red Asterisk marks) do seem to contain transcripts from both (Rebuttal Fig.1b). However, the dominant transcripts are that of the chondrocyte lineage (Rebuttal Fig.1c), and so on balance we have kept the annotation as digit. We have added a final sentence to the limitation sections to mention this issue with visium (lines 696-698).

Rebuttal Fig.1 The spot clusters and gene expression level of spatial data of human hindlimb at stage of PCW6.2. **a.** Spatially resolved spots showing the clusters of spatial data with different resolution. **b.** Spatially resolved heatmaps across tissue sections showing spatial expression pattern of known marker genes for chondrocyte (*COL2A1*, *SOX9*, *COL9A3*) and interdigital mesenchyme (*RDH10*, *BMP2*, *BMP7*). **c.** UMAP plot showing the expression level of known marker genes for chondrocyte (*COL2A1*, *SOX9*, *COL9A3*) and interdigital mesenchyme (*RDH10*, *BMP2*, *BMP7*) in cluster 17 which was annotated as digit 1.

Very minor:

- Adding vertical lines between the clusters in the heatmaps could be helpful for the reader.

Thanks for this helpful comment. We've added the vertical lines between the clusters for each heatmap.

- Fig. 4f- Forelimb section PCW5.6 instead of 5.5

apologies if this was not clear - the stages indicated in Fig.4f are actually correct, and the forelimb and hindlimb sections are from two different embryos at PCW5.5 and 5.6, respectively.

- Fig. 5 legend- indicate asterisks. Also, 2 versions for panel c in this figure were upload.

Thank you- we have now added the asterisks description in the fig.5 legends. and uploaded the correct version of Fig.5.

- To make the figures self-explanatory, please indicate in the Extended Data legends whether the panels show immunofluorescence or in situ hybridization

Thank you- we have checked each figure and added the description of immunofluorescence or in situ hybridization in corresponding legends.

Comments upon Referee 1: “posterior prevalence”

An example of misunderstanding of key terms is “posterior prevalence” (line 265-279), which the authors use in describing the spatial expression pattern of HOX genes. Posterior prevalence is a general property attributed to vertebrate HOX proteins describing the dominant effect of more posterior HOX proteins over the function of anterior orthologs in common (overlapping) areas of expression, a term coined by Denis Duboule in 1991 (D. Duboule, Patterning in the vertebrate limb, Curr. Opin. Genet. Dev., 1:211-216) which has nothing to do with their spatial colinearity, or domains of gene expression as being used in the manuscript (e.g. line 311-312).

One error still persists regarding misuse of the term “posterior prevalence” (Lines 296-297): “*HOXD11* was downregulated in the great toe, in keeping with its posterior prevalence.” I suggest that the authors revise their statement to reflect the fact that *HOXD11* expression exhibits a posterior bias **and does not extend to digit 1**, similar to what has been observed in mice and chicks, where it never reaches the territory of digit 1.

Thank you. This is now corrected.

Reviewer Reports on the Second Revision:

Referees' comments:

Referee #2 (Remarks to the Author):

Upon revisiting your manuscript and thoroughly reviewing the detailed responses to the points raised in my and others prior reviews, I am gratified to observe the notable improvements you've brought to the paper. Your conscientious attention to each request and your comprehensive responses underscore your commitment to academic rigor and the enhancement of the manuscript's overall quality. This review finds me largely content with the progress authors have made, and it is evident that your revisions have substantially strengthened the manuscript.

Referee #3 (Remarks to the Author):

The authors have accurately addressed and incorporated my comments in the revised version of their paper.